# From oil field to geothermal reservoir: ~~First a~~Assessment for geothermal utilization of two regionally extensive Devonian carbonate aquifers in Alberta, Canada

Leandra M. Weydt[1], Claus-Dieter J. Heldmann[1], Hans G. Machel[2], Ingo Sass[1,3]

[1]Department of Geothermal Science and Technology, Technische Universität Darmstadt, Schnittspahnstraße 9, 64287 Darmstadt, Germany

[2]Earth and Atmospheric Sciences, University of Alberta, Edmonton, AB T6G 2E3, Canada

[3]Darmstadt Graduate School of Excellence Energy Science and Engineering, Jovanka-Bontschits-Straße 2, 64287 Darmstadt, Germany

*Correspondence to*: Leandra M. Weydt (weydt@geo.tu-darmstadt.de)

**Abstract**

The Canadian Province of Alberta has one of the highest per capita $CO_2$-equivalent emissions in Canada, predominantly due to industrial burning of coal for the generation of electricity and the mining operations in the oil sands deposits. Alberta's geothermal potential could reduce $CO_2$-emission by substituting at least some fossil fuels with geothermal energy.

The Upper Devonian carbonate aquifer systems within the Alberta Basin are promising target formations for geothermal energy. To assess their geothermal reservoir potential, detailed knowledge of the thermo- and petrophysical rock properties is needed. An analogue study was conducted on two regionally extensive Devonian carbonate aquifers, the Southesk-Cairn Carbonate Complex and the Rimbey-Meadowbrook Reef Trend, to furnish a preliminary assessment of the potential for geothermal utilization. Samples taken from outcrops were used as analogue to equivalent formations in the reservoir and correlated with core samples of the reservoir. Analogue studies enable determination and correlation of facies related rock properties to identify sedimentary, diagenetic, and structural variations, allowing more reliable reservoir property prediction. Rock samples were taken from several outcrops of Upper Devonian carbonates in the Rocky Mountain Front Ranges as well as from four drill cores from the stratigraphically equivalent Leduc and three drill cores of the slightly younger Nisku Formation in the subsurface of the Alberta Basin. The samples were analyzed for several thermo- and petrophysical properties, i.e., thermal conductivity, thermal diffusivity and heat capacity, as well as density, porosity and permeability. Furthermore, open-file petrophysical core data retrieved from the AccuMap database were used for correlation.

The results from both carbonate complexes indicate good reservoir conditions regarding geothermal utilization with an average reservoir porosity of about 8 %, average reservoir permeability between $10^{-12}$ and $10^{-15}$ m², and relatively high

thermal conductivities ranging from 3 to 5 W m$^{-1}$ K$^{-1}$. The most promising target reservoirs for hydrothermal utilisation are the completely dolomitized reef sections. The measured rock properties of the Leduc Formation in the subsurface show no significant differences between the Rimbey-Meadowbrook reef trend and the Southesk-Cairn Carbonate Complex. Differences between the dolomitized reef sections of the examined Leduc and Nisku Formation are also minor to insignificant, whereas the deeper basinal facies of the Nisku Formation differs significantly.

In contrast, the outcrop analogue samples have lower porosity and permeability, likely caused by low-grade metamorphism and deformation during the Laramide Orogeny that formed the Rocky Mountains. As such, the outcrop analogues are no valid proxies for the buried reservoirs in the Alberta Basin.

Taken together, all available data suggests that dolomitization enhanced the geothermal properties, but depositional patterns and other diagenetic events, e.g. fracturing, also played an important role.

## 1 Introduction

Canada currently emits about 730 Mt a$^{-1}$ $CO_2$-equivalent, of which the Province of Alberta emits nearly 300 Mt a$^{-1}$ (Environment Canada, 2016, 2017: last reliable numbers are for 2015). Therefore Alberta belongs to the five provinces in Canada with the highest emission rates (Environment Canada, 2016, 2017). The main reason for this pattern is the industrial generation of energy (electricity and heat) from coal and gas, which currently provide about 40 % each of the energy mix in this province, along with the huge mining operations of the oil sands deposits. The oil sands industry alone currently accounts for about 10 % of Canada's $CO_2$ -equivalent emissions (Canadas Oil Sands, 2017), tendency rising. However, the trend of increasing $CO_2$-emissions could be significantly reduced if alternative and/or renewable energy sources were implemented to a larger degree. For Canada to meet or at least approach the targets of the Paris Accord from 2015 regarding the reduction of $CO_2$-emissions, geothermal energy should become part of the energy mix in Alberta. This appears feasible because, although this province is characterized as a 'low enthalpy region' (Grasby et al., 2012; Jones et al., 1985, Lam and Jones, 1985 and 1986) with a moderate average geothermal gradient of 33.2 °C km$^{-1}$ and an average heat flow of 60.4 W m$^{-2}$ in the WCSB, recent studies using data from several tens of thousands of oil and gas wells suggest that at least some of the Upper Devonian carbonate aquifers are suitable for geothermal utilization (Weides and Majorowicz, 2014).

The area around the town site of Hinton in the western region of the Alberta Basin (Fig. 1) is of particular interest because well data analysis indicates flow rates of more than 400 m³ h$^{-1}$ and temperatures up to 150 °C at depths of approximately 5 km (Lam and Jones, 1985). While such conditions suggest a reasonable to good potential for geothermal utilization, previous studies provided very few hard data on the geothermal reservoir properties of the rocks (e.g. Weides and Majorowicz, 2014; Weides et al., 2013; Nieuwenhius et al., 2015; Ardakani and Schmitt, 2016).

Previous studies predominantly focused on determination of heat flow, geothermal gradients and reservoir temperature (e.g. Garland and Lennox, 1962; Majorowicz and Jessop, 1981; Lam et al., 1982; and more recent Majorowicz et al. 2012 and numerous more described in Weides et al., 2015), while only a few considered water chemistry and recovery (e. g. Lam and

Jones, 1985; 1986). More recent studies considered parameters like porosity and permeability (e.g. Weides et al., 2013, Weides and Majorowicz, 2014, Ardakani and Schmitt, 2016) or injection and production rates in combination with reservoir temperatures (Ferguson and Ufondu, 2017).

Within the study area, thermal properties measured on core samples exist only for the Hinton-Edson area. Beach et al. (1987)

gives a good overview of thermal conductivity of 13 different rock types of the Mesozoic, Cenozoic and Paleozoic sediments in this area of the Alberta Basin, but not for specific formations or how the properties might change within the reservoir. However, an extensive data base on parameters such as porosity, permeability, thermal conductivity, and others, is necessary for a geothermal assessment, especially with complete parameter sets for every single sample (Clauser, 2006; Sass and Götz, 2012; Homuth et al., 2015).

This paper reports the results of a pilot study that forms the first phase of a larger project entitled 'MalVonian' (semantically amalgamated from Malm Formation and Devonian Period), a joint undertaking by researchers from the Technische Universität Darmstadt and the University of Alberta. The main objective of Project MalVonian is to characterize the suitability of three regionally extensive carbonate aquifer systems for geothermal utilization, and perhaps to identify which of them is the most promising one with regard to heating homes or generating electricity. One of these aquifer systems is the

Upper Jurassic Malm-Aquifer in the foreland basin of the Alps in southern Germany, the other are two of the four Devonian aquifers in the Canadian foreland basin of the Rocky Mountains in Alberta (Fig. 3). Despite their different ages, these aquifer systems show many similarities regarding rock types, thicknesses, depth and deformation (both form structural homoclines in the subsurface), and hydrogeological properties.

The Malm-Aquifer in Southern Germany has already been proven to be suitable as a geothermal reservoir (Birner et al.,

2012; Homuth et al., 2015; Wolfgramm et al., 2017). However, the overall subsurface data base is relatively scarce because it is mainly based on drill cores from exploration wells, which of there are few in areas without hydrocarbons. There are 28 active geothermal wells around Munich (Böhm et al., 2013) and several more spread over a greater distance in Southern Germany. Therefore, in order to obtain a large enough data base for upscaling and correlation of geothermal reservoir characterization, Homuth et al. (2015) used samples not only from the four wells within the reservoir, but also from 19

outcrops in the shallow parts of the homocline. The samples were analyzed for thermal- and petrophysical properties like density, porosity, permeability, thermal conductivity, thermal diffusivity as well as specific heat capacity. Furthermore the samples were classified with respect to the dominant reservoir facies types in order to identify facies dependent trends. This approach is known as the '(outcrop) analogue study concept' (Homuth et al., 2015). This concept entails that properties of rock units in the target subsurface reservoir are compared to stratigraphically equivalent rock units in outcrop, which are an

analogue to the subsurface target. Such analogue studies offer a cost-effective opportunity in areas with a low density of drill holes, in order to investigate and correlate facies, diagenetic, petrophysical and thermophysical properties from outcrops to the subsurface. Using such a data base, a thermofacies classification enables the identification of heterogeneities and production zones and also includes detailed analyses of the lithologies and their thermophysical properties on different scales (1. macroscale = outcrops, 2. mesoscale = samples, 3. microscale = thin sections). After Sass and Götz (2012), the multi-

scale concept allows an extrapolation of the results into the deep subsurface (within a conceptual model) and thus enables more precise reservoir modelling and prediction.

In contrast to the Malm-Aquifer in southern Germany, the Devonian aquifer systems in Alberta have been investigated much more extensively courtesy of the oil and gas industry, which drilled more than 600000 wells in the Alberta Basin over the past seven decades. Following the discovery of the first prolific oil reservoir in a Devonian reef at the town site of Leduc in 1947, the Devonian section has been penetrated by more than 200000 drill holes, whereas the remainder were completed at higher stratigraphic levels. Tens of thousands of core meters are stored and publicly accessible at the Core Research Centre in Calgary, as is a giant data base on the drilled wells, including downhole well logs, results of drill stem tests, and petrophysical data (mainly porosity and permeability data from plugs). However, while facies, diagenesis, and structure are well characterized and found to be amazingly variable across the basin (e.g., Switzer et al., 1994; Machel, 2010), the potential of the Devonian strata as geothermal reservoirs has not been assessed or only in a very superficial manner (e.g. Weides and Majorowicz, 2014). Accurate thermal properties are critical parameters for a reliable geothermal assessment (Popov et al., 2016). The aim of this work was to create an initial data set of rock properties (relevant to geothermal modeling) specific to the Upper Devonian aquifer systems which have become of particular interest for geothermal utilization and also for identifying variations of rock properties within the reservoir.

**2 Project Framework and Study Area**

The first part of the project is an assessment of the geothermal potential of three Upper Devonian aquifer systems within the Alberta Basin. One is the Southesk-Cairn Carbonate Complex (SCCC), a carbonate platform that straddles the boundary between the Rocky Mountains and its foreland basin, in that the western part of the complex is exposed in several thrust sheets of the Rocky Mountain front ranges while the eastern part is buried deeply in the foreland basin (Fig. 1), where it hosts a series of oil and gas reservoirs in reefs that sit on top a platform. The second is the Rimbey-Meadowbrook Reef Trend (RMRT), which is a series of reefs sitting on top and along the western margin of another underlying carbonate platform, entirely located in the subsurface of south-central Alberta and stretching for several hundred kilometres (Fig. 1). Most of our work is on these two aquifers. In addition, for comparison we also investigated a small part of a third Devonian aquifer, the Nisku Reef Trend in west-central Alberta.

The platforms and juxtaposed reefs have been exploited for oil and gas for several decades, and they formed the backbone of the Canadian petroleum industry until the oil sands in east-central Alberta came on stream on a large scale in the 1990s (Switzer et al., 1994; Machel, 2010). Therefore the carbonate platforms are in focus of interest due to the general increasing public interest in repurposing abandoned oil and gas wells. The Southesk-Cairn Carbonate Complex was chosen, because a part of this complex is exposed in the Rocky Mountains and therefore it is suitable for an analogue study as defined above. The Rimbey-Meadowbrook Reef Trend, although not exposed anywhere, is similarly suited because the rocks are of the same stratigraphic age and lithology. Furthermore, the two largest cities of the province, Calgary and Edmonton, are located

very near and/or almost on top of this reef trend, hence, geothermal facilities in this reef trend, if realized, could serve well over half of the population of Alberta (about 2.2 million people). In addition, we also investigated samples from three cores of Nisku reefs located between the other two platforms (marked as 'Nisku Reef Trend' in Fig. 1), to have at least a few samples of a higher reservoir level for comparison.

Within the WCSB, the geothermal gradient and heat flow varies from 20 °C km$^{-1}$ to over 55 °C km$^{-1}$ and 30 mW m$^{-2}$ to 100 mW m$^{-2}$, respectively (Majorowicz et al., 2012; Weides and Majorowicz, 2014). Areas with higher heat flow values are identified in the northern part of the basin. Compared to the study area, these zones are sparsely populated and thus not chosen for this pilot study.

**3 Geological Framework**

The Western Canada Sedimentary Basin (WCSB) is a large geological feature that is located mainly in Alberta east of the Rocky Mountains and in the adjacent provinces of Saskatchewan and Manitoba, as well as in the northern United States and in northeastern British Columbia (Grasby et al., 2012). The WCSB basin is divided into the Alberta Basin and the Williston Basin, the border of which roughly coincides with the Alberta-Saskatchewan border. Our study was conducted entirely within the Alberta Basin, the relevant parts of which are shown in Figures 1 and 2.

The sedimentary evolution of the WCSB throughout the Phanerozoic was intimately related to the tectonic evolution of the region and the formation of the Rocky Mountains (Wendte, 1992; Price, 1994; Switzer et al., 1994). Accordingly, the Devonian succession was deposited on the passive margin of the ancestral North American continent under mostly subtropical, open-marine conditions. Stratigraphically the Devonian succession is subdivided into four groups, each containing a carbonate platform, and separated from one another by marls and shales. From top to bottom they have been

numbered and named by the oil industry as D1 (Wabamun Group), D2 (Winterburn Group, with the Nisku Formation as the main carbonate platform and associated reefs), D3 (Woodbend Group, with Leduc reefs and the underlying Cooking Lake carbonate platform), and the D4 (Beaverhill Lake Group) (Figs. 3 and 4). Our study is focused on the Leduc Formation as the main target formation, with marginal reference to the Nisku Formation.

The regional distribution of the D3 aquifers in the subsurface of Alberta (Fig. 1) is delineated by seismic. West of the limit of

the disturbed belt, which marks the current physiographic boundary between the foreland basin and the Rocky Mountains, D3-aged strata crop out in the Front Ranges, where they are torn up into segments that are exposed in a series of imbricated thrust sheets (schematically shown in Fig. 2). At the time of deposition, these strata extended much farther westward as an expansive platform into what is now British Columbia (as reconstructed in Fig. 1).

This region of the WCSB has undergone four orogenies since the Devonian period (1. Antler (Devonian-Carboniferous), 2.

Sonoma (Late Permian), 3. Columbian (Jurassic-Early Cretaceous) and 4. Laramide (Mid-Late Cretaceous-Tertiary); Machel, 2010), which ultimately resulted in the wedge-shaped, triangular geometry in cross section of the foreland basin and its sedimentary filling (Fig. 2), now generally referred to as the Alberta Basin. East of the limit of the disturbed belt the

sedimentary layers now form a structural homocline that dips westward, whereby the Devonian strata increase in depth from zero in the east, where they crop out in a few places, to more than 6 km near the limit of the disturbed belt in the west.

The D3 Cooking Lake Formation is about 75 m in thickness and forms the foundation for about one hundred, up to 260 m thick Leduc-aged reefal buildups, which are surrounded by basin-filling shales of the Duvernay, Majeau Lake and Ireton Formations (Amthor et al., 1993 and 1994). The overlying D2 Nisku platforms are much smaller in lateral extent than the underlying D3 platforms, and much thinner with max. about 80 m of thickness. The reefal facies in the Nisku Formation is called the Zeta Lake Member (e.g. Loegering and Machel, 2003, Fig. 4).

As is the case with most deeply buried carbonate sequences, the Devonian carbonates in Alberta underwent extensive diagenesis, so much so that their reservoir properties, be it for hydrocarbon storage or for geothermal utilization, are now dominated by diagenetic alterations. Both the Southesk-Cairn Carbonate Complex and the Rimbey-Meadowbrook Reef Trend were affected by more than 20 recognizable diagenetic processes, including multiple phases of dissolution and cementation (mainly by calcite, dolomite, and anhydrite), pervasive dolomitization, and in the deeper part of the basin also by thermochemical sulfate reduction and injection of metamorphic fluids via squeegee-type fluid flow (Amthor et al., 1993 and 1994; Mountjoy et al., 1999 and 2001; Machel and Buschkuehle, 2008, Machel, 2010; Kuflevskyi, 2015).

Hydrologically, the Cooking Lake and Nisku platforms both belong to the Upper Devonian Hydrogeological Group (UDHG) (Rostron et al., 1997). This group consists of four aquifers (D1 – D4, Fig. 3) separated by three aquitards, underlain by shales and marls of the Waterways Formation and overlain by Carboniferous shales. Within the Southesk-Cairn Carbonate Complex, the aquitards within the UDHG are thinning towards the Rocky Mountains and are absent in some areas, thus allowing for hydrologic connection between the aquifers in the system.

Parts of the Upper Devonian succession are exposed in the Rocky Mountain Front Ranges. Of particular interest for the current study is the Southesk-Cairn Carbonate Complex in an area west and southwest of Hinton (as marked in Fig. 1), where it is relatively easily accessible at or near several roadcuts. However, except for a few old studies from remote and almost inaccessible areas such as the Ancient Wall and Miette reef complexes (Mountjoy, 1965; Mountjoy and McKenzie, 1974; Mattes and Mountjoy, 1980), only one 'modern' study is available that provided data on the diagenetic alteration of outcrops in this region, i.e., from Nigel Peak (Köster et al., 2008).

**4 Material and Methods**

Subsurface-outcrop analog studies as defined previously have been proven successful for geothermal exploration in the German Malm Formation (Homuth et al., 2015). A similar data base of thermo- and petrophysical rock properties was furnished for the outcrop samples from Alberta at the laboratory of the Technische Universität Darmstadt (Germany). Seven cores of the target reservoir formations were investigated in the Calgary Research Centre (CRC) and at the University of Alberta core lab in Edmonton. Table 1 gives the source locations and associated reef complexes of the cores and outcrops.

The exact coordinates are added to Appendix B. For closer examination, thin sections made from representative plugs of the analogue- and reservoir samples were prepared and petrographically analyzed.

Additional data was taken from the AccuMap database (IHS Markit, 2017) for validation and correlation with the lab measurements obtained in the course of this study. These core analyses include particle density, permeability and porosity.

As no outcrops of the Upper Devonian exist within the Alberta Basin and no outcrops of the Rimbey-Meadowbrook Reef Trend (RMRT) exist in the Front Ranges (see Fig. 1), analogue samples were taken from three outcrops each of the Southesk Cairn Carbonate Complex and from the adjacent Fairholme Complex in the Front Ranges.

Petrophysical parameters, i.e., density, porosity, and permeability, as well as thermophysical parameters, i.e., thermal conductivity, thermal diffusivity and specific heat capacity, were measured on these samples. In total 141 plugs were drilled
from 38 outcrop samples. All the parameters listed previously were measured on each plug for direct correlation whenever possible. To ensure reproducibility the plugs were measured in oven dry conditions and cooled down to room temperature in a desiccator (20 °C).

Density, porosity and permeability were measured after Hornung and Aigner (2004) four times for each sample. Particle and bulk density measurements were accomplished by using a helium pycnometer and a powder pycnometer, thereby measuring
the particle and bulk volume, respectively. Porosities were calculated from the resulting differences in volume. The accuracy of the method is 1.1 % (Micromeritmics, 1997 and 1998). For direct comparison with the provided data in the AccuMap data base only particle density is presented here.

Matrix permeability of the outcrop samples was determined with a column permeameter using different air pressure levels from 1 to 3 bar. This method is based on Darcy's law enhanced by factors for compressibility and viscosity of gases (Jaritz,
1999). It allows calculation of the intrinsic permeability ($K_i$) from apparent permeability ($K_a$) by using the Klinkenberg method (Klinkenberg, 1941). Thereby, intrinsic permeability describes the aquifer matrix only and does not consider fluid properties (Languth and Voigt, 2004). The intrinsic permeability corresponds to the effective gas permeability of air under infinitely high pressure. As it is not possible to determine permeability under infinitely high pressure, the column permeameter measures the apparent gas permeability of air with at least five pressure stages (Jaritz, 1999). Afterwards, the
apparent permeability is plotted in the Klinenberg plot to calculate the intrinsic permeability. Measurement accuracy varies from 5 % for high permeable rocks ($K \geq 10^{-14}$ m²) to 400 % for impermeable rocks ($K \leq 10^{-16}$ m²) (Filomena et al., 2014).

For determination of thermal conductivity and thermal diffusivity a thermal conductivity scanner was used (Popov et al., 1999), allowing non-destructive as well as contactless measurements by using infrared sensors. To minimize the transmission of optical heater radiation to reference standards and rock samples resulting from optical transparent surfaces
(Popov et al., 2016), black paint was applied along a scan line on the sample surfaces as well as on the standards. Both parameters were measured three to four times on each plug (Fig. 5). The measurement accuracy is 3 % (Lippman and Rauen, 2009).

Specific heat capacity was measured with a heat-flux differential scanning calorimeter after Schellschmidt (1999). Crushed pieces of every rock sample were heated at a steady rate from 20 °C up to 200 °C within a period of 24 h. Specific heat

capacities were derived from the resulting temperature curves through heat flow differences. The measurement accuracy is 1 % (Setaram Instrumentation, 2009).

For direct comparison with the target reservoir, cores for each area were selected from the AccuMap database. For both carbonate complexes (SCCC and RMRT) two cores of the D3 Leduc Formation were analyzed. The associated wells 5-22 (RMRT) and 16-18 (SCCC) are located in the central portion of the Alberta Basin, whereby the wells 10-31 (RMRT) and 2-36 (SCCC) represent areas close to the Rocky Mountains (see Fig. 1). To analyse differences between the D3 and the overlying D2 Devonian aquifer systems, three representative cores of the D2 Nisku Formation were examined at the University of Alberta corelab. These wells are located within the Nisku Reef Trend (Brazeau area, Fig. 1) and represent the reef (7-33), the bank-edge reef (2-19) and the bank (11-32), respectively (Fig. 4) (bank = platform in this context). The cores of the reef (7-33) and the bank-edge reef (2-19) comprise the Lobstick, Bigoray and Zeta Lake Member, while the bank (11-32) contains core samples of the Dismal Creek Member.

The study included further core analyses and measurements of thermal conductivity and permeability on core samples 5 cm to 70 cm long. Thermal conductivity was determined on the mantle surface at dry conditions in the same procedure described for the outcrop analogue samples (as shown in Fig. 5). Permeability was measured using a mini permeameter (Hornung and Aigner, 2004), a portable variation of the column permeameter, which allows point measurements on the planar or mantle surface of each core sample. The device allows measurements only at one air pressure level; therefore, the measured values represent apparent permeabilty. To obtain intrinsic permeability values, it would have been necessary to remeasure a representative selection of core samples by using the column permeameter. At this stage of the project, it was not possible to remove samples of the analyzed cores for this purpose. For this reason, the apparent permeability values represent the range of magnitude only. After Jaritz (1999) the difference between apparent and intrinsic permeability is smaller than 13 % for values greater than 50 mD ($\sim 5\cdot10^{-15}$ m²). Apparent permeability was measured at centimeter intervals on the mantle surface of each core sample.

In the course of petrographic analysis, the limestone textures were addressed using the classification of Dunham (1962) in combination with Embry and Klovan (1971). Grain size was addressed using Folk (1962), pore types using Choquette and Pray (1970), and pore sizes using Luo and Machel (1995). The dolomites/dolostones were classified using Sibley and Gregg (1987). Representative photographs of each analyzed formation in the field or in the core laboratories are added to Appendix A.

For an initial evaluation of the geothermal potential (listed in Table 2), the rock properties of the outcrop and core samples (Fig. 8 to 10) were classified into five levels of potential, as previously done in a 3D structural model of the German federal state of Hesse (Bär et al., 2011; Arndt et al., 2011; Bär and Sass, 2014). Within the 3D structural model, volumetric stratigraphic grids (SGrids) were created for each particular unit. After parameterizing the grids based on an extensive database of petro- and thermophysical rock properties for each unit combined with data from more than 4150 wells (e. g. results of pump tests and in-situ temperature measurements), a multi criteria approach was used to incorporate the relevance of the rock and reservoir properties for geothermal systems (Arndt et al, 2011). Threshold values ranging from 'very low' to

'very high' were defined for every parameter to specify the geothermal potential. These are based on experience in geothermal exploitation in Germany and particularly consider technical and economic factors. For example, the minimum temperature for district heating is defined as 60 °C and 100 °C defines the minimum temperature where electricity productionis technically possible. At temperatures above 120 °C (in combination with production rates above 50 m³/h),
electricity production becomes economically intresting. The classification of potential is explained in detail in Arndt et al. (2011). Reservoir properties are not considered here.

## 5 Results

### 5.1 Petrography Subsurface Core Samples (Figure 6, Plates 1 and 2)

The investigated Leduc Formation mainly represents intensively dolomitized stromatoporoid and coral-rich reefs and reef
margin lithologies with dissolution enlarged vugs and molds. The formation consists of dark grey to light grey, medium to coarse crystalline skeletal wackestones to floatstones/rudstones and especially in well 5-22 Amphipora grainstones (Plate 1C). Well 10-31 (Leduc, RMRT) constitutes the exception (Plate 1F), representing one of the deepest sections of the reef at ~ 4300 m bgl (= below ground level, here and thereafter). It is located within the Strachan pool close to the fold and thrust belt and comprises partially dolomitized to completely dolomitized permeable zones with interbedded nonporous limestones
(Mountjoy and Marquez, 1997).

The investigated Nisku cores comprise four members. The Lobstick Member mainly contains dark grey to black, fine crystalline, nonporous argillaceous limestones with a lenticular fabric (Plate 2F). The Bigoray Member constitutes of fine bedded, coarse crystalline dolomitized mud. Frequently, dark grey, fine crystalline limestone breccia are interbedded in the dolomitized mud (Plate 2D and 2E) and dissolution seams are quiet common. Well preserved corals, mollusks or small sized
vugs filled with anhydrite or calcite are present at some depth intervals. The Zeta Lake Member represents the reef and comprises light grey, fine crystalline, nearly nonporous dolomudstones to bioclastic floatstones/rudstones with molds of a variety of corals, brachiopods and stromatoporoids (Plate 2A – 2C). Last but not least, the Dismal Creek Member comprises light grey, nonporous, fine crystalline dolomudstones with abundant dissolution seams and argillaceous dolomudstone breccia with anhydrite nodules (Plate 2G – 2I).

Different types of fractures are present in all investigated wells. The Nisku and especially the Leduc Formation in the deeper parts of the basin contain abundant (1) subhorizontal to vertical hairline factures, commonly filled with bitumen that connect vugs and molds especially within the floatstones, packstones and rudstones (Fig. 6d) and to a lesser extent (2) mm to cm wide fractures that are mainly filled with calcite and occur randomly throughout the matrix (Fig. 6c). Additionally to type 1, narrow fractures and fissures, approximately <0.5 – 1 mm wide, are present within the floatstones, packstones and rudstones
of the upper third part of well 5-22 (Leduc, RMRT). These features tend to extend radially from vugs and molds (Fig. 6a) and form a network throughout the matrix, in places filled with bitumen (Fig. 6b). These networks probably explain the high vertical permeability values up to $10^{-10}$ m² in well 5-22.

In the Strachan pool (well 10-31, Leduc) close to the Rocky Mountains in the deepest part of the RMRT, bitumen commonly fills such fractures. Marquez and Mountjoy (1996) defined three types of hairline fractures in this area (1. = subhorizontal and subvertical, 2. = radial around vugs and molds, 3. = random in the matrix). These fractures probably resulted from overpressuring of the well-sealed reservoir during deep burial driven by thermal cracking of crude oil to gas, possibly aided

by tectonic compression, both happening at the same time during the Late Cretaceous - Early Tertiary coninciding with the peak phase of the Laramide orogeny (appr. 80 -55 Ma., English and Johnston, 2004). After Marquez and Mountjoy (1996), such hairline fractures are restricted to well-sealed buildups at depths greater than ~3500 m bgl. The Wizard Lake pool of the RMRT (5-22, central basin) never reached such burial depths (Machel, 2010), thus microfractures in this pool might have some other origin.

Permeability-reducing cementation and/or replacement by anhydrite is present within the Leduc Formation in the deeper core intervals of wells 10-31 and 2-36, and also in all Nisku cores. Most anhydrite appears as milky-white, some as clear, coarse-crystalline masses with cm-dimensions (especially in wells 7-33 and 2-19, Nisku), or as nodules (in well 10-31, as described in Machel, 1993). In addition, coarse-crystalline, white calcite is present as a late-diagenetic cement in almost all cores. Together, the late diagenetic precipiates of bitumen, anhydrite, and calcite all serve to reduce pre-existing porosity by

up to 70 % at some core intervals.

**5.2 Outcrop petrography (Figure 7, Plate 3)**

All investigated outcrops are situated in the Front Ranges in close proximity to major thrust faults.

The outcrops consist mainly of dark grey to mouse grey reefal carbonates adjacent to or with interfingering dark grey, argillaceous off-reef sections. Most of the outcrops expose nonporous, massive carbonate rocks equivalent to the Leduc

Formation (D3). Rocks equivalent to the Nisku Formation (D2) are exposed only at Toma Creek and Nigel Peak. Most of the exposed strata are pervasively foliated, jointed and fractured, with the faults dipping mostly west to southwestward (Plate 3A). Nigel Peak is the only location, where the rocks dip northeastward. The intense deformation of these rocks is likely a result of - and reflects close proximity to - major thrust faults. Generally, deformation decreases with larger distance to the thrust faults. Almost all collected rock samples comprise calcite filled veins and hairline fractures. Hydraulic conductivity in

the outcrops seems to be bound to fracture networks. Tectonic compression during the Laramide orogeny created different sets of fractures that haven't been observed in the subsurface core samples. However, it is presumable that tectonism also enhanced the deep reservoir close to the Thrust and Fold belt of the Rocky Mountains.

The investigated parts of the Fairholme Complex are partially to pervasively dolomitized, while the stratigraphically equivalent parts of the SCCC are largely present as limestones. Dolomitization is only recognized here along or in close

vicinity to fault planes (Plate 3C). Exceptions are the pervasively dolomitized Cairn and Grotto Formation at Toma Creek. The off-reef sections of both carbonate complexes are represented as argillaceous limestones.

Dolomitization, metamorphism and tectonism affected the rocks in all outcrops, but to a different degree. Consequently, this leads to vast differences in the formation's appearance and properties in the outcrops as shown in Fig. 7. These differences, even on a very small scale, affect the rock properties and are reflected in the results (Fig. 8 and Table 2).

### 5.3 Thin Sections (Plates 4-6)

Almost all rocks in the RMRT, both of the reefal Leduc Formation and of the underlying Cooking Lake platform, are comprised of grey and brownish, nonplanar to planar-s, medium to coarse crystalline dolomite (Amthor et al. 1993, 1994; Kuflevskiy, 2015). Most porosity is intercrystal. Independent of sample depth, secondary (dissolution) voids (0.5 – 3 mm in diameter) are common and connected three-dimensionally in most samples (Plate 4). Zoned or unzoned dolomite cement, partially filling larger secondary pores, is abundant. Many pore rims are coated with bitumen.

The Nisku Formation has similar petrographic characteristics (Plate 5), except for core intervals that had escaped dolomitization partially or completely. For example, in the partially dolomitized sections of the Bigoray Member in wells 7-33 and 2-19, brownish micritic matrix predominates, and dolomite forms only small clusters. Overall, matrix-selective dolomitization was the most diagenetic important process that affected these rocks, followed by precipitation of anhydrite, calcite, and bitumen, volumetrically in this order of abundance.

Matrix-selective, fine crystalline grey matrix dolomite is the most common dolomite type in the outcrop analogue samples (Plate 6). Primary porosity seems to be completely absent. Nonplanar grey matrix dolomite with grain sizes up to 700 µm and cm-sized secondary pores dominates in the Peechee Formation at Grassi Lakes and in the Cairn Formation at Toma Creek (Plate 6, B and D). The rocks at these locations are similar to the investigated subsurface reservoir samples. Two

fracture phases can be identified: Narrow veinlets (µm-size, Plate 6C) that are cross-cut by larger fractures (0.5 mm – 2 mm, better recognizable in hand specimens), both filled with calcite cements postdating stylolitization. Both events were observed in almost all samples of Jasper Railroad outcrop. At some outcrops they occur only in single formations (Perdrix Formation at Nigel Peak) or the second phase appears solitary. A third calcite cementation phase is indicated by white sparry calcite (Appendix A Plate 6F) in Dysphillid corals of the Grotto Formation as described in Koester et al. (2008).

Calcite twins occur in all cementation phases confirming progressed deformation and diagenetic alteration (Ferrill, 2004). Bitumen is observed in intercrystal pores or spread over the matrix as small spots with varying amount in all formations.
The rocks of the ouctrops, here considered as analogues to the subsurface reservoirs, were affected by similar diagenetic processes as those in the Leduc and Nisku Formations (e.g. Mountjoy et al., 1999 and 2001). However, there also are distinct differences. Most important in the current context, limestones are much more common in the outcrops.

### 5.4 Thermo- and petrophysical parameters

### 5.4.1 Density and Porosity

Core analyses taken from the AccuMap database including grain density, porosity and permeability, predominantly represent the dolomitized upper sections of the analyzed cores. Thus, the data is not representative for the whole formations. The
results (Fig. 8a) indicate constant rock density from 2.80 to 2.85 g cm$^{-3}$ for almost all investigated cores. Well 10-31 (Leduc, RMRT) constitutes the exception comprising partially dolomitized to completely dolomitized permeable zones with interbedded nonporous limestones.

Porosity ranges from less than 1 % up to 23 %. Porosity values below 5 % (e.g. the bank facies, well 11-32, Nisku) were found in dense mudstones/wackestones. Reefal zones (floatstones/packstones/rudstones) show an increased porosity (well 7-
33, Nisku). The bank-edge reef (2-19, Nisku) features intermediate porosity from 2.5 % to 10 %. Regarding the whole basin, porosity in the dolomitized sections is nearly independent of depth but is controlled by depositional patterns. Even though originating from greater depth (~ 4100 m bgl) the rocks of the western Leduc Formation (well 2-36, SCCC) indicate the highest discovered porosity in comparison to more central basin samples of the Leduc Formation (well 16-18, SCCC at 2700 m depth).

Dolomite rocks seem to be more resistant to chemical compaction during burial than limestones, retaining porosity. This generally observed phenomenon has already been described for the Leduc Formation in Mountjoy et al. (2001) and Amthor et al. (1994). Low porosity (< 5 – 10 %) in core profiles might also indicate diagenetic processes like post-depositional cementation, bitumen plugging, anhydrite cementation and replacement (Drivet and Mountjoy, 1997). Regarding well 10-31, the Leduc Formation's hydrothermal reservoir properties are considerably positive in two selective layers only (Fig. 8a),
constituting an exception to the other wells.

### 5.4.2 Thermal Conductivity and Permeability

Thermal conductivity of the reservoir samples varies between 2.4 W m$^{-1}$ K$^{-1}$ (bank-edge reef, well 2-19, Nisku) and 5.5 W m$^{-1}$ K$^{-1}$ (reef facies, well 7-33, Nisku). Table 3 provides an overview of average thermal conductivities (arithmetic mean and standard deviation) for each analyzed member and formation. There is no difference between the Leduc Formation of the
RMRT and the SCCC. Within the Nisku Formation, the argillaceous limestones of the Lobstick Member show the lowest thermal conductivity, while thermal conductivity of the Zeta Lake Member is within the range of the Leduc Formation. Thermal conductivity of the analogue outcrop samples is significantly lower. The Fairholme Complex analogue samples indicate a difference between off-reef (argillaceous limestones from the Perdrix and Mt. Hawk Formation) and dolomitized reef areas: The formations representing the off-reef, show the lowest density and thermal conductivity ranging about
2.6 g cm$^{-3}$ and 2.4 - 3.0 W m$^{-1}$ K$^{-1}$, respectively. The rock density of the dolomitized reef areas is within the range of the subsurface reservoir samples, but thermal conductivity is significantly lower. Likewise, the thermo-physical properties seem to be density controlled and increase with decreasing clay content and increasing dolomite content (Table 2). However,

results of the analogue samples taken from SCCC don't follow this trend and don't show any facies dependencies. Samples taken from the outcrops Jasper Railroad and Mt. Greenock represent mainly limestones or dolomitic limestones. The sample set shows no correlation between thermal conductivity and porosity Fig. 8c)

The analogue samples reveal loss of porosity, likely due to the low grade metamorphosis during orogeny of the Rocky Mountains. Rock permeability of the analogue samples is low to very low ($10^{-18}$ m² to $10^{-15}$ m²). Permeability is restricted to secondary or tertiary porosity from faults and fractures or karstification (Fig. 8d). Thin section analyses of representative samples confirm the very low permeability caused by almost no (inter-granular) porosity and sealed fissures (Plate 6). Due to the tectonic overprint, the analogue samples indicate low potential for hydrothermal utilization, only. They reflect no real reservoir conditions depict in Fig. 9 to 11 and make the measured rock properties unsuitable creating a conceptual model. Instead, reservoir data show increased permeability with increasing porosity probably as an effect of connected pore space. After Sass and Götz (2012) permeability varying from $10^{-15}$ m² up to $10^{-10}$ m² can be categorized as low permeable to permeable. In the deeper basin sections permeability is affected by fractures and faults (e.g. bank facies, well 11-32, Nisku), while the influence of matrix porosity increases within the dolomitized reef sections. The permeability value of $10^{-12}$ m² indicates the transition from low permeable to permeable (and thus from turbulent to laminar fluid flow depending on the hydraulic gradient). According to the thermofacies concept (Sass and Götz, 2012) the Leduc and Nisku Formation can be classified as 'transitional systems'. Transitional systems are defined as predominantly low permeable geothermal reservoirs, where stimulation is needed for economic and technical reasons. The highly permeable, extensively dolomitized reef zones represent promising reservoirs for hydrothermal utilization with predominantly convective heat and laminar fluid flow.

Correlation of thermal conductivity and permeability, confirms the higher matrix variability of the reef facies of the Nisku Formation (well 7-33) compared to the Leduc Formation (Fig. 8.e).

### 5.4.3 Subsurface Core Profiles (Figures 8 - 10, Plates 1 and 2)

Regarding the rock property measurements (core sample scale) of the Leduc Formation (Fig. 10 and 11) at reservoir scale (Kuflevskiy, 2015) they indicate rather a homogenous matrix (with permeability deviating by three orders of magnitude and porosity varying between 5 to 10 %), in rocks comprised predominantly of stromatoporoid- and coral-rich grainstones/wackestones to floatstones/rudstones. Regarding the rock properties at meter scale, permeability and porosity are strongly correlated to the original fabric type depending on pore type and pore size (described in detail in Drivet, 1993), that leads to a higher horizontal permeability than vertical permeability. One remarkable exception is well 5-22 (central basin, RMRT) with higher vertical permeability values indicating a good interconnection between scattered vugs and molds and in combination with narrow fractures. The thermophysical rock properties of 5-22 indicate a high to very high potential for geothermal utilization (Fig. 11).

Measurements of rocks from the Nisku Formation (well 7-33, Fig. 9) show significant differences for thermal conductivity and apparent permeability between the off-reef Lobstick Member, which consists of dark grey limestones (mainly mudstones/wackestones), and the dolomitized reef sections of the Zeta Lake Member, which consists of floatstones,

rudstones and packstones. Thermal conductivity and apparent permeability show a positive correlation (Fig. 9), likely due to the higher dolomite content. Data of intrinsic permeability and porosity taken from the AccuMap database follow the same trend (here only available for the Zeta Lake Member). Thermal conductivity seems to be independent of depth and controlled mainly by mineral content, i.e., the extent of dolomitization, but also by grain size. After Clauser and Huegens (1995) pure dolomite has a higher thermal conductivity ($\sim$ 5.0 W m$^{-1}$ K$^{-1}$) than calcite ($\sim$ 3.40 W m$^{-1}$ K$^{-1}$), while clay minerals possess very low thermal conductivities (1.9 – 2.6 W m$^{-1}$ K$^{-1}$). A thermal conductivity above 5.0 W m$^{-1}$ K$^{-1}$ might also result from reduced conduction resistances caused by larger dolomite grains ($\sim$ 700 µm) with reduced grain surface-volume-ratios in the absence of pore space. The extensively dolomitized core intervals between 3060 m and 3080 m depth (Fig. 9) have thermal conductivity values above 4.5 W m$^{-1}$ K$^{-1}$ and possess the highest porosity and permeability values within the permeable range. The same pattern is present in well 2-19 (bank-edge reef).

The here presented data set was analyzed under lab conditions (20 °C) and represents matrix properties only (representative at cm-scale, macroscale). It is to emphasize that rock property measurements can differ with samples size. Measurements at plug scale do not represent larger features as large vugs and molds (bigger than sample size), karstification or fracture zones and most likely underestimate porosity or permeability of the outcrop/reservoir. Especially in carbonate reservoir porosity and permeability can be very variable.

In order to utilize the rock properties in a geological model, they need to be corrected for reservoir conditions and subsequently transferred to reservoir scale (macroscale). For example thermal conductivity: With given information about reservoir fluid properties and porosity, thermal conductivity can be recalculated for water-saturated conditions (Clauser and Huenges, 1995; Popov et al., 2003). Temperature dependency models are used to transfer thermal properties to reservoir conditions as described in Vosteen and Schellschmidt, (2003) and Sommerton, (1992). In general, thermal conductivity increases with increasing water content, porosity and pressure, but decreases with increasing temperature (Clauser and Huenges, 1995). For example, thermal conductivity of matrix dominated limestones of the Jurassic Malm Formation in Southern Germany ranges between 1.35 – 2.62 W m$^{-1}$ K$^{-1}$ at 20 °C and dry conditions, 1.60 – 2.79 W m$^{-1}$ K$^{-1}$ at 20 °C and saturated conditions and 1.41 – 2.25 W m$^{-1}$ K$^{-1}$ at reservoir conditions (150°C). Geothermal reservoirs have a smaller margin to be economically profitable than oil reservoirs. Therefore Rühaak et al. (2015) tested different upscaling procedures for thermal conductivity to testify their accuracy. Thereby the intention is to keep as much of the small scale information as possible. The results indicate that harmonic and geometric mean upscaled values reflect most accurately local values.

A reliable porosity/permeability prediction is crucial for reservoir characterization and modeling (Borgomano et al., 2008) and mostly has to be carried out with a limited number of core measurements. Upscaling techniques for porosity, permeability and hydraulic conductivity have been subject of several studies and are described in Clauser (1992), Renard and de Marsily (1997) and Farmer (2002). Previous studies in the WCSB analyzed a high amount of well data and used the calculation of arithmetic and geometric mean values to upscale their parameters from plug to reservoir scale (Weides et al., 2013, Ardakani and Schmitt, 2016).

Borgomano et al. (2013) analyzed the porosity-permeability relationship of plug samples combined with detailed facies analysis to identify the predictability of these parameters. Additionally, the creation of vertical and horizontal histograms and variograms can help to identify the heterogeneity, anisotropy and lateral distribution of the properties and whether the upscaling from plug to reservoir scale is linear or not. However, the diagenetic overprint within the analyzed Upper Devonian aquifer systems makes a prediction even more difficult.

**6 Discussion and Conclusions**

The investigated parts of the Leduc Formation in the Rimbey-Meadowbrook Reef Trend and the Southesk-Cairn Carbonate Complex are stromatoporoid- and coral-rich reef core to reef margin lithologies. They are pervasively dolomitized and relatively homogenous across the whole study area (Kuflevskiy, 2015; Amthor et al., 1994), except in some reefs in the deepest part of the basin close to the fold and thrust belt, where limestone intervals are interbedded with dolostones (Marquez and Mountjoy, 1996).

The dolomitized reef sections in both aquifers show minor differences in term of the measured rock properties. According to the thermofacies concept, both aquifer systems can be categorized as 'transitional systems' (Sass and Götz, 2012). The thermofacies concept was developed for low and mid enthalpy systems and is based on rock property measurements of about 11000 plugs drilled from sedimentary rocks in Central Europe. According to these findings reservoir characteristics are strongly influenced by the facies type and thus serves to distinguish between hydrothermal, transitional/enhanced and petrothermal systems. Transitional systems are defined as mainly low permeable, low to mid enthalpy geothermal reservoirs with permeability lower than $10^{-12}$ m², where stimulation is needed for economic and technical reasons.

The main objective of this study is to identify areas and/or depth intervals that are promising for geothermal utilization. Based on our current data base, only some areas, notably the extensively dolomitized and highly fractured reef zones (floatstone, packstone and rudstone facies) with permeability values up to $10^{-12}$ m² and thermal conductivity values $> 4$ W m$^{-1}$ K$^{-1}$, are promising reservoirs for hydrothermal utilization. Nonetheless, rock property measurements produce more or less conservative results and represent matrix properties only. Including additional information on stress field, secondary porosity and karstification into a 3D structural model will lead to a more detailed reservoir prediction for geothermal reservoirs.

The investigated parts of the Nisku Formation are comprised of a variety of lithofacies with variable reservoir properties. The only part of the Nisku that possess reservoir properties similar to the Leduc is the dolomitized Zeta Lake Member.

The measured thermal conductivity values presented in this work are within the range of data sets included in previous studies. Some examples are given in Table 4. Further data sets are also listed in Grasby et al. (2012). Measurements of thermal conductivity of whole drill cores have shown that thermal conductivity is independent of depth in the study area. Additionally, thermal conductivity shows no correlation with porosity. Similar findings were made in Jones et al. (1984).

With the exception of the Hinton-Edson area, no 'hard' data exists for thermal conductivity in the study area. Beach et al. (1987) and Jones et al. (1984) provide average thermal conductivity values for different lithologies measured on several-hundred water-saturated core samples with a divided bar apparatus mainly taken from wells in the Hinton-Edson area. The data set provided in Beach et al. (1987) represents mean values for different rock types of Mesozoic, Cenzoic and Paleozoic

sediments and is more useful for large scale observations. Thermal conductivity given for dolomites is about 1 W m$^{-1}$ K$^{-1}$ lower than that presented in this work (note: not recalculated for water saturated conditions). In Jones et al. (1984), thermal conductivity is given for four geological formation groups. Thermal conductivity given for dolomitic limestones fits very well to the results for partially dolomitized limestones shown in Table 3. However, the core samples which were used for the thermal conductivity measurements in the Hinton-Edson area are mainly taken from very porous and permeable zones and,

in some cases, from very small depth intervals (Jones et al., 1984). Therefore, they do not represent all relevant parts of the reservoir. The other thermal conductivity values presented in Table 4 are included to demonstrate the variability of this parameter for different facies and/or reservoirs. According to Popov et al. (2016), thermal rock properties are critical parameters for thermo-hydrodynamic models and for predicting the lifetime performance of geothermal systems. Therefore an accurate determination of thermal properties of each relevant formation is necessary.

As a generally observed phenomenon, investigated porosity in the dolomitized reef sections is nearly independent from reservoir depth. Amthor et al. (1994) conducted a detailed study on porosity and permeability changes within the Leduc Formation. According to their findings, there is an overall decrease of porosity and permeability in the Leduc Formation with increasing burial depth in limestones, partially dolomitized limestones and also dolostones. At depths greater than 2000 m dolostones contain a significantly higher porosity and permeability than limestones, resulting in a much better reservoir

quality. Porosity and permeability primarily depend on depositional facies (mudstone/wackestone to floatstone/rudstone) and the degree of dolomitization, which results in generally higher horizontal permeability than vertical permeability in the investigated wells. In addition, two types of fractures are common in both formations: (1) irregular networks of hairline fractures, commonly filled with bitumen, connecting vugs and molds especially within the packstone, floatstone and rudstone facies; (2) mm-wide and cm-long subhorizontal to vertical fractures that are filled mainly with calcite and occur

randomly throughout the matrix. The origin of these microfractures is not certain, but they might not be restricted to the well-sealed pools in the deepest part of the basin as described in Marquez and Mountjoy (1996). Furthermore, while dolomitization and fracturing enhanced the reservoir properties, cementation by anhydrite, calcite, and bitumen plugged pores and pore throats to variable degrees, thereby reducing reservoir porosity and thus permeability significantly (see also Amthor et al 1993, 1994). These processes are not identifiable in log files, which is a well-known fact for the oil industry,

but it should be regarded and validated in further reservoir evaluation for geothermal purposes. It might be also important for the extrapolation of rock properties to reservoir scale during modelling. Additionally, most of the AccuMap data is only available for some (expected) high permeable zones and therefore statistically not representative for both formations in general. Therefore, detailed core analysis is essential.

However, a good correlation between porosity, permeability, density and thermal conductivity is given in core intervals where post-depositional cementation and bitumen precipitation was minor to absent, as shown in well 7-33-48-12W5 (Fig. 9). At some depth levels only broken core pieces are present in almost all investigated wells. This might be a hint for hydraulic active pathways, as it is the case in the German Molasse Basin where hydraulic conductivity is mainly bound to fracture networks. The investigated fracture networks might be important for stimulation and enhancement of the reservoir for hydrothermal utilisation.

There are notable differences between the subsurface reservoirs and the investigated outcrops, depending mainly on the extent of dolomitization and tectonic overprinting. The outcrop samples of the Fairholme Complex—chosen as the analogue for the RMRT reservoir —indicate geothermally relevant differences between the off-reef facies (argillaceous limestones) and the reef facies (dolomitized), while the carbonates of the SCCC outcrops are predominantly limestones without such a differention. Differences in rock properties between reef and off-reef facies are comparable to differences between the basinal (Lobstick Member) and reef facies (Zeta Lake Member) in the Nisku Formation. However, measured thermo- and petrophysical properties reveal much lower porosities and permeabilities, likely due to the low grade metamorphism in the Rocky Mountain Front Ranges. Also, the tectonic compression during the Laramide orogeny created sets of fractures and faults that are absent in the subsurface reservoirs, except perhaps in the immediate vicinity to the limit of disturbed belt (as shown in Fig.1). Hence, considering that the stratigraphically equivalent subsurface reservoirs were not tectonized and/or metamorphosed like the outcrop samples, the petrophysical properties measured on the outcrop samples from the Rocky Mountains are not useful as analogues to the subsurface reservoirs in the Alberta Basin. Therefore, creating a conceptual model for Alberta such as that for the Malm-Aquifer in Germany (Homuth et al., 2015) is not possible, at least not with the current data base.

The classifications by Sass and Götz (2012) and Bär et al. (2011) were applied on the rock properties for a first assessment. The application of the thermofacies concept (Sass and Götz, 2012) is useful in an early exploration stage or in cases where detailed reservoir information is not available but it does not replace further exploration. Likewise, the threshold values applied in Bär et al. (2011) were defined for a low enthalpy region in Germany and are used as indicators, which will likely need to be redefined for the Upper Devonian aquifer systems with increasing experience of geothermal exploitation in this specific reservoir.

Rock property measurements provide matrix properties only (mesoscale). Furthermore, they need to be corrected for reservoir conditions and transferred to reservoir scale. For a more reliable geothermal assessment, further reservoir parameters (macroscale) like reservoir temperature, flow rates, heat flow, potentiometric surfaces, TDS and $H_2S$ content etc. are needed. Therefore it is necessary to evaluate well data provided in the AccuMap or GeoScout database. These databases comprise an enormous amount of well data from various companies from the last seven decades. However, there is no information given about the quality of the data. In particular, temperature data were identified as inaccurate (Weides and

Majorowicz, 2014). Furthermore, heat flow values have been corrected for paleoclimatic surface temperature forcing (Majorowicz et al., 2012).

As the Leduc Formation is relatively homogenous within the Alberta Basin, reservoir temperature is one of the crucial parameters. An important point will be to identify 'hot spots' at an economical depth. Likewise, flow rates are very variable at a local scale (Lam and Jones, 1985) and need to be analyzed to evaluate the economic geothermal potential.

Furthermore, the aquifer systems need to be operated as transitional systems which need stimulation. This point needs to be considered during cost calculation. Most natural geothermal systems need stimulation for economic and technical reasons (Sass and Götz, 2012). According to the AccuMap database, reservoir stimulation was also a common process for increasing the productivity during oil and gas production in the reservoir. Previous studies (Hofmann et al., 2014) indicate that stimulation treatments can increase the economic feasibility for geothermal utilization in the WCSB for at least some reservoir formations.

A limiting factor for geothermal utilization in the study area is the complex hydrogeology. Both formations (Leduc and Nisku) contain highly concentrated water with average TDS values of approximately 200 g l$^{-1}$ (Rostron et al., 1997; Michael et al., 2003). Within the SCCC, salinity increases with increasing distance from the Rocky Mountains ('squeegee flow', Buschkuehle and Machel, 2002; Machel and Buschkuehle, 2008). This parameter can be also very variable at a local scale. In the Hinton-Edson area, salinity ranges from less than 50 g l$^{-1}$ up to 180 g l$^{-1}$ (Lam and Jones, 1985). Likewise, the Nisku Reef Trend is well known for its (in some areas over pressured) sour gas pools (Bachu et al., 2008). Problems like scaling and corrosion during operation can lead to higher production costs or in the worst case scenario to the abandonment of the well. These problems are not solved in the geothermal industry yet but should be addressed in the current discussion about a geothermal pilot project in the WCSB.

For this reasons a close cooperation with the oil industry is important. Using the experience in operating this reservoir, the existing equipment and infrastructure likely reduces exploration risks and costs for a geothermal project which could provide new business strategies.

For the next phase of the project, further parameters (e.g. heat capacity, thermal diffusivity and ultrasonic wave velocity) will be measured on well core samples to complement the data set provided in this study. It is also considerable to extent the study area or include further formations to cover areas with higher heat flow or temperatures. Well data provided in the AccuMap or GeoScout databases will be evaluated, interpreted and probably mapped to identify the most promising areas for geothermal utilization in the reservoir. Due to the high amount of well data, this will be only possible on a regional scale. The construction of a regional geological 3D model, including already existing data from previous studies (Majorowicz et al., 2012; Nieuwenhuis et al., 2015), is planned for the most promising areas.

Considering geothermal utilization, dolomitization enhanced all analyzed rock properties. Although the thermo- and petrophysical rock properties seem to be relatively homogeneous with respect to the whole reservoir, they are strongly correlated to depositional facies, which were modified by several diagenetic events (dolomitization, fracturing, emplacement

of secondary anhydrite, calcite cementation and bitumen precipitation). These differences in fabric, mineral content, and texture affect the geothermal properties and lead to facies dependent trends and must be considered for identification of preferred zones.

**Acknowledgements**

This study has received funding from the DAAD – German Academic Exchange Service in the framework of the 'MalVonian' project. We would like to thank Jean Borgomano, Jacek Majorowicz,and three anonymous reviewers for their constructive comments.

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

**Table 1: Source location of examined cores and outcrops**

| Nr. | Well-ID (Abbreviation) | Location | Formation/Depth intervals (TVD) |
|---|---|---|---|
| 1 | 7-33-48-12W5 (7-33) | Nisku Reef Trend, Brazeau field, reef | Nisku (D2), 3058.38 m – 3154.00 m |
| 2 | 2-19-48-12W5 (2-19) | Nisku Reef Trend, Brazeau field, bank-edge reef | Nisku (D2), 3150.00 m – 3213.00 m |
| 3 | 11-32-47-12W5 (11-32) | Nisku Reef Trend, Brazeau field, bank | Nisku (D2), 3154.00 m – 3195.00 m |
| 4 | 5-22-48-27W4 (5-22) | RMRT, Wizard Lake field, central basin | Leduc (D3), 1827.00 m – 2013.50 m |
| 5 | 10-31-37-9W5 (10-31) | RMRT, Strachan field, western basin | Leduc (D3), 4296.5 m – 4328.38 m |
| 6 | 16-18-61-15W5 (16-18) | SCCC, Windfall, central basin | Leduc (D3), 2741.00 m – 2779.77 m |
| 7 | 2-36-54-23W5 (2-36) | SCCC, Obed, western basin | Leduc (D3), 4068.77 m – 4095.00 m and 4145.00 m – 4165.39 m |
| **Nr.** | **Outcrop** | **Location** | **Analyzed Formations** |
| 1 | Toma Creek | SCCC | Cairn (D3) and Grotto (D2) |
| 2 | Jasper Railroad | SCCC | Perdrix, Cairn and Peechee (D3) |
| 3 | Mt. Greenock | SCCC | Peechee and Perdrix (D3) |
| 4 | Nigel Peak | Fairholme Complex | Perdrix (D3), Mt. Hawk and Grotto (D2) |
| 5 | Grassi Lakes | Fairholme Complex | Peechee and Cairn (D3) |
| 6 | Gap Lake | Fairholme Complex | Peechee and Cairn/Perdrix (D3) |

**Table 2: Rock properties of outcrop analogue samples**

| Parameters | ρ [g cm⁻³] | N | φ [%] | N | K_i [m²] | N | λ [W m⁻¹ K⁻¹] | N | α [10⁻⁶ m² s⁻¹] | N | c_p [J kg⁻¹ K⁻¹] | N |
|---|---|---|---|---|---|---|---|---|---|---|---|---|
| **Fairholme Complex** | | | | | | | | | | | | |
| Mt Hawk (D2) | 2.71 ±0.006 | 10 | 2.95 - | 1 | 2.44E-18 ±4.5E-18 | 7 | 2,71 ±0.23 | 8 | 1.66 ±0.25 | 8 | 782.44 ±1.40 | 2 |
| Grotto (D2) | 2.85 ±0.006 | 14 | 4.21 ±0.93 | 5 | 3.95E-18 ±4.6E-18 | 8 | 3,14 ±0.23 | 7 | 1.8 ±0.25 | 7 | 849.34 ±2.33 | 2 |
| Perdrix (D3) | 2.69 ±0.026 | 17 | 3.07 ±1.24 | 5 | 2.51E-18 ±5.2E-18 | 11 | 2.58 ±0.30 | 12 | 1.51 ±0.34 | 12 | 813.11 ±0.25 | 2 |
| Peechee (D3) | 2.83 ±0.040 | 26 | 6.6 ±2.34 | 19 | 1.50E-17 ±3.7E-17 | 13 | 3.31 ±0.56 | 6 | 1.98 ±0.35 | 7 | 832.96 ±15.82 | 8 |
| Cairn (D3) | 2.84 ±0.011 | 13 | 4.32 ±1.29 | 5 | 8.46E-18 ±2.1E-17 | 8 | 3,12 ±0.26 | 8 | 1.73 ±0.32 | 4 | 847.00 ±7.29 | 2 |
| **Southesk Cairn Carbonate Complex** | | | | | | | | | | | | |
| Grotto (D2) | 2.77 ±0.025 | 6 | - | - | 9.40E-16 ±1.1E-15 | 5 | 2.4 ±0.13 | 6 | 1.86 ±0.23 | 7 | 830.00 ±26.87 | 2 |
| Peechee (D3) | 2.73 ±0.039 | 16 | 3.60 ±1.50 | 9 | 6.70E-16 ±2.1E-15 | 16 | 2.46 ±0.09 | 7 | 1.36 ±0.03 | 7 | 816.30 ±14.60 | 6 |
| Cairn (D3) | 2.74 ±0.110 | 39 | 2.90 ±0.27 | 15 | 1.06E-14 ±3.3E-14 | 33 | 2.93 ±0.42 | 23 | 1.72 ±0.44 | 16 | 796.30 ±46.67 | 6 |

Classification of potential after Bär et al. (2011)

| Parameters | very low | low | medium | high | very high |
|---|---|---|---|---|---|
| Permeability [m²] | < 5·10⁻¹⁵ | > 5·10⁻¹⁵ | > 1·10⁻¹³ | > 5·10⁻¹³ | > 4·10⁻¹² |
| Thermal conductivity [W m⁻¹ K⁻¹] | < 1,25 | > 1,25 | > 2,0 | > 3,0 | > 5,0 |
| Thermal diffusivity [10⁻⁶·m² s⁻¹] | < 0,6 | > 0,6 | > 1,0 | > 1,5 | > 2,0 |

**Table 3: Average thermal conductivity values of the anlyzed reservoir core samples**

| Formation | Aquifer | λ [W m$^{-1}$ K$^{-1}$]* | N |
|---|---|---|---|
| Leduc (RMRT) | | 4.25 ± 0.59 | 128 |
| Leduc (SCCC) | D3 | 4.23 ± 0.43 | 82 |
| **Leduc (total)** | | **4.24 ± 0.53** | **210** |
| Dismal Creek | | 3.13 ± 0.44 | 15 |
| Lobstick | | 2.24 ± 0.38 | 23 |
| Bigoray | D2 | 2.81 ± 0.43 | 27 |
| Zeta Lake | | 4.15 ± 0.96 | 41 |
| **Nisku (total)** | | **3.25 ± 1.02** | **106** |
| **Lithology** | **Wells** | λ [W m$^{-1}$ K$^{-1}$] | N |
| Limestone | 7-33, 2-19, 10-31 | 2.32 ± 0.40 | 27 |
| Partially dolomitized | 7-33, 2-19 | 2.81 ± 0.43 | 27 |
| Dolostone | all | 4.19 ± 0.65 | 262 |

*arithmetic mean values ± standard deviation, N = number of analyzed samples

**Table 4: Compilation of average thermal conductivity values**

| Source | $\lambda$ [W m$^{-1}$ K$^{-1}$] | | | | Measurement device | Saturated/ Dry |
|---|---|---|---|---|---|---|
| | Limestone | Dolomitic limestone | Dolostone | Shales | | |
| This work | 2.32 ± 0.40 (27) | 2.81 ± 0.43 (27) | 4.19 ± 0.65 (262) | - | TCS | dry |
| Beach et al. (1987) Mesozoic to Palezcoic sediments | 2.42 ±0.88 (679) | - | 3.1 ± 1.4 (254) | 1.38 ± 0.42 (72) | Divided bar apparatus | sat |
| Jones et al. (1984) Upper Devonian to top of Precambrian | - | 2.98 ± 1.19 | - | - | Divided bar apparatus | sat |
| Homuth et al. (2015) Jurassic Malm Formation, Germany | 1.72 - 4.87 (418) grain dominated/ reefal facies | - | - | - | TCS | dry |
| | 1.35 - 2.62 (478) matrix dominated/ basin facies | - | - | - | TCS | dry |
| Götz et al. (2014) Trias, Hungary | 2.4 - 2.5 | - | 2.0 - 3.4 | - | TCS | dry |
| Götz et al. (2014) Miocene, Hungary | 1.1 - 1.2 | - | - | - | TCS | dry |

TCS = Thermal Conductivity Scanner, arithmetic mean values ± standard deviation, () = number of analyzed samples

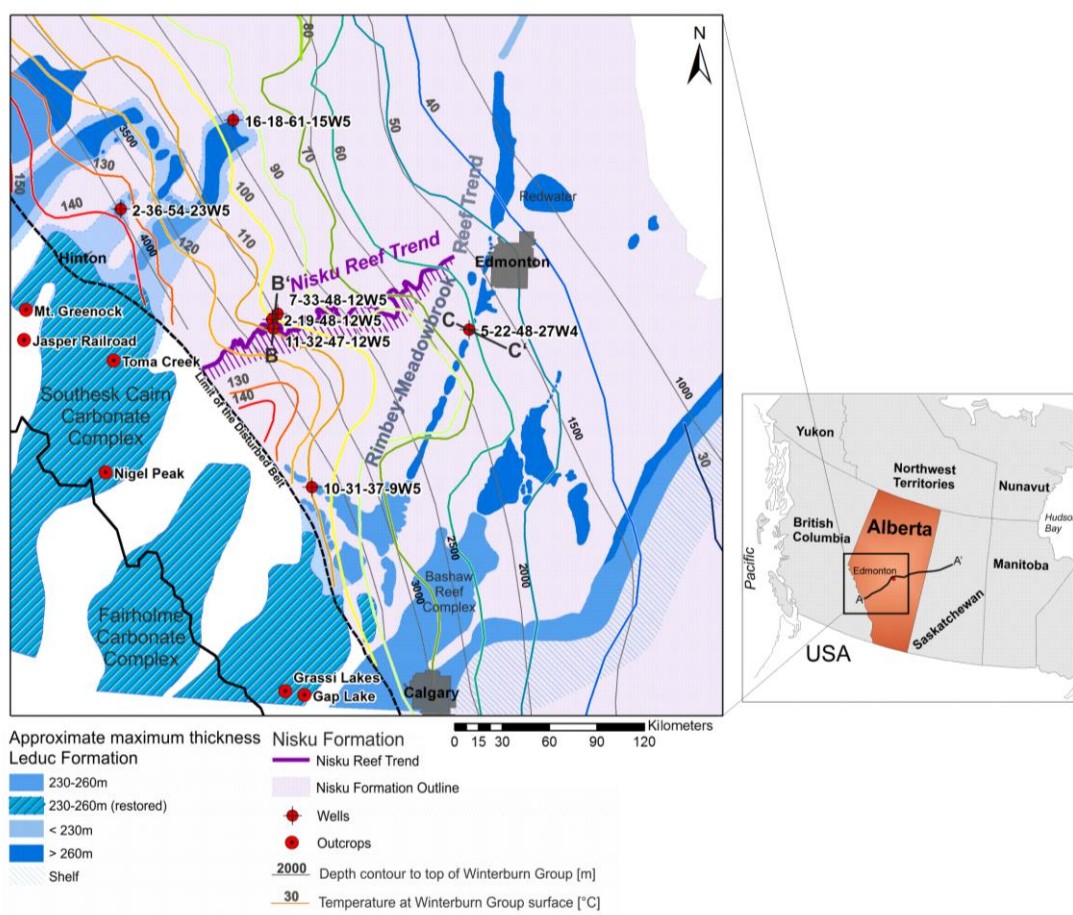

**Figure 1: Spatial distribution and approximate thickness distribution of Upper Devonian carbonate platforms in Alberta and neighboring British Columbia. The outlines of the carbonate platforms are based on seismic (map modified from Switzer et al., 1994). The wavy colored contours indicate temperatures on top of the Winterburn Group, as calculated from data derived from more than 26400 wells (Majorowicz and Weides, 2014). The flat grey lines are depth contours marking the top of the Winterburn Group. See text for further explanation.**

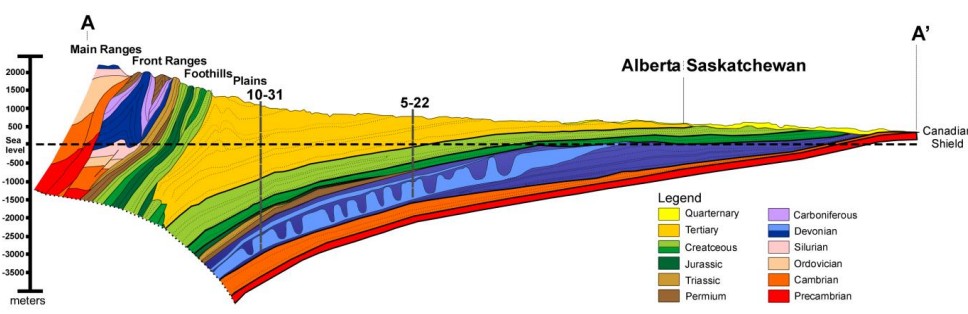

**Figure 2: Schematic W-E cross section of the Alberta Basin (modified and simplified from Price, 1994). Line A-A' is shown in the inset in Fig. 1.**

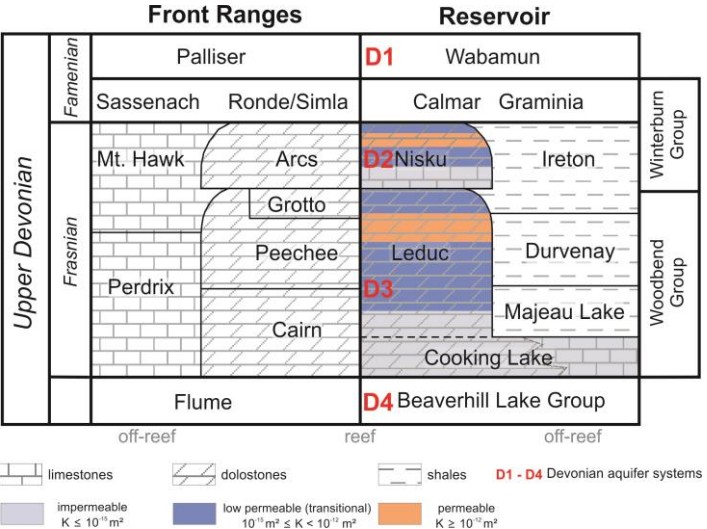

**Figure 3: Simplified lithostratigraphy for the Alberta Basin and the adjacent Rocky Mountains, and schematic classification of permeability zones (based on Switzer et al., 1994).**

## Nisku Reef Trend

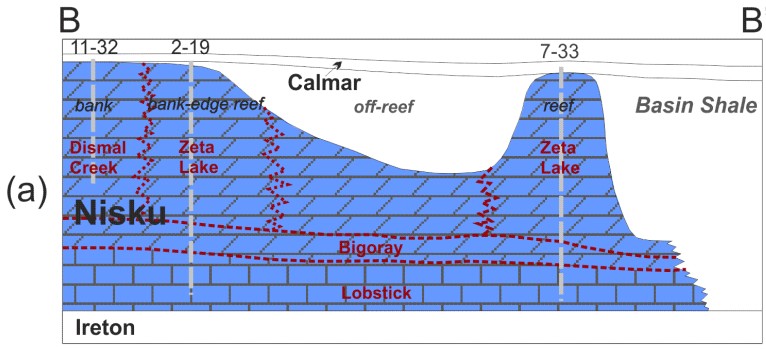

## Rimbey-Meadowbrook Reef Trend

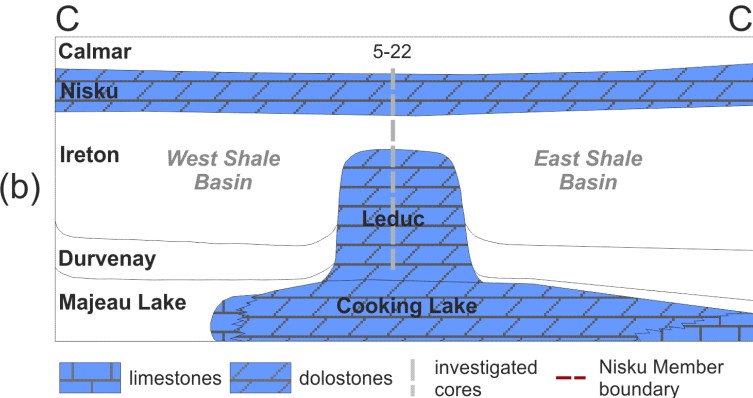

**Figure 4: Schematic cross sections of the Nisku Reef Trend (a) and the Rimbey-Meadowbrook Reef Trend (b) in central Alberta, based on Switzer et al. (1994). Line B – B' and line C – C' are shown in Fig. 1.**

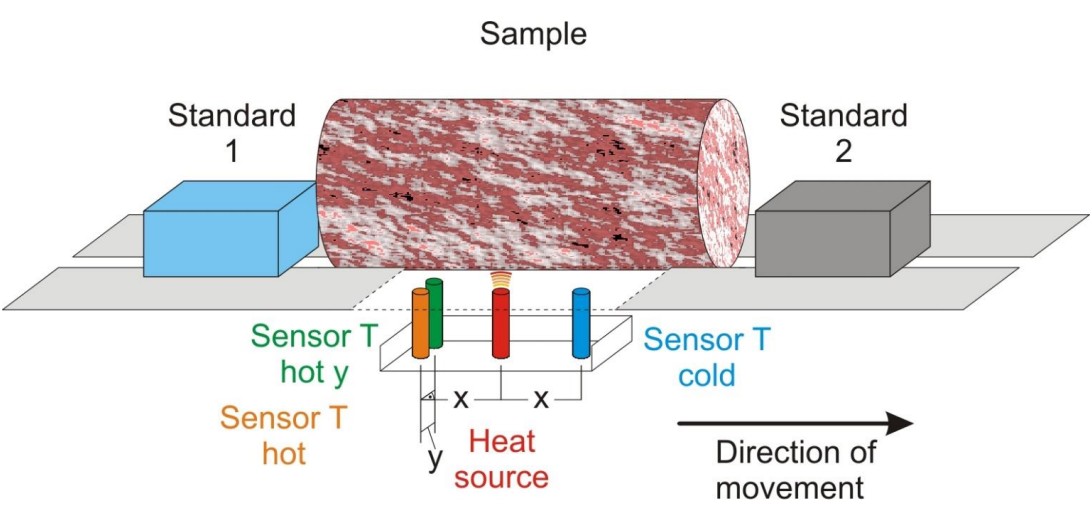

**Figure 5: Scheme of thermal conductivity measurements of the analyzed core samples according to Popov et al. (1999) retrieved from Sass and Götz (2012).**

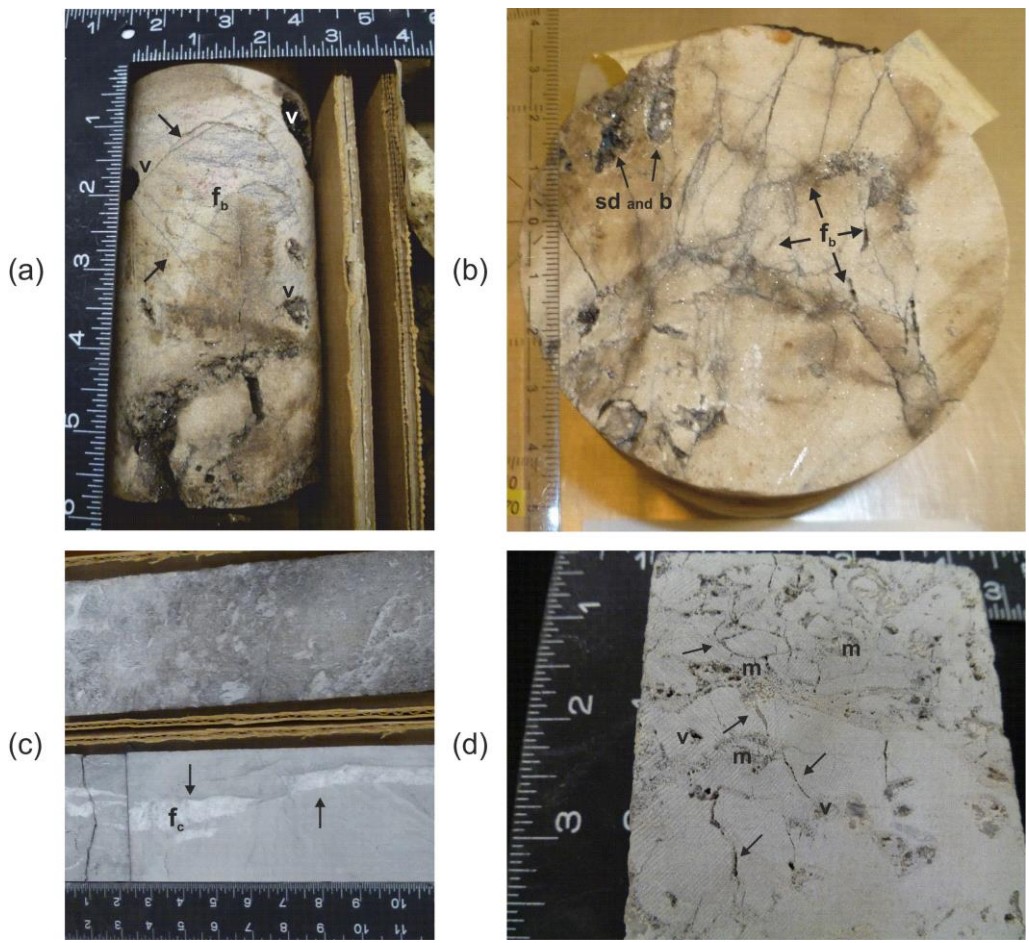

**Figure 6:** (a) and (b) Hairline fractures filled with bitumen $(f_b)$ extending subvertically to radially from vugs $(v)$ and molds $(m)$ in a stromatoporoid floatstone at 1863.24 m depth (Leduc Formation, well 5-22, RMRT). **'sd and b' stands for saddle dolomite covered with bitumen.** Calcite filled fractures $(f_c)$ in a dolomudstone (c) at 3062.9 m depth representing the upper part of the Zeta Lake Member in well 7-33 (Nisku). Irregular fractures (black arrows as indicators) connecting vugs and molds (d) representing the middle part of the Zeta Lake Member at 3086.4 m depth in well 7-33.

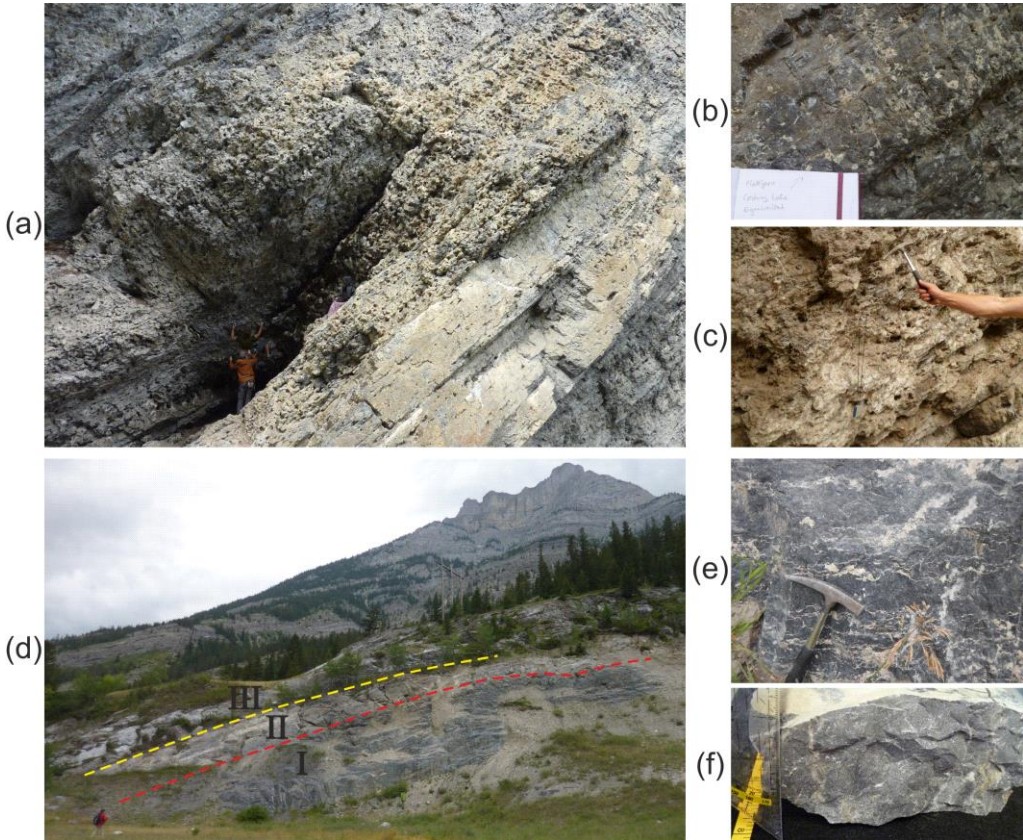

**Figure 7: Dolomitized reefal beds of the Peechee Member (a) with countless molds of bulbous stromatoporoids (c) at Grassi Lakes. Highly fractured, nonporous, massive dolostones of the Lower Cairn platform (b), likely equivalent to the Cooking Lake platform representing the base at Grassi Lakes. Overview of sample location Gap Lake showing three stratigraphic units (d), whereby III stands for Mississippian Exshaw shale, II for the Peechee Member and I for either the Cairn or Perdrix Formation. Dark grey, massive and coarsely bedded partially dolomitized carbonates of the Cairn/Perdrix Formation (e) and of the mouse grey Peechee Member (f) at Gap Lake.**

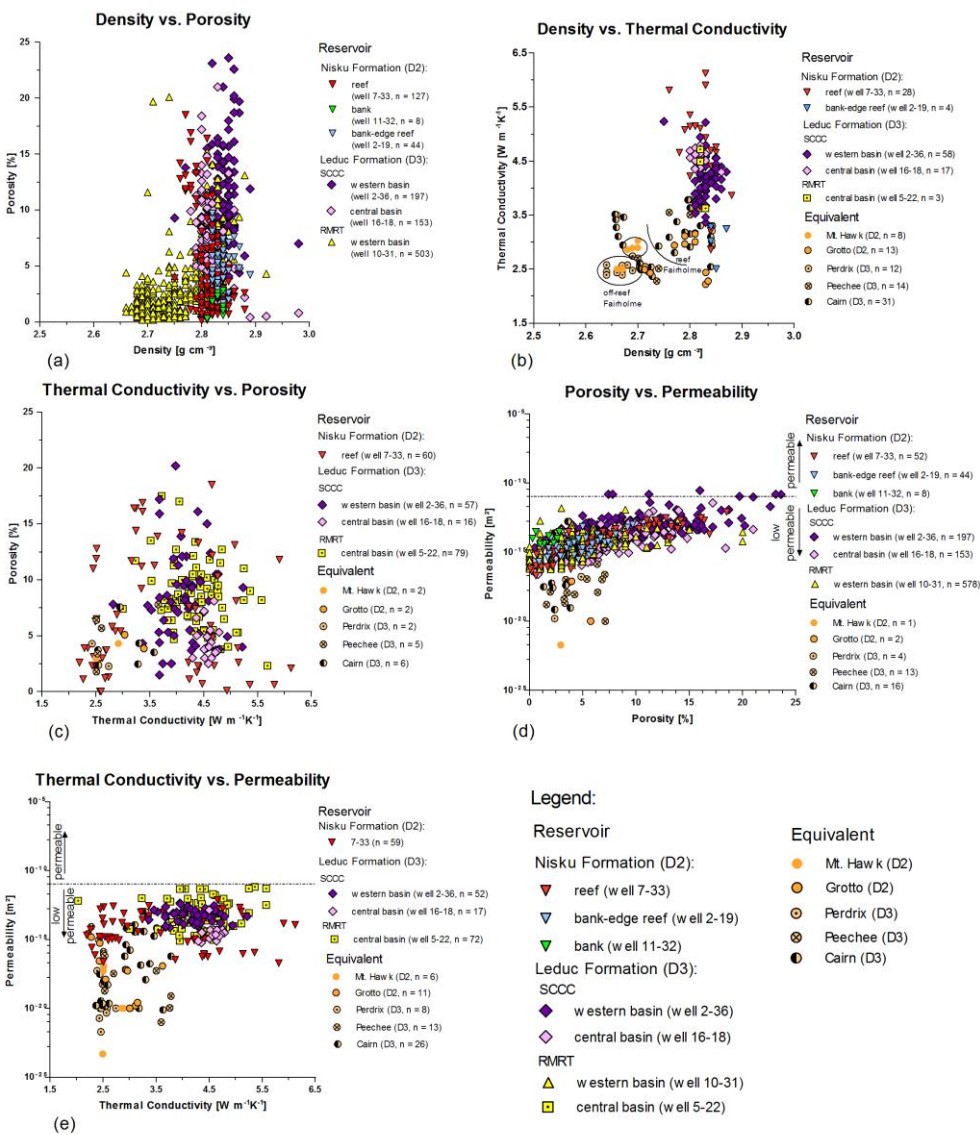

**Figure 8: Correlation of petro- and thermophysical rock parameters analyzed at dry conditions.**

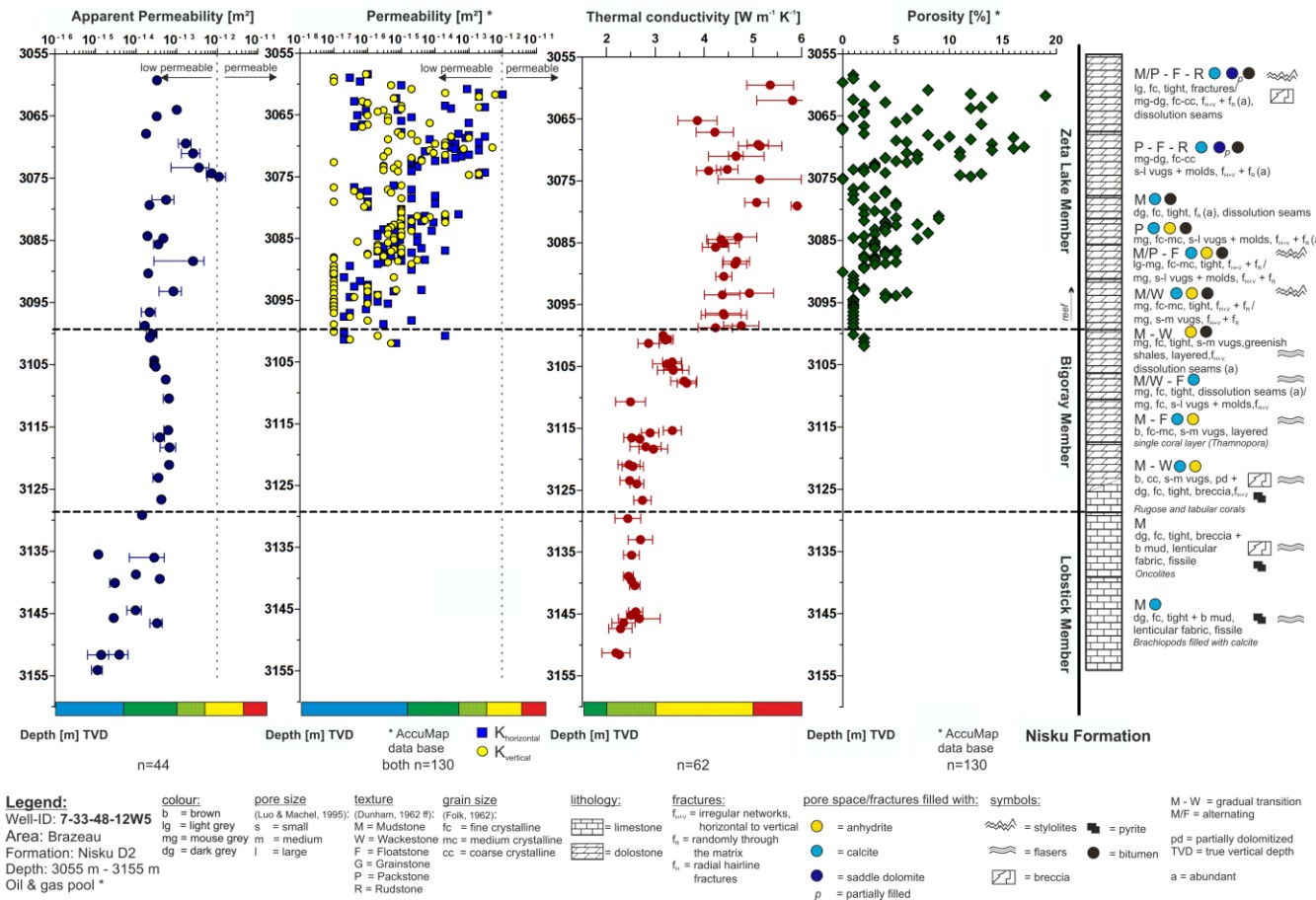

**Figure 9: Rock properties of well 7-33-48-12W5 (Nisku Formation). Apparent permeability and thermal conductivity are displayed as arithmetic mean values with standard deviation. Permeability and porosity values were retrieved from the AccuMap data base (marked with *, here and hereafter). N represents the number of analyzed core samples.**

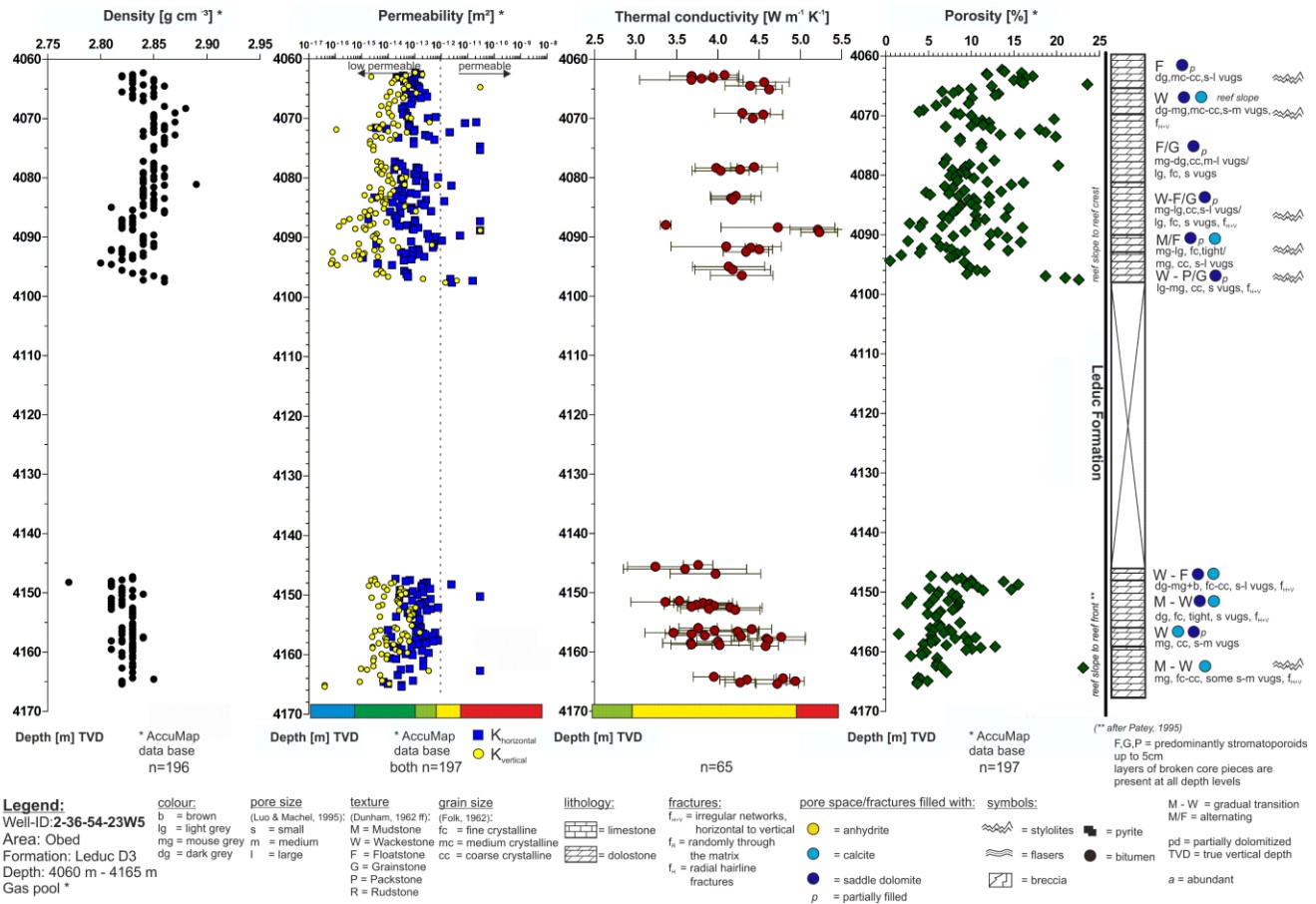

**Figure 10: Rock properties of well 2-36-54-23W5 (Leduc Formation, SCCC). Thermal conductivity is displayed as arithmetic mean values with standard deviation. Density, permeability and porosity were retrieved from the AccuMap data base. Apparent permeability is not displayed here, but lies in the range of the permeability values taken from the AccuMap database. N represents the number of analysed core samples.**

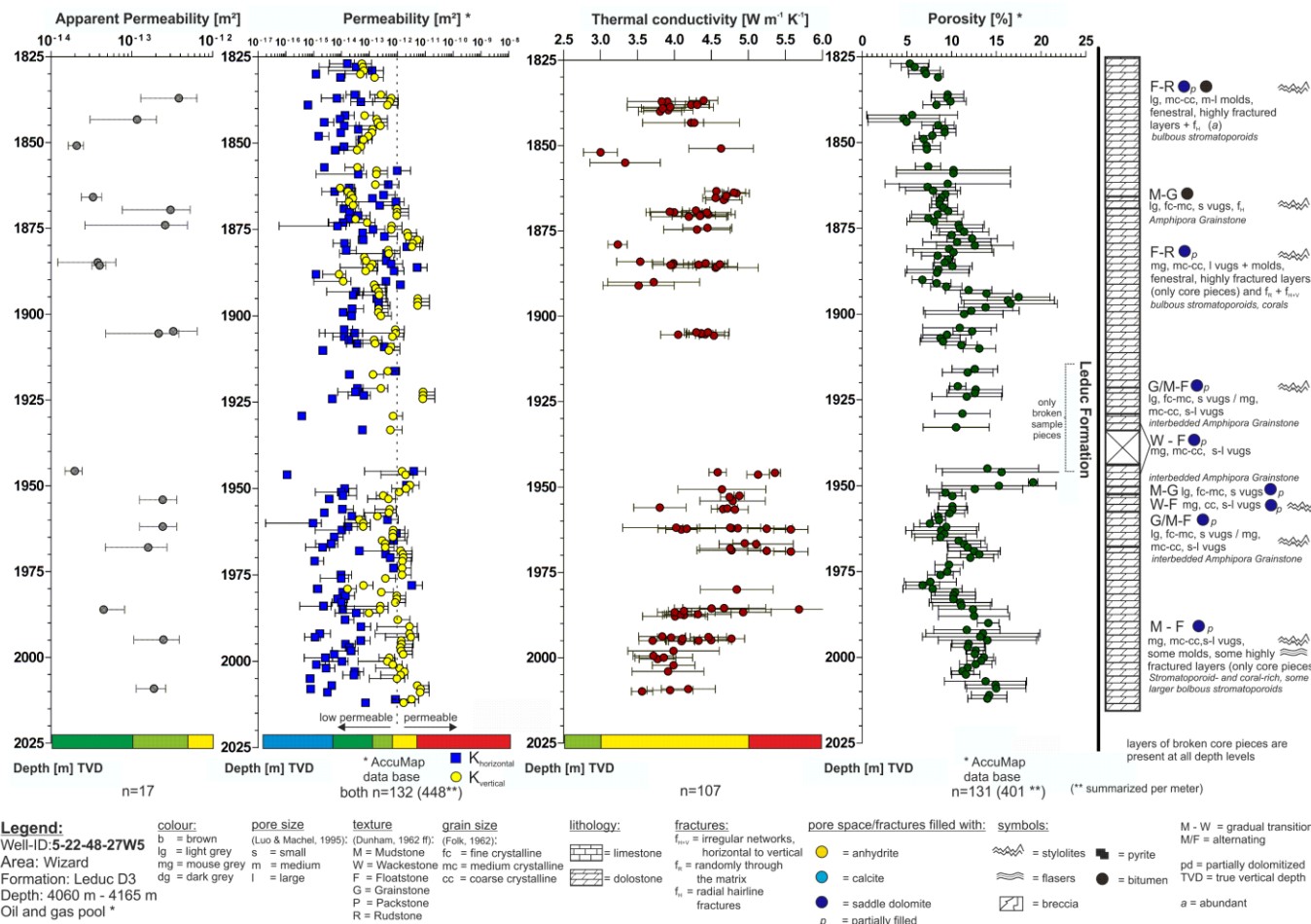

**Figure 11: Rock properties of well 5-22-48-27W5 (Leduc Formation, RMRT).** Apparent permeability and thermal conductivity are displayed as arithmetic mean values with standard deviation for every analyzed core sample. Permeability and porosity were retrieved from the AccuMap data base. Due to the high amount of measurements and for clarity reasons, the values were summarized per meter and are also displayed as mean values with standard deviation.

**Appendix A**

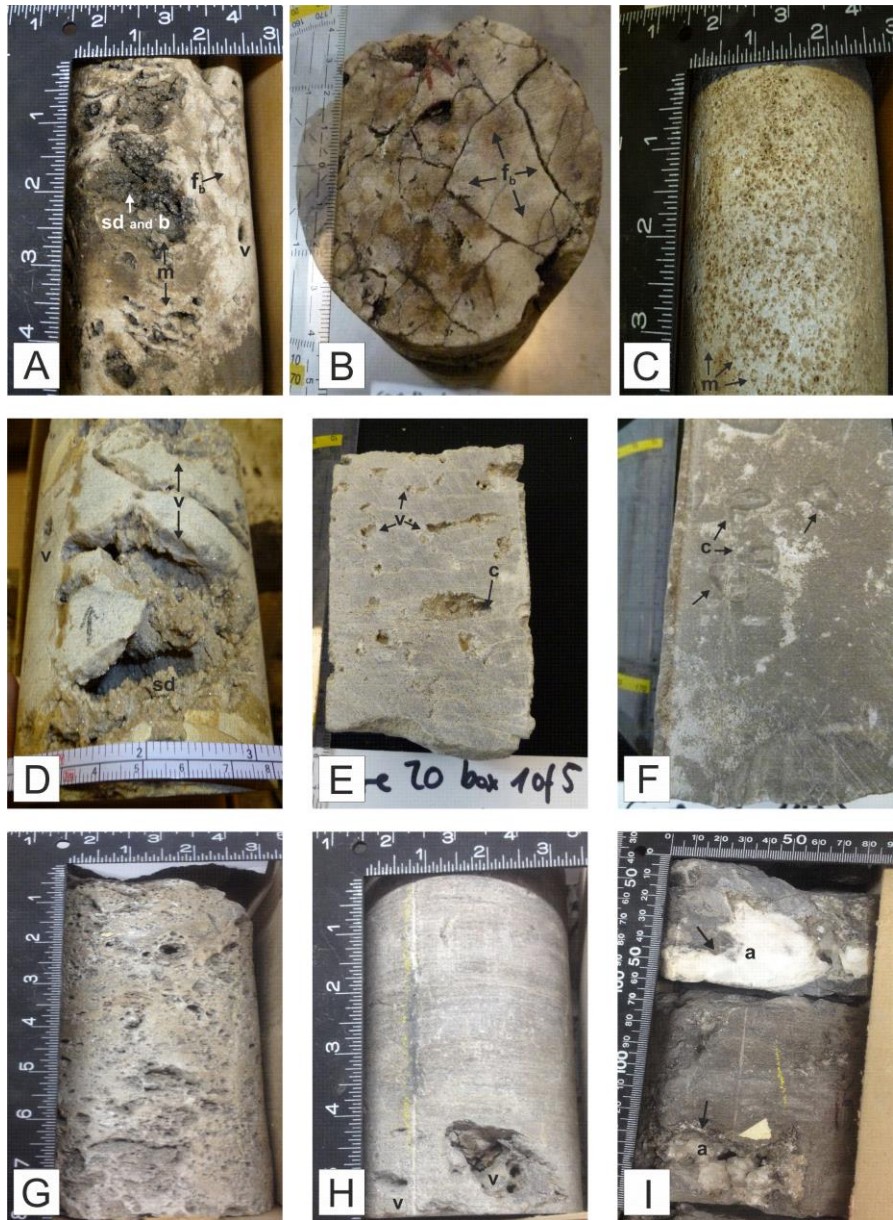

**Plate 1: Representative core samples of the Leduc Formation from the RMRT and the SCCC. Except for (F), all samples are dolostones. The scale bar is in inch and cm. (A) and (B) Light grey Stromatoporoid-floatstone (medium crystalline) with large vugs and abundant subvertical fractures (f$_b$) filled with bitumen (1843.1 m bgl, 5-22-48-27W4, RMRT). 'sd and b' stands for saddle dolomite covered with bitumen. (C) Light grey Amphipora grainstone with small sized tubular molds (1957 m bgl, 5-22-48-27W4). 'm' = molds. (D) Coarse crystalline Stromatoporoid-floatstone/rudstone with large vugs that are lined with saddle dolomite (sd = saddle dolomite, 1993.4 m bgl, 5-22-48-27W4). (E) Core slab of skeletal wackestone/floatstone with calcite cement (c) crystals lining scattered molds and vugs (4290 m bgl, 10-31-37-9W5, RMRT). (F) Core slab of nonporous, nondolomitized mudstone facies in the Strachan pool close to the Rocky Mountains (4306.8 m bgl, 10-31-37-9W5). Brachiopod molds (black arrows) are filled with microsparitic calcite cement (c). (G) Coarse crystalline packstone/floatstone facies with a variety of molds of corals and stromatoporoids (4095 m bgl, 2-36-54-23W5M, SCCC). (H) Fine crystalline dolomudstone with scattered large vugs (4159.75 m bgl, 2-36-54-23W5M). (I) Irregularly shaped masses of replacive anhydrite (a) in medium to dark grey replacement dolomite. Anhydrite near the bottom of this photo is partially infiltrated by 'dead oil' (4164.78 m bgl, 2-36-54-23W5M).**

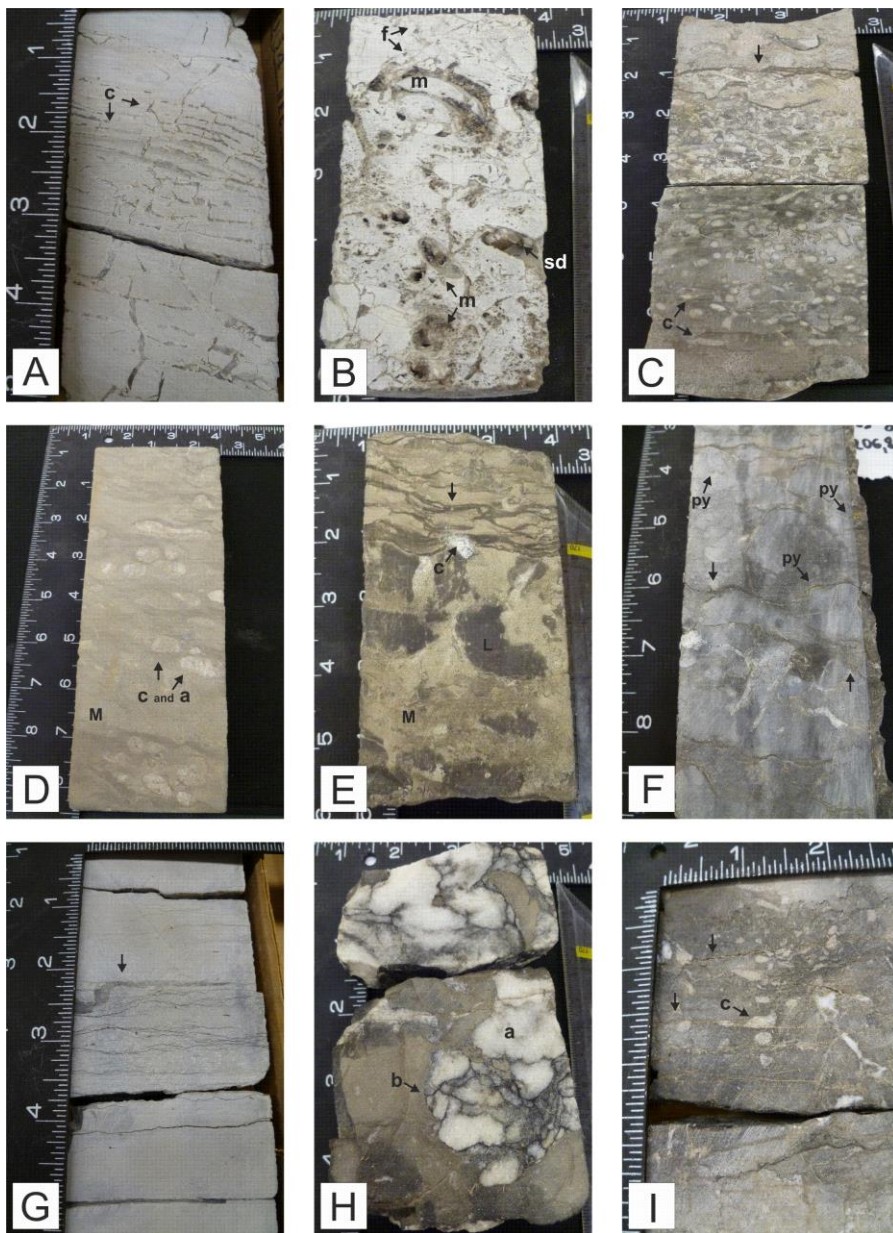

**Plate 2: Representative core samples from the Nisku Formation. (A) Dolomitized fenestral laminite, which is a dolomudstone with abundant subhorizontal voids that are cemented with sparry crystals (c). This sample represents the Zeta Lake Member (3064.7 m bgl, reef, 7-33-48-12W5, Nisku). (B) Light grey, fine crystalline bioclastic floatstone/rudstone, with molds (m) of a variety of corals, brachiopods and stromatoporoids, connected by a network of hairline fractures (f). Some saddle dolomite (sd) lines some of the larger pores (3071.7 m bgl, Zeta Lake Member, 7-33-48-12W5). (C) Thamnopora-rich medium to coarse crystalline floatstone (3112.1 m bgl, Zeta Lake Member, 7-33-48-12W5). The vugs are filled with limpid calcite (c). Dissolution seams (black arrow pointing downwards) and fine joints are spread over the matrix. (D) Coarse crystalline dolomitized mud (M) with scattered corals filled with anhydrite and calcite (c and a, 3122.26 m°bgl, Bigoray Member, 7-33-48-12W5). (E) Dark grey fine crystalline limestone breccia (L) interbedded in coarse crystalline dolomitized mud (M) representing the upper portion of the Bigoray Member (3118.7 m bgl, 7-33-48-12W5). Abundance of dissolution seams (black arrow pointing downwards) and few vugs filled with sparitic calcite (c) in the top. (F) Dark grey to black, fine crystalline mudstone with a lenticular fabric. Small pyrite nodules (py, golden spots) occur throughout the matrix. Irregular horizontal fractures (black arrows) extend from the center to the edges. (F) represents the Lobstick Member (3187.5 m bgl, bank-edge-reef, 2-19-48-12W5). (G) Nonporous, fine crystalline dolomudstone with abundant dissolution seams (black arrow) in the center representing the Dismal Creek Member (3172.25 m bgl, bank, 11-32-47-12W5). (H) Argillaceous dolomudstone breccia with replacive anhydrite nodules (3184.25 m bgl, Dismal Creek Member, 11-32-47-12W5). The coarse anhydrite (a) is rimmed with bitumen (b). (I) Dark grey, fine crystalline dolomudstone with fine horizontal fractures (black arrows pointing downwards) and some limpid calcite (3193.4 m bgl, bank, Dismal Creek, 11-32-47-12W5).**

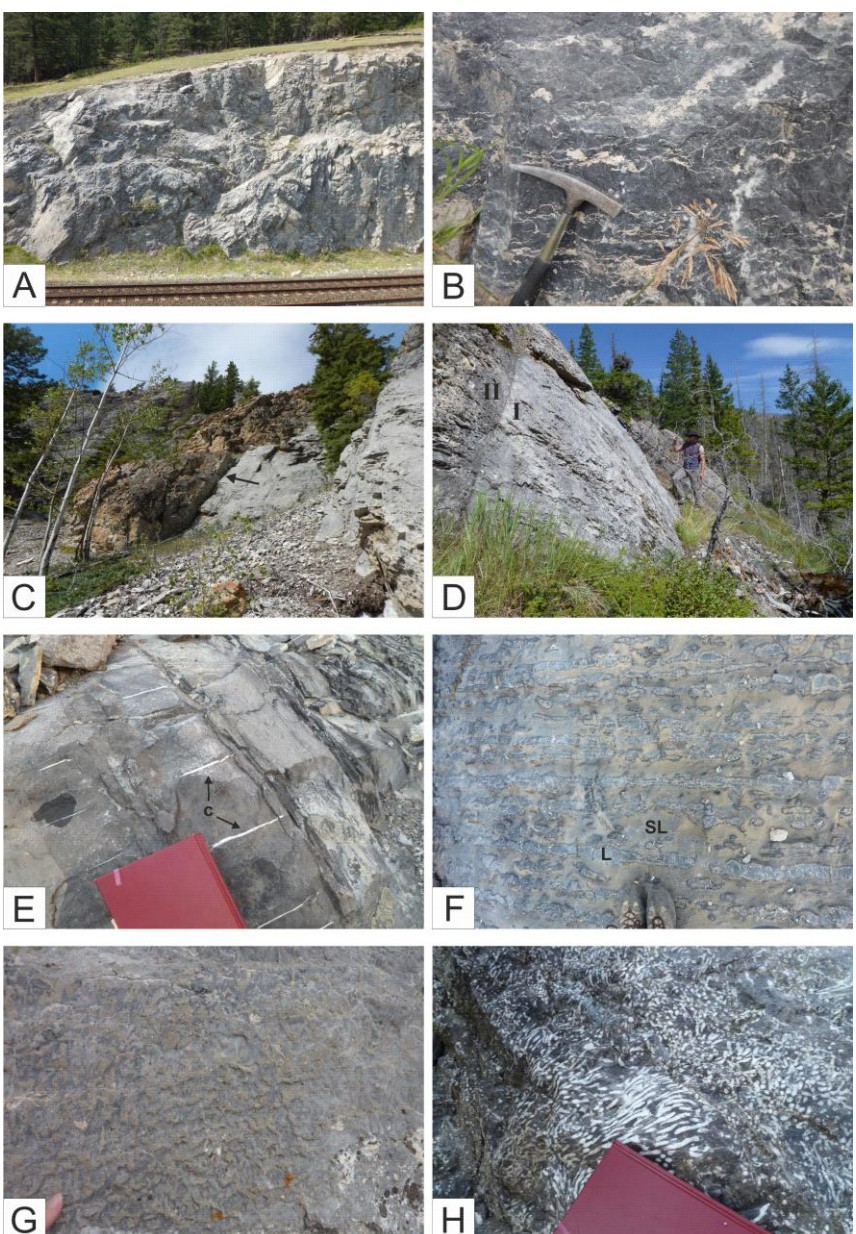

**Plate 3: Field photographs from the analogue outcrops representing the five analyzed formations. (A) Massive Upper Devonian bedrock comprising the Cairn and Peechee Formation directly next to the CN railroad tracks close to Jasper. The mouse grey carbonates (mainly limestones or partially dolomitized limestones) are in close proximity to thrust faults and are pervasively foliated, jointed and fractured, with the faults dipping mostly southwestward. This outcrop represents deposition near the margin of the SCCC. (B) Dark grey, massive and coarsely bedded partially dolomitized carbonates of the Cairn Formation at outcrop Gap Lake. This outcrop is also part of a thrust sheet and is overlain by Mississippian Exshaw shale. (C) Dolomitized parts (black arrow) along a major fault at sample location Mount Greenock, recognized by a medium brown coloration. (D) Steeply inclined, massive medium grey carbonates of the Peechee Formation (I, on the right), laterally juxtaposed against dark grey off-reef carbonates and marls of the Perdrix Formation (II, on the left) at outcrop Mount Greenock northeast of Jasper. The exposed carbonates are predominantly limestones and represent deposition near the margin of the SCCC. (E) The Perdrix Formation at outcrop Nigel Peak contains interbedded dark grey, massive limestones and dark grey shales. The bed thickness ranges from 20 to 50 cm. Note the about 15 cm long calcite filled vugs (c) perpendicular to the bedding. At outcrop Nigel Peak the Devonian carbonates crop out in the river bed (rocks dipping northeastward) and belong to the Fairholme Complex. (F) The dark grey limestones with brown silty layers in the Mount Hawk Formation at outcrop Nigel Peak contain abundant diagenetic bedding. Both, the Perdrix and the Mount Hawk represent the "off-reef". (G) Dark grey, massive limestones of the Perdrix Formation at outcrop Gap Lake showing an irregular, weathered surface. (H) The Grotto Formation at outcrop Nigel Peak consist of fine-grained, dark grey dolostones with abundant Disphyllid corals preserved in growth position.**

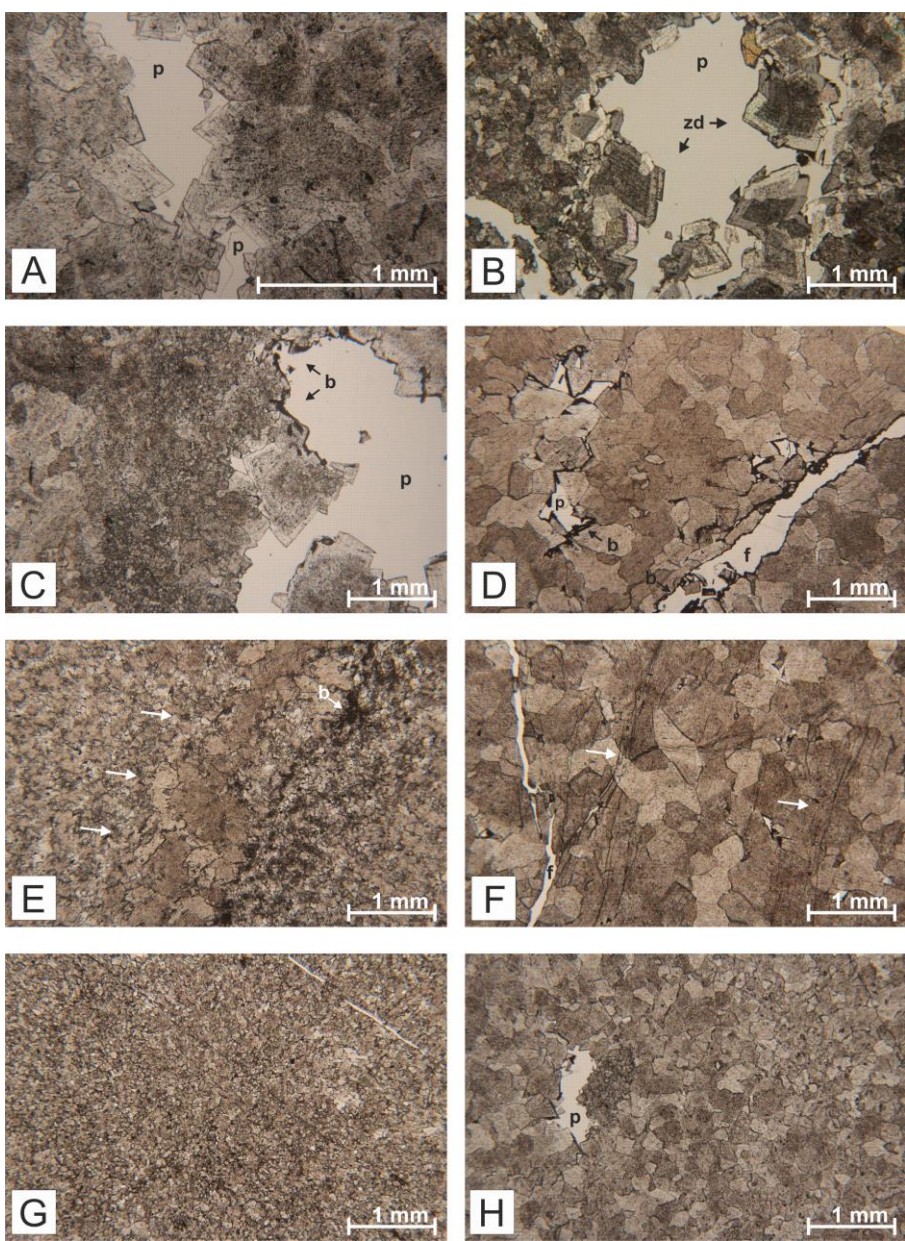

**Plate 4: Unstained thin sections from the Leduc Formation, RMRT (A-F) and the SCCC (G-H). All images taken in Plane Polarized Light (PPL) except for (B), which was taken in Crossed Polarized Light (XPL). (A – C) Planar-s to non-planar replacement dolomite with grain sizes up to 700 μm at different depth levels (A = 1827.1 m, B = 1895 m and C = 1984.4 m bgl, Wizard Lake field, well 5-2248-27W4, central basin). Concentrically zoned dolomite cements (zd, up to 400 μm) extend into the pore spaces (p) and are lined with bitumen (b, in C). (D-F) Replacement dolomite from the Strachan field (D = 4285.5 m, E = 4308.3 m and F = 4325.1 m bgl, well 10-31-37-9W5, western basin). (D) Mosaic of planar-s dolomite (200 - 400 μm) with fractures (f) and intercrystal pores, rimmed with bitumen (b). (E) Fine crystalline dolomite (10 – 50 μm) on the left crossed by a healed fracture in the center-right (white arrows) filled with coarser, nonplanar dolomite (300 μm) and disseminated bitumen (b) in intercrystal pores. F Coarse planar-s dolomite (up to 700 μm) with fractures (f) and possible pressure/deformation twinning (white arrows). Fine (G) and medium (H) grained, predominantly non-porous replacement dolomite from the Obed field in the SCCC (well 2-36-54-23W5, western basin, G = 3939.84 m and H = 4078.28 m bgl).**

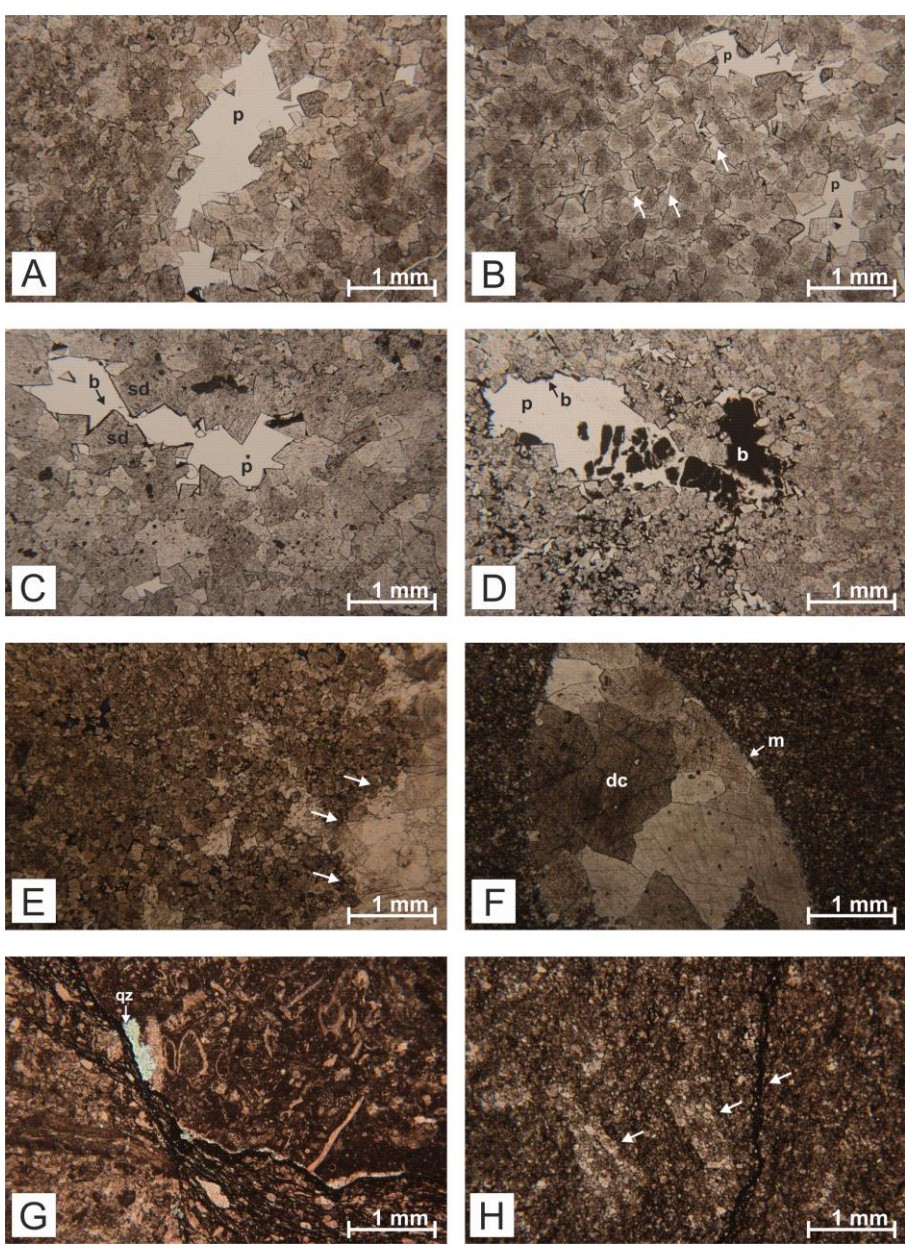

**Plate 5:** Unstained thin sections from the Leduc Formation (A-D, SCCC) and the Nisku Formation (E-F). (A) Planar-s matrix dolomite (50 – 150 μm) with intercrystal pores (white arrows) and minor amounts of dolomite cement (A = 4147.87 m bgl). (B) Porous matrix dolomite (200 – 400 μm) lined with bitumen (b) grown into the pore space (p; B = 4160.52 m bgl). Both (A) and (B) are from well 2-36-54-23W5 (SCCC, Obed field). (C) Coarse planar-s matrix dolomite and some saddle dolomite (sd) lining the pore space, with a thin coat of bitumen (b; 2761.48 m bgl). (D) Brown porous matrix dolomite with bitumen (b) scattered in intercrystal pore spaces and partially lining the large void. (C) and (D) are both from well 16-18-61-15W5, 2777 m bgl (Windfall field, SCCC). (E) Fine crystalline dolomite and coarse planar-s dolomite (white arrows) in former pore space on the right (3150.80 m bgl, Zeta Lake Member, 2-19-48-12W5, bank-edge-reef). (F) Coarse crystalline dolomite cement (dc, 800 μm) in secondary void (coral mold) surrounded by fine crystalline matrix dolomite, upper Bigoray Member (3173 m bgl). (G) Residual coral (?) on the left and shell fragments (gastropods?) in micritic matrix (3205.8 m bgl, Lobstick Member). Light grey, oblong crytstal in center-left is authigenic quartz (qz). (F) - (G) represent the bank-edge-reef from well 2-19-48-12W5. (H) Fine crystalline matrix dolomite with ghost of allochems and dissolution seams (white arrows), representing the Dismal Creek Member at 3190.60 m bgl (11-32-47-12W5, bank, Nisku Formation).

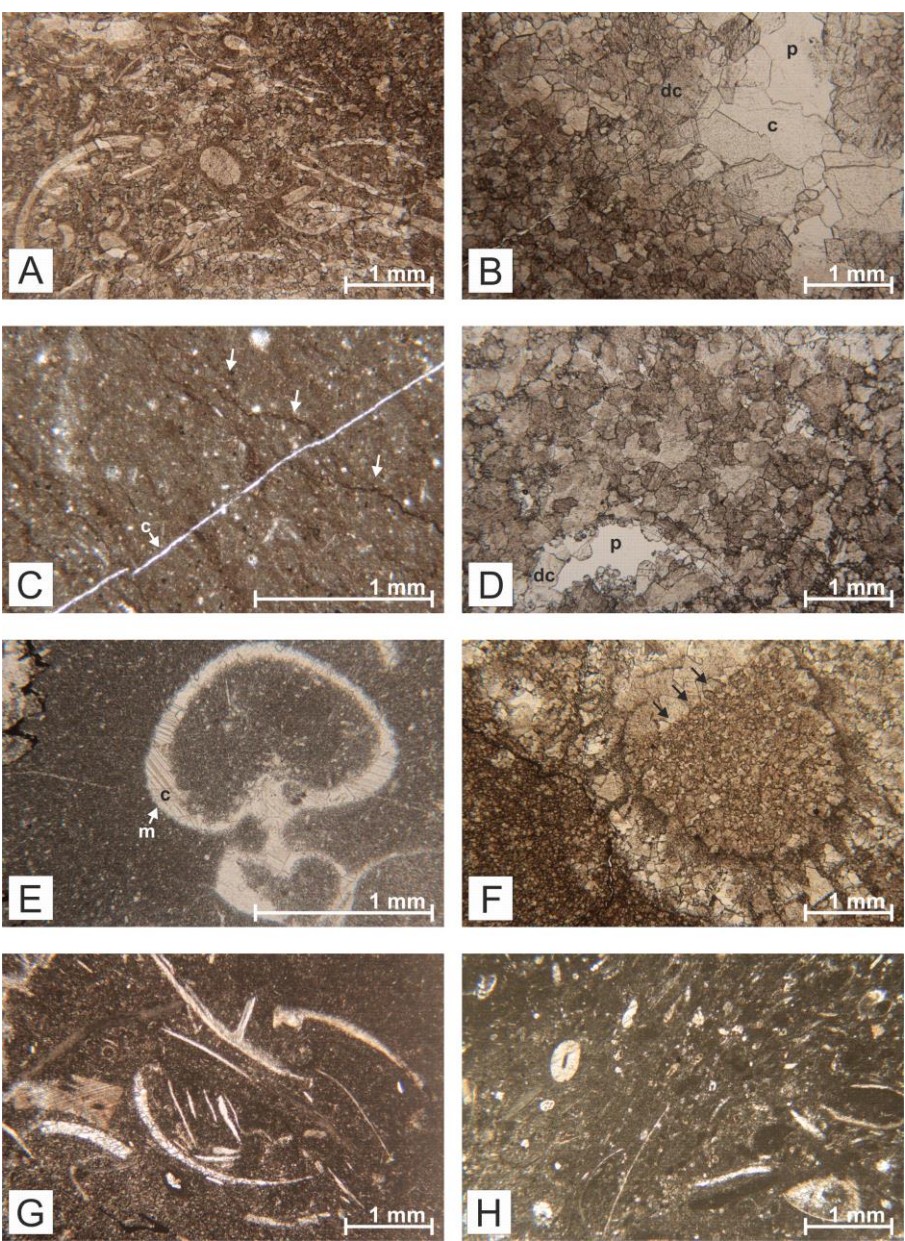

**Plate 6:** Unstained thin sections from outcrop samples. Most samples are completely dolomitized, except where noted otherwise. (A) Bioclastic (gastropods, mollusks) grainstone facies, Lower Cairn platform at Grassi Lakes. (B) Medium to ~~Coarse~~ coarse crystalline replacive dolomite (lower left) with void space that is lined with coarse crystalline dolomite cement **(dc)**, in turn overgrown by coarse crystalline calcite mosaic cement **(c)** that fills the remaining -pore space **(p)**. Cairn Formation at Toma Creek. Hairline cracks occur near the lower left corner. (C) Microstylolitic dissolution seams **(white arrows)** and narrow calcite veinlet **(c)** in dolomitized micritc limestone matrix. Perdrix Formation at Jasper Railroad. (D) Coarse crystalline planar-s to non-planar replacement dolomite (crystal diameters up to 500 μm) with small fossil mold **(p)** that is lined by dolomite cement **(dc)**. Peechee Formation at Grassi Lakes. (E) Fossil mold (goniatite?) filled with microsparitic calcite cement **(c)** in micritic limestone matrix. Peechee Formation at Jasper Railroad. (F) Geopedal filling **(black arrows)** in Disphyllid coral embedded in formerly mud matrix, now all dolomitized. Grotto Formation at Nigel Peak. (G) Floatstone facies, undolomitized, with fragments of tribolites, mollusks and enchinoderms. Perdrix Formation at Nigel Peak. (H) Floatstone facies, undolomitized, dominated by echinoderm fragments. Cairn / Perdrix Formation at Gap Lake.

**Table 5: Coordinates of the well locations and outcrops in the Front Ranges**

| Nr. | Well-ID | Location | Latitude dec deg (WGS84) | Longitude dec deg (WGS84) | X (UTM WGS84) | Y (UTM WGS84) | Zone |
|---|---|---|---|---|---|---|---|
| 1 | 7-33-48-12W5 | Nisku Reef Trend | 53.183281 | -115.692914 | 587345 | 5893457 | 11U |
| 2 | 2-19-48-12W5 | Nisku Reef Trend | 53.150544 | -115.741671 | 584151 | 5889757 | 11U |
| 3 | 11-32-47-12W5 | Nisku Reef Trend | 53.099565 | -115.723309 | 585480 | 5884108 | 11U |
| 4 | 5-22-48-27W4 | RMRT | 53.1541428 | -113.8754164 | 307740 | 5893279 | 12U |
| 5 | 10-31-37-9W5 | RMRT | 52.226049 | -115.272296 | 618005 | 5787587 | 11U |
| 6 | 16-18-61-15W5 | SCCC | 54.2814211 | -116.2296687 | 550152 | 6015107 | 11U |
| 7 | 2-36-54-23W5 | SCCC | 53.7027132 | -117.2535089 | 483264 | 5950476 | 11U |
| Nr. | Outcrop | Location | Latitude dec deg (WGS84) | Longitude dec deg (WGS84) | X (UTM WGS84) | Y (UTM WGS84) | Zone |
| 1 | Toma Creek | SCCC | 52.844521 | -117.218291 | 485298 | 5854997 | 11U |
| 2 | Jasper Railroad | SCCC | 52.919106 | -118.053117 | 429194 | 5863791 | 11U |
| 3 | Mt. Greenock | SCCC | 53.087822 | -118.063258 | 428790 | 5882568 | 11U |
| 4 | Nigel Peak | Fairholme Complex | 52.218794 | -117.180125 | 487695 | 5785389 | 11U |
| 5 | Grassi Lakes | Fairholme Complex | 51.072316 | -115.405563 | 611704 | 5659076 | 11U |
| 6 | Gap Lake | Fairholme Complex | 51.058531 | -115.231311 | 623948 | 5657822 | 11U |

