# Peer review of "From oil field to geothermal reservoir: First assessment for geothermal utilization of two regionally extensive Devonian carbonate aquifers in Alberta, Canada"

_Solid Earth, 2017_

## Short Comment (SC1) · 20 Dec 2017

Geothermal potential of Hinton, Alberta are has been recently getting attention in the scientific papers and in the Alberta media (CBC etc„). In fact, the area is in the deepest part of the Western Canada Sedimentary Basin WCSB and due to this large depth and at moderate geothermal gradient temperatures as high as 150 C were measured (Kushibor et.al., 1984). It takes some 5km drilling to get it in the deepest rather highly mineralized waters in the basinal sediments. The area has just an average geothermal gradient (see Figure 1) of 30-35 C/km.The geothermal gradient in the Western Canada

[Figure]

Sedimentary Basin WCSB ranges from 20 to 55 °C/km, with an average value of 33.2 °C/km (Weides and Majorowicz et al, 2014).

In the commented article the citation: "The area around the town site of Hinton in the western region of the Alberta Basin (Fig. 1) is of particular interest because it has heat flows of up to 80 mW m$^{-2}$ and temperatures up to 150 C at depths of about 5km" , credited to (Majorowicz and Weides, 2014) requires some corrections:

1.The reference should be Weides and Majorowicz (2014) as given below in the References.

2, Heat flow in the cited map in Weides and Majorowicz (2014, their Fig. 3) for the studied area of the Alberta basin is not reaching 80 mW m$^{-2}$ It is less than 70mW m$^{-2}$. The heat flow in the WCSB generally ranges from 30 to 100 mW/m2, being 60.4 mW/m2 on average according to Weides and Majorowicz (2014). The heat flow values has been corrected for paleoclimatic surface temperature forcing (Majorowicz et., al. 2012).

The attached average geothermal gradient map (Fig.1) shows that there are much 'hotter' areas in the WCSB in Alberta and these are to the north and east of deep part of the foreland basin in the Hinton area. These, however, will not give us temperatures of 150 C in the sediments, as the basin is shallower in the highest heat flow, geothermal gradient areas to the north. In Hinton, we need to drill 5km at some modest 30 C per km temperature gain to get to 150C. However, such deep wells are expensive and economics of drilling two 5km wells into the deepest sedimentary horizons will end up with extremely high mineralized waters and rather poor porosity/permeability .(Lam and Jones, 1985). Therefore, it may not be feasible for the geothermal power project. Drilling shallower wells into better conditions of Devonian aquifers of some 2-4km will come with the tradeoff of lower temperatures (<120 C ) not feasible for economic geothermal power production. These would be good for district heating applications (Majorowicz and Moore, 2014).

References:

Gray, A., Majorowicz, J., and Unsworth, M, 2012, Investigation of the geothermal state of sedimentary basins using oil industry thermal data: Case study from Northern Alberta exhibiting the need to systematically remove biased data, IOP J.Geoph. Eng., JGE/428217/PAP/128312, 2012.

Kushigbor,C., Lam, HL., Majorowicz, J.A, Rahman,M.:Estimates of terrestrial thermal gradients and heat flow variations with depth in the Hinton-Edson area of the Alberta basin derived from petroleum bottom-hole temperature data, Geophysical Prospecting, 32 (6), 1111-1130, 1984.

Lam, H., and Jones, F.: Geothermal energy potential in the Hinton-Edson area of west-central Alberta, Canadian Journal of 30 Earth Sciences, 22, 369–383, https://doi.org/10.1139/e85-036, 1985.

Majorowicz,J., Gosnold,W., Gray,A., Safanda, J.,Klenner,R., Unsworth, M.: Implications of post-glacial warming for northern Alberta heat flow-correcting for the underestimate of the geothermal potential, GRC Transactions, 36, 693-698, 2012.

Majorowicz, J. and Moore, M.:The feasibility and potential of geothermal heat in the deep Alberta foreland basin-Canada for $CO_2$ savings, Renewable Energy 66, 541-549, 2014.

Weides, S.,and Majorowicz, J.: Implications of Spatial Variability in Heat Flow for Geothermal Resource Evaluation in Large Foreland Basins: The Case of the Western Canada Sedimentary Basin, Energies, 7, 2573–2594, doi: 10.3390/en7042573, 2014.

................................................................................................................................Figures

Figure 1. Average geothermal gradient. Color scale is in °C/km.

[Figure]

**Fig. 1.**

**Legend**

**Oil Sand Areas**

**Thickness**
- <10m
- >10m
- >30m

Basement Depth (m)

Rivers

Parks

Provincial Boundaries

**<VALUE>**
- <10
- 10-15
- 15-20
- 20-25
- 25-30
- 30-35
- 35-40
- 40-45
- >45

Labeled locations: Peace River, Ft McMurray, Slave Lake, Cold Lake, HINTON, Edson, Edmonton

Scale: 0  50  100  200 Kilometers

---

## Referee Comment (RC1) · Anonymous Referee #1 · 21 Dec 2017

The presented dataset of thermo- and petrophysical properties of Upper Devonian carbonates from of the Alberta Basin is used to assess the geothermal potential of two main aquifer systems. The recent interest in the development of geothermal energy from deep sedimentary basins world-wide makes this study relevant to the international community and the topic is within the scope of SE. The paper is well structured, the abstract clearly reflects the methods, findings and results, scientific methods and assumptions are valid and clearly outlined, figures and tables are relevant and of good quality, interpretations and conclusions are plausible. Cited references are appropriate

and up to date.

However, besides the thermo- and petrophysical properties, the heat flow of the basin has to be taken into account to assess the economic feasibility. In the conclusions I would like to see this point addressed – alternatively the authors might add an outlook where they list the next steps to localize the most promising area with regard to depth and temperature including the available information from published data. Eventually, it is the economically recoverable heat (ERH) which increases the feasibility and this should be clearly stated at the end of the paper. The dataset clearly shows that the aquifer systems under discussion have to be operated as transitional system and thus need stimulation. Again, a point to be considered for economic operation. Since the target formations seem to be homogenous and mostly dolomitized, ultimately the depth of the reservoir with the relevant temperatures is crucial.

Please, check the reference list for consistency (also fonts). Can you please add the coordinates of the outcrop and well locations in Table 1.

---

## Referee Comment (RC2) · Anonymous Referee #2 · 28 Dec 2017

It is good paper which comes up with new data on the Upper Devonian aquifers potentially useful for the geothermal energy applications in the Western Canadian Sedimentary basin (WCSB) in its deep part area. Data from the outcrops are compared to deep drilling data . Important observations are made:

1. "As such, the outcrop analogues are no valid proxies for the buried reservoirs in the Alberta Basin. "

2. "the outcrop analogue samples have lower porosity and permeability, likely caused

by low-grade metamorphism and deformation during the Laramide Orogeny that formed the Rocky Mountains. As such, the outcrop analogues are no valid proxies for the buried reservoirs in the Alberta Basin. "

3. "Considering geothermal utilization, dolomitization enhanced all analyzed rock properties. "

I list below some comments to be considered by the Authors:

a) This paper gives new data on thermal conductivity & thermal diffusivity of carbonate rock samples. Thermal scanner was used (Popov et al., 1999). Some information on the samples prep. and orientation should be given (saturated or dry; whether or not the thermal conductivities are known to be the vertical (perpendicular thermal conductivity)? Authors also give results on density, porosity, permeability , etc.. As to compare above thermal conductivity , porosity new measurements with previously published results, I would recommend reference to Beach et al., Geothermics, Vol. 16, No. I, pp. 1-16, 1987 with averages based on hundreds of thermal conductivity and porosity for carbonates and other rock types from mainly Hinton- Edson area. Beach et al write : "The average thermal conductivity values for limestone, dolomite, shale, siltstone and sandstone were determined from analysis of measurements on drill-hole cores using divided-bar apparatus at UofA in the late 80th. . Table 1 in Beach et al (1987) gives the mean of the measured conductivities and uncertainties which are standard deviations from the means for the five rock types., ... Most of the samples were from the Hinton-Edson region of Alberta, " Some of the averages for the carbonate cores are: l Limestone 679 samples , thermal conductivity 2.42 +/- 0.88 ; porosity 3.2% ; Dolomite 254 samples, thermal conductivity 3.1 +/- 1.4, porosity 2.2 ; Anhydrite 7samples, thermal conductivity 5.8 +/- 1.1 – etc.,etc.

b) There are many statements related to an assessment of the geothermal energy potential of the carbonate aquifers and reefs in the study area. While porosity, permeanbility , temperature conditions, thermal conductivity ,diffusivity, are important to

such evaluation it is not possible to recommend geothermal energy potential without take on other parameters like the hydraulic head, piezometric surfaces, mineralization of aquifer fluids and most important estimate of potential flow rates at well head. In that sense cited by the authors paper by Jones and Lam Can. J. Earth Sci. 1985 went farther and gives such information (see their figs.10-12 and their Appendix figures). I recommend that their results be described, evaluated and briefly discussed in the scope of geothermal energy eval..

c) It is not entirely justified to make statements in the paper like this one : " great opportunities for further work toward potential geothermal utilization of its Devonian subsurface aquifers, especially because of the vast number of drillholes and attendant data bases, both public (AccuMap, Gescout, and others) as well as in the petroleum industry. Once promising aquifers or parts thereof have been identified, new economic strategies and industries could spring up in Alberta, for example by repurposing idle oil and gas wells for geothermal utilization. " There is no estimate of flow rates, energy to be produced and no economic evaluation of such projects given by the refereed paper!. I recommend to remove above over-optimistic statements and stick in to new findings on the aquifers and role of dolomitizantion in enhancing aquifer properties or the Authors should address other properties leading to estimate of potential flow rates which with temperature drop evaluation can give an estimates of energy available. At their paper The Autors do not address the issue of potential brine production as they do not address parameters needed to estimate it in their paper.

Re. References:

1.Majorowicz and Weides (2014) should be changed to Weides and Majorowicz (2014).

2. Reference to Beach et al is:

BEACH,R.D.W., JONES, F.W., MAJOROWICZ,J.A. (1987) HEAT FLOW AND HEAT GENERATION ESTIMATES FOR THE CHURCHILL BASEMENT OF THE WESTERN CANADIAN BASIN IN ALBERTA, CANADA, Geothermics, Vol. 16, No. I, pp. 1-16,

---

## Referee Comment (RC3) · Anonymous Referee #3 · 23 Jan 2018

This paper examines Devonian reefs as potential geothermal reservoirs in the Hinton area of Alberta. There are critical flaws with this work though. Most important is a shocking lack of reference to previous work, including numerous papers on the Hinton Geothermal Potential and numerous papers of the hydrogeological properties of the same aquifers studied. As it stands this works is completely out of context of earlier work and its not clear that it adds much new information to what is already known. Another major concern is that the authors do not seem to know the study area well and make several false claims as well as some simply outrageous political comments

that have no place in a science journal. In the current form I really don't think the MS should be published. The authors need to do a major rewrite that includes some basic background research to place their work in context and to be able to demonstrate what new knowledge they are adding. Other issues are listed below.

1) Introduction: the reason why Alberta has such a high percapita $CO_2$ emission is that it is developing the worlds second largest oil field, but with a very small population base (approximately 1o% of that of Saudi Arabia that is developing the worlds largest oil field). Most of Alberta oil is exported to the US, so its questionable $CO_2$ accounting to log it all against the producer rather than the consumer. Therefore the introduction provides some misleading statistics that have questionable value in a science paper. 2) Introduction: The reason for such small hydro usage in Alberta is that southern Alberta is the driest part of Canada – there are very limited opportunities for hydro in the province. But as a nation Canada has over 70% of power production by renewable energy, one of the cleanest grids in the world. So again, the intro has very misleading information. 3) Introduction: To suggest that there is a political climate that favours business over environment, and its doubtful if Alberta will want to transition to a cleaner energy system is simply outrageous – politics does not belong in a science paper. Besides, Albertans have recently elected a government that has one of the most aggressive environmental programs in North America, including implementing the largest carbon tax in Canada. Such statements that speak to the politics of a place the authors do not live in, and to speculate about future decisions Albertan's will make, have absolutely no place in a science paper. 4) The authors make a false claim that there is no geothermal utilisation. For direct heat use Canada is about 7th in the world (see summary by Raymond 2015). Also, within Alberta the Leduc reef is already being used for direct heat. There are also cleaver thermal storage systems in southern Alberta as well as direct heat use of waters produced from the Western Canada Sedimentary Basin in Saskatchewan, as well as planned drilling this year for electrical production wells. The authors clearly need to do more research on the state of geothermal usage in their study area. 5) Page 3, line 9: what is Malm? 6) Page 3,

line 10. This gets very confusing,. Are you meaning that the overall project studies 3 aquifers or this paper? On line 12 you say there are only two aquifers in Alberta you study which is the subject of this paper. Its also not clear what is meant by two of four? What four? 7) Page 3, ln 17: This discussion on German aquifers seems out of place in a paper on Alberta, what is the relevance? I would remove this section. 8) Page 3, ln 23: It seems very odd to introduce looking at outcrops to understand the subsurface as some kind of new approach. Geologists in the petroleum and mining industry have been doing this pretty much ever since geology was invented – its pretty much the very foundation of geology in fact. This is nothing new and certainly not an original idea of Homuth et al 2015. 9) Page 4, ln 2: Delete 'literally' and also, use the Canadian spelling of "Centre" not 'Center' as that is the formal spelling of the Core Centre. 10) Page 4, ln 3 ".., results of drill stem test.." 11) Page 4, ln 6: There is a surprising lack of reference to numerous previous geothermal studies in the Hinton area, including some on the same reef systems. As well, since these units also produced major oil fields there has been extensive research conducted on the hydrogeology. A simple web search for terms like 'Hinton geothermal' or 'Nisku hydrogeology' will provide the authors with numerous papers of relevance that should be cited. Rather than 'superficially' I would say the area has been extensively studied. Try a bit of background research as part of your study! 12) Page 4, ln 9: here you say 3 aquifers and above it was two? 13) Page 4, ln 21: the formal name is 'Rocky Mountains' 14) Page 4, ln 25: closer to 2 million. 15) Page 5, ;n 32: how does this reef trend relate to the Leduc ? need more clear descriptions of everything, same for page 6, ln 9 and 26. Lots of various terms used with no clear description what they all are. 16) Page 9, ln 11: need ref for timing of larimide 17) Page 9, line 13: if there are similar fractures in both, isn't I more reasonable that they have the same origin, rather than invoking two different ones? 18) Section 5.2 this is not petrography

---

## Referee Comment (RC4) · J. Borgomano (Referee) · 12 Feb 2018

[referee-annotated manuscript omitted]

---

## Author Comment (AC1) · 11 Mar 2018

Leandra M. Weydt[1], Claus-Dieter J. Heldmann[1], Hans G. Machel[2], Ingo Sass[1,3]

[1]Department of Geothermal Science and Technology, Technische Universität Darmstadt, Schnittspahnstraße 9, 64287 Darmstadt, Germany
[2]Earth and Atmospheric Sciences, University of Alberta, Edmonton, AB T6G 2E3, Canada
[3] Darmstadt Graduate School of Excellence Energy Science and Engineering, Jovanka-Bontschits-Straße 2, 64287 Darmstadt, Germany

*Correspondence to*: Leandra M. Weydt (weydt@geo.tu-darmstadt.de)

**Author's comment on "Referee comment 1 – Review MS se-2017-129" by Anonymous Referee 1**

Dear Reviewer,

Thank you for your useful comments. Please see the answers and changes below:

**Referee 1 – C2 line 1:** *"However, besides the thermo- and petrophysical properties, the heat flow of the basin has to be taken into account to assess the economic feasibility. In the conclusions I would like to see this point addressed – alternatively the authors might add an outlook where they list the next steps to localize the most promising area with regard to depth and temperature including the available information from published data. Eventually, it is the economically recoverable heat (ERH) which increases the feasibility and this should be clearly stated at the end of the paper."*

**Answer:** We agree with Referee 1 that heat flow is an important parameter and should be taken into account during the assessment of the economic feasibility of a geothermal reservoir. This study was intended to create an initial data set of Upper Devonian carbonate rock properties relevant to geothermal modelling. Statements about economic feasibility were not planned for this early stage of the project, however, a short section will be added to the chapter "discussion and conclusions" summarizing the most important parameters necessary to assess the economic feasibility of this reservoir.

Outlook: To complement the data set presented in this study, measurement of further parameters (e. g. thermal diffusivity, specific heat capacity and ultrasonic wave velocity) on well core samples has been planned. Well data provided in the AccuMap or GeoScout databases will be evaluated, interpreted and probably mapped to identify the most promising areas for geothermal utilization in the reservoir. Due to the high amount of well data, this will be only possible on a regional scale.

The construction of a regional geological 3D model, using already existing data from previous studies (Majorowicz et al., 2012; Nieuwenhuis et al, 2015), is planned for the most promising areas.

**Referee 1 – C2 line 7:** *"The dataset clearly shows that the aquifer systems under discussion have to be operated as transitional system and thus need stimulation. Again, a point to be considered for economic operation."*

**Answer:** Agreed. This point needs to be considered during cost calculation and for economic operation. We will add this point in chapter 6 "Discussion and conclusions".

**Referee 1 – C2 line 12:** *"Please, check the reference list for consistency (also fonts)."*

**Answer:** Thank you very much for the detailed proofreading. The reference list was checked and corrected accordingly.

**Referee1 – C2 line 12:** *"Can you please add the coordinates of the outcrop and well locations in Table 1."*

**Answer:** A list of coordinates of the outcrops and well locations was added to Appendix B.

**Table 1: Coordinates of the well locations and outcrops in the Front Ranges**

| Nr. | Well-ID | Location | Latitude (WGS84) | Longitude (WGS84) | X (UTM WGS84) | Y (UTM WGS84) | Zone |
|---|---|---|---|---|---|---|---|
| 1 | 7-33-48-12W5 | Nisku Reef Trend | 53.183281 | -115.692914 | 587345 | 5893457 | 11U |
| 2 | 2-19-48-12W5 | Nisku Reef Trend | 53.150544 | -115.741671 | 584151 | 5889757 | 11U |
| 3 | 11-32-47-12W5 | Nisku Reef Trend | 53.099565 | -115.723309 | 585480 | 5884108 | 11U |
| 4 | 5-22-48-27W4 | RMRT | 53.1541428 | -113.8754164 | 307740 | 5893279 | 12U |
| 5 | 10-31-37-9W5 | RMRT | 52.226049 | -115.272296 | 618005 | 5787587 | 11U |
| 6 | 16-18-61-15W5 | SCCC | 54.2814211 | -116.2296687 | 550152 | 6015107 | 11U |
| 7 | 2-36-54-23W5 | SCCC | 53.7027132 | -117.2535089 | 483264 | 5950476 | 11U |
| Nr. | Outcrop | Location | Latitude (WGS84) | Longitude (WGS84) | X (UTM WGS84) | Y (UTM WGS84) | Zone |
| 1 | Toma Creek | SCCC | 52.844521 | -117.218291 | 485298 | 5854997 | 11U |
| 2 | Jasper Railroad | SCCC | 52.919106 | -118.053117 | 429194 | 5863791 | 11U |
| 3 | Mt. Greenock | SCCC | 53.087822 | -118.063258 | 428790 | 5882568 | 11U |
| 4 | Nigel Peak | Fairholme Complex | 52.218794 | -117.180125 | 487695 | 5785389 | 11U |
| 5 | Grassi Lakes | Fairholme Complex | 51.072316 | -115.405563 | 611704 | 5659076 | 11U |
| 6 | Gap Lake | Fairholme Complex | 51.058531 | -115.231311 | 623948 | 5657822 | 11U |

Further corrections:

Table 1: The depth levels of well 2-36 and 16-18 in Table 1 were reversed. The analysed depth interval of well 16-18 is now "2741.00 m to 2779.77 m" and the analysed depth intervals of well 2-36 are "4068.77 m to 4095.00 m" and "4145.00 m to 4165.39 m" as shown in Fig. 9.

5

Table 2 – Perdrix Formation: The number of measured plugs for the density measurements is N = 17.

Page 6 line 22: It is IHS instead of HIS – also in the references. I apologize for the unfortunate auto correction.

**References**

10  Majorowicz,J., Gosnold,W., Gray,A., Safanda, J.,Klenner,R., Unsworth, M.: Implications of post-glacial warming for northern Alberta heat flow-correcting for the underestimate of the geothermal potential, GRC Transactions, 36, 693-698, 2012.

Nieuwenhuis, G., Lengyel, T., Majorowicz, J., grobe, M., Rostron, B., Unsworth, M. J. and Weides, S.: Regional-Scale Geothermal Exploration Using Heterogeneous Industrial Temperature Data; a Case Study from the Western Canadian

15  Sedimentary Basin, Proceedings of the World Geothermal Congress,19-25 April 2015, Melbourne, Australia, 5 pp., 2015.

---

## Author Comment (AC2) · 11 Mar 2018

Leandra M. Weydt[1], Claus-Dieter J. Heldmann[1], Hans G. Machel[2], Ingo Sass[1,3]

[1]Department of Geothermal Science and Technology, Technische Universität Darmstadt, Schnittspahnstraße 9, 64287 Darmstadt, Germany
[2]Earth and Atmospheric Sciences, University of Alberta, Edmonton, AB T6G 2E3, Canada
[3] Darmstadt Graduate School of Excellence Energy Science and Engineering, Jovanka-Bontschits-Straße 2, 64287 Darmstadt, Germany

*Correspondence to*: Leandra M. Weydt (weydt@geo.tu-darmstadt.de)

**Author's comment on "Short comment 1 – Heat flow in the Hinton area, Alberta, Canada" by Jacek Majorowicz**

Thank you very much for the kind advice and the detailed proofreading. In accordance with your suggestions for improvement, we have amended the relevant parts in the manuscript as shown below:

**Jacek Majorowicz – C2 line 7:** "*1.The reference should be Weides and Majorowicz (2014) as given below in the References.*"

**Answer:** I apologize for this mistake. It has been corrected accordingly.

**Jacek Majorowicz – C2 line 9:** "*2, Heat flow in the cited map in Weides and Majorowicz (2014, their Fig. 3) for the studied area of the Alberta basin is not reaching 80 mW m-2 It is less than 70mW m2. The heat flow in the WCSB generally ranges from 30 to 100 mW/m2, being 60.4 mW/m2 on average according to Weides and Majorowicz (2014). The heat flow values has been corrected for paleoclimatic surface temperature forcing (Majorowicz et., al. 2012).*"

**Answer:** Thank you very much for this correction. The associated section in chapter "1 Introduction" in page 3 line 1 has been changed to: "The area around the town site of Hinton in the western region of the Alberta Basin (Fig. 1) is of particular interest because well data analysis indicates flow rates of more than 400 m³ h$^{-1}$ and temperatures up to 150 °C at depths of approximately 5 km (Lam and Jones, 1985)." General information about the geothermal gradient and heat flow in the WCSB was added to page 2, line 30: "This appears feasible because, although this province is characterized as a 'low enthalpy region' (Grasby et al., 2012; Lam and Jones, 1985 and 1986) with a moderate average geothermal gradient of 33.2 °C km$^{-1}$ and an average heat flow of 60.4 W m$^{-2}$ in the WCSB, recent studies using data from several tens of thousands of oil and gas

wells suggest that at least some of the Upper Devonian carbonate aquifers are suitable for geothermal utilization (Weides and Majorowicz, 2014)".

**Jacek Majorowicz – C2 line 15**:*"The attached average geothermal gradient map (Fig.1) shows that there are much 'hotter' areas in the WCSB in Alberta and these are to the north and east of deep part of the foreland basin in the Hinton area."*

**Answer**: This is correct. The two carbonate complexes were selected because

- A lot of communities in Alberta are located in this area (this applies especially for the Rimbey-Meadowbrook Reef Trend)
- There is increasing public interest in geothermal energy utilization in the Hinton-Edson area (Southesk-Cairn Carbonate Complex)
- the general growing interest in repurposing abandoned oil and gas wells to find new possibilities to reduce the demand of fossil fuels and to reduce $CO_2$-emissions
- there are similarities in rock type, depth and structure of the Devonian aquifer systems with the Jurassic Malm-aquifer in the Southern German Molasse Basin, which offers the possibility of knowledge transfer between Alberta and Germany.
-

**Jacek Majorowicz – C2 line 20**:"*However, such deep wells are expensive and economics of drilling two 5km wells into the deepest sedimentary horizons will end up with extremely high mineralized waters and rather poor porosity/permeability (Lam and Jones, 1985).*"

**Answer:** This pilot study was intended to be an initial inquiry into the the Upper Devonian carbonates with respect to geothermal utilization and to create an initial data set of rock properties relevant to geothermal exploration and modelling. To assess the economic feasibility of this reservoir more data is needed, e.g. the corrected heat flow and temperature data (Majorowicz et al., 2012; Nieuwenhuis et al., 2015), salinity, flow rates, potentiometric surfaces, and hydraulic heads to name just a few. The high TDS content in the formation waters of the Leduc and Nisku formations (Rostron et al., 1997; Bachu et al., 2008) is a critical parameter for geothermal production.

Therefore, well data provided in the AccuMap or GeoScout databases need to be evaluated and interpreted carefully, which is beyond of the scope of this study. A short section will be added to the chapter "discussion and conclusions" to provide an overview of further steps.

We don't agree with the statement that porosity and permeability are generally poor at these depth levels. According to Amthor et al. (1994), there is an overall decrease of porosity/permeability with depth in the Leduc Formation in the WCSB, but it has also been shown that especially the dolomitized reef sections retain their porosity/permeability compared to limestones at the same depth level or compared to well cores which are located in the shallower parts of the basin. An example is presented in this manuscript: Well 2-36-54-23W5 (>4 km depth), located in the western part of the Southesk-

Cairn Carbonate Complex, shows the highest porosity and permeability of all wells in this dataset. Porosity and permeability can be highly variable within aquifers at a local scale. However, it must be taken into account that high porosity and permeability values do not naturally guarantee high flow rates. Therefore, a careful evaluation of the well data and other existing information on the reservoir must be carried out to localize the most promising areas.

**5    References**

Amthor, J. E., Mountjoy, E. W., and Machel, H. G.: Regional-scale porosity and permeability variations in Upper Devonian Leduc buildups: implications for reservoir development and prediction in carbonates, AAPG Bulletin,78, 1541–1559, 1994.

Bachu, S., Buschkuehle, M., Michael, K.: Subsurface characterization of the Brazeau Nisku Q pool reservoir for acid gas injection, Energy Resources Conservation Board, ERC B/AGS special Report 095, 62 p., 2008.

Grasby, S. E., Allen, D. M., Chen, Z., Ferguson, G. J., Kelman, M., Majorowicz, J., and Therrien, R.: Geothermal Energy Resource Potential of Canada, Geological Survey of Canada, Calgary, AB, Canada, 322 pp., 2012.

Lam, H., and Jones, F.: Geothermal energy potential in the Hinton-Edson area of west-central Alberta, Canadian Journal of Earth Sciences, 22, 369–383, https://doi.org/10.1139/e85-036, 1985.

Lam, H., and Jones, F.: An investigation of the potential for geothermal energy recovery in the Calgary area in southern Alberta, Canada, using petroleum exploration data. Geophysics, 51, 1661–1670, doi: 10.1190/1.1442215,1986.

Nieuwenhuis, G., Lengyel, T., Majorowicz, J., grobe, M., Rostron, B., Unsworth, M. J. and Weides, S.: Regional-Scale Geothermal Exploration Using Heterogeneous Industrial Temperature Data; a Case Study from the Western Canadian Sedimentary Basin, Proceedings of the World Geothermal Congress,19-25 April 2015, Melbourne, Australia, 5 pp., 2015.

Rostron, B. J., Toth, J., and Machel, H. G.: Fluid flow, hydrochemistry, and petroleum entrapment in Devonian reef complexes, south-central Alberta, Canada, in: Basin-wide diagenetic patterns: integrated petrologic, geochemical, and hydrologic considerations, Montanez, I. P., Gregg, J. M., and Shelton, K. L. (Eds.), Society for Sedimentary Geology Special Publication, No. 57, 139–155, 1997.

Majorowicz,J., Gosnold,W., Gray,A., Safanda, J.,Klenner,R., Unsworth, M.: Implications of post-glacial warming for northern Alberta heat flow-correcting for the underestimate of the geothermal potential, GRC Transactions, 36, 693-698, 2012.

Weides, S.,and Majorowicz, J.: Implications of Spatial Variability in Heat Flow for Geothermal Resource Evaluation in Large Foreland Basins: The Case of the Western Canada Sedimentary Basin, Energies, 7, 2573–2594, doi: 10.3390/en7042573, 2014.

---

## Author Comment (AC3) · 11 Mar 2018

Leandra M. Weydt[1], Claus-Dieter J. Heldmann[1], Hans G. Machel[2], Ingo Sass[1,3]

[1]Department of Geothermal Science and Technology, Technische Universität Darmstadt, Schnittspahnstraße 9, 64287 Darmstadt, Germany
[2]Earth and Atmospheric Sciences, University of Alberta, Edmonton, AB T6G 2E3, Canada
[3] Darmstadt Graduate School of Excellence Energy Science and Engineering, Jovanka-Bontschits-Straße 2, 64287 Darmstadt, Germany

*Correspondence to*: Leandra M. Weydt (weydt@geo.tu-darmstadt.de)

**Author's comment on "Referee comment 2 – Reviewer 2 Comments to: From oil field to geothermal reservoir: First assessment for geothermal utilization of two regionally extensive Devonian carbonate aquifers in Alberta, Canada" by Anonymous Referee 2**

Dear Reviewer,

Thank you for processing our manuscript and for your very valuable comments. Your questions and advice are answered below.

**Referee 2 – C2 line 8:** "*Some information on the samples prep. and orientation should be given (saturated or dry; whether or not the thermal conductivities are known to be the vertical (perpendicular thermal conductivity)?"*

**Answer:** Thermal conductivity was measured according to the method of Popov et al. (1999) on dry samples as shown in Fig. 1. Therefore, thermal conductivity of the core samples represents the "horizontal" thermal conductivity. We will add Fig. 1 to the relevant chapter, Material and Methods, in the manuscript.

Sample preparation: To minimize the transmission of optical heater radiation into reference standards and rock samples resulting from optical transparent surfaces (Popov et al., 2016), black paint was applied along a scan line on the sample surfaces as well as on the standards. Measurement of the plane surfaces of the core samples in order to estimate the thermal anisotropy was not allowed.

[Figure]

Figure 1: Schema of the thermal conductivity measurements of the analyzed core samples (Sass and Götz, 2012).

**Referee 2 – C2 line 11:** *"As to compare above thermal conductivity , porosity new measurements with previously published*
5    *results, I would recommend reference to Beach et al., Geothermics, Vol. 16, No. I, pp. 1-16, 1987 with averages based on*
*hundreds of thermal conductivity and porosity for carbonates and other rock types from mainly Hinton- Edson area."*
**Answer:** Ok.
The data set included in Beach et al. 1987 comprises thermal conductivity values measured on several hundred "water-
saturated porous samples" (Beach et al. 1987, p.3). These thermal conductivity values were measured with a divided-bar
10   apparatus and are stated as the vertical or perpendicular thermal conductivity.
This data set was not mentioned in the manuscript because it does not contain any information about the origin (well location
and depth) of the samples, a reference to the different formations in the basin, nor a detailed rock description. The data set
provided in Beach et al. (1987) gives a good overview of thermal conductivity of 13 rock types of the Mesozoic, Cenzcoic
and Paleozoic sediments in the Hinton-Edson area. The thermal conductivity values of each rock type represent mean values
15   calculated from different depth levels with varying thickness. The data set gives no information about specific formations
and how the properties change within a formation. Therefore, this data set is more useful for large scale observations. In
Jones et al. (1984) thermal conductivity in the Hinton-Edson area (most likely the same data set – 936 water saturated
samples from 48 wells measured with a divided bar apparatus) is given for four geological formation groups. Jones et al.
(1984) states that although a lot of samples were analysed, that some came from very small depth intervals and most of the

cores are from very porous and permeable formations. Therefore, this data set is not representative for all relevant parts in the reservoir.

The aim of this study was to provide a data set specific to the Upper Devonian carbonates which have become of particular interest for geothermal utilization and also for identifying variations of rock properties within its aquifer systems. According to Popov et al (2016), thermal rock properties are critical parameters for thermo-hydrodynamic models and for predicting the lifetime performance of geothermal systems. Therefore, an accurate determination of thermal properties of each relevant formation in the reservoir is necessary. The advantage of the method from Popov et al. (1999) is that it offers quick, non-destructive and contact free measurements on plane or cylindrical samples. Cutting the core samples to a specific size is not required and thus it allowed us to measure all available and intact core samples of each selected well core. Measurements on full size core samples are useful for analysing the formation's heterogeneity.

The thermal conductivity values presented in this manuscript were measured on dry samples, not on saturated samples. It is to emphasize that thermal conductivity values measured on dry samples as well as thermal conductivity values measured on water-saturated samples do not reflect the real conditions in the reservoir and need to be corrected before modelling.

**Referee 2 – C2 line 25:** "*b) There are many statements related to an assessment of the geothermal energy potential of the carbonate aquifers and reefs in the study area. While porosity, permeanbility , temperature conditions, thermal conductivity ,diffusivity, are important to such evaluation it is not possible to recommend geothermal energy potential without take on other parameters like the hydraulic head, piezometric surfaces, mineralization of aquifer fluids and most important estimate of potential flow rates at well head. In that sense cited by the authors paper by Jones and Lam Can. J. Earth Sci. 1985 went farther and gives such information (see their figs.10-12 and their Appendix figures). I recommend that their results be described, evaluated and briefly discussed in the scope of geothermal energy eval..*"

**Answer:** Ok. As mentioned before, at this early stage of the project, we focused on examining the Upper Devonian carbonates for geothermal purposes and to measure rock properties which are relevant to geothermal exploration and modelling. The classifications by Sass and Götz (2012) and Bär et al. (2011) were used for the initial evaluation of the measured rock properties. We do not claim that they replace further investigation. As mentioned in the manuscript (p. 13, line 22) "rock property measurements produce conservative results and represent matrix properties only" and additional parameters need to be integrated in a geological model for a reliable reservoir prediction.

Therefore, our statements are not contradictory to the comments by Referee 2. Due to the high number of well data (several thousand wells) that need to be evaluated for a reliable assessment of the Upper Devonian aquifer systems in the study area, the parameters required by Referee 2 are not included in this manuscript. This will be considered in the next phase of the project.

To make this clear, we will add a short section about the most relevant parameters and give an outlook of the next steps according to the hints of Referee 2.

**Referee 2 – C3 line 8:** *"It is not entirely justified to make statements in the paper like this one: ..."*

**Answer:** Agreed.

**Referee 3 – C3 line 20:** *"At their paper The Autors do not address the issue of potential brine production as they do not address parameters needed to estimate it in their paper."*

**Answer:** Thank you very much for this advice. As mentioned above, this manuscript focuses on rock properties. It was not intended for statements about economic and technical risks during production.

Regarding the recent efforts in the Hinton-Edson area to create an initial geothermal project, it is important to consider the complex hydrogeology in these aquifer systems. Both formations (Leduc and Nisku) contain highly concentrated waters with average TDS values of approximately 200 g/l (Rostron et al., 1997; Michael et al., 2003). Likewise, the Nisku Formation is well known for its sour gas pools (Bachu et al., 2008).

Problems like scaling and corrosion during operation can lead to higher production costs or, in the worst case scenario, to the abandonment of the well. These problems have not been solved in the geothermal industry yet.

The hydrochemistry of the aquifer systems in the study area has been the subject of several previous studies (e.g. Lam and Jones, 1985; Rostron et al., 1997; Buschkuehle and Machel, 2002; Michael et al., 2003, Machel and Buschkuehle, 2008) because it was also the main interest of the oil industry. As Lam and Jones (1985) have showed, these parameters can be very variable on a local scale. In the Hinton-Edson area the salinity ranges from less than 50 g/l up to 180 g/l.

**References**

Bachu, S., Buschkuehle, M., Michael, K.: Subsurface characterization of the Brazeau Nisku Q pool reservoir for acid gas injection, Energy Resources Conservation Board, ERC B/AGS special Report 095, 62 p., 2008.

Bär, K. **,**Arndt, D. **,** Fritsche, J.-G. , Kracht, M. **,** Hoppe, A.**,** and Sass, I.**:**
3D-Modellierung der tiefengeothermischen Potenziale von Hessen — Eingangsdaten
und Potenzialausweisung, Zeitschrift der Deutschen Gesellschaft für Geowissenschaften, 162, 371–388, 2011.

Beach, R, D. W., Jones, F. W. and Majorowicz, J. A.: HEAT FLOW AND HEAT GENERATION ESTIMATES FOR THE CHURCHILL BASEMENT OF THE WESTERN CANADIAN BASIN IN ALBERTA, CANADA, Geothermics, 16, 1-16, 1987.

Buschkuehle, B. E., and Machel, H. G.: Diagenesis and paleofluid flow in the Devonian Southesk-Cairn carbonate complex in Alberta, Canada. Marine and Petroleum Geology, 19, 219–227, 2002.

Jones, F. W., Kushigbor,C., Lam, HL., Majorowicz, J.A, and Rahman,M.:Estimates of terrestrial thermal gradients and heat flow variations with depth in the Hinton-Edson area of the Alberta basin derived from petroleum bottom-hole temperature data, Geophysical Prospecting, 32 , 1111-1130, 1984.

Lam, H., and Jones, F.: Geothermal energy potential in the Hinton-Edson area of west-central Alberta, Canadian Journal of Earth Sciences, 22, 369–383, https://doi.org/10.1139/e85-036, 1985.

Machel, H. G., and Buschkuehle, B. E.: Diagenesis of the Devonian Southesk-Cairn Carbonate Complex, Alberta, Canada: Marine Cementation, Burial Dolomitization, Thermochemical Sulfate Reduction, Anhydritization, and Squeegee Fluid Flow, Journal of Sedimentary Research, 78, 366–389, 2008.

Michael, K., Machel, H. G., and Bachu, S.: New insights into the origin and migration of brines in deep Devonian aquifers, Alberta, Canada, Journal of Geochemical Exploration, 80, 193–219, 2003.

Popov, Y. A., Sass, P. D., Williams, C. F., and Burkhardt, H.: Characterization of rock thermal conductivity by high resolution optical scanning, Geothermics, 28, 253–276, https://doi.org/10.1016/S0375-6505(99)00007-3, 1999.

Popov, Y., Beardsmore, G, Clauser, C. and Roy, S.: ISRM Suggested Methods for Determining Thermal Properties of Rocks from Laboratory Tests at Atmospheric Pressure, Rock Mech Rock Eng, 49, 4179-4207, https://doi.org/10.1007/s00603-016-1070-5, 2016.

Rostron, B. J., Toth, J., and Machel, H. G.: Fluid flow, hydrochemistry, and petroleum entrapment in Devonian reef complexes, south-central Alberta, Canada, in: Basin-wide diagenetic patterns: integrated petrologic, geochemical, and hydrologic considerations, Montanez, I. P., Gregg, J. M., and Shelton, K. L. (Eds.), Society for Sedimentary Geology Special Publication, No. 57, 139–155, 1997.

Sass, I. and Götz, A.: Geothermal reservoir characterization: a thermofacies concept, Terra Nova, 24, 142–147, doi: 10.1111/j.1365-3121.2011.01048.x, 2012.

---

## Author Comment (AC4) · 11 Mar 2018

Leandra M. Weydt[1], Claus-Dieter J. Heldmann[1], Hans G. Machel[2], Ingo Sass[1,3]

[1]Department of Geothermal Science and Technology, Technische Universität Darmstadt, Schnittspahnstraße 9, 64287 Darmstadt, Germany
[2]Earth and Atmospheric Sciences, University of Alberta, Edmonton, AB T6G 2E3, Canada
[3]Darmstadt Graduate School of Excellence Energy Science and Engineering, Jovanka-Bontschits-Straße 2, 64287 Darmstadt, Germany

*Correspondence to*: Leandra M. Weydt (weydt@geo.tu-darmstadt.de)

**Author's comment on "Referee comment 4 – general comment" by Jean Borgomano**

Dear Jean Borgomano,

Thank you very much for your very valuable and helpful comments to improve our manuscript. Please see our answers below.

**Referee 4 – C1 line 11:** *"a missing topic in this interesting chapter is the "upscaling-donwscaling" of the properties (reservoir and geothermal)"*

**Answer:** We agree that upscaling and downscaling of the rock properties is essential for reservoir modelling. As far as we did not model in this study, we did not elaborate this point in this manuscript. For the upscaling of the rock properties from plug to reservoir scale, further data should be included, which is beyond the scope of this study.

So we agree to give information about "upscaling-downscaling" according to the hint of Referee 4:

- upscaling in porous media: e.g. Renard et al. (1997), Farmer (2002)
- stratigraphy based upscaling permeability and porosity in carbonate platforms: e. g. Leonide et al. (2012), Borgomano et al. (2013), Brigaud et al. (2014)
- upscaling of thermal conductivities: Rühaak et al. (2015).

Reservoir properties of carbonate rocks and their upscaling are related to architecture and facies. Nevertheless, our results show that (pervasive) dolomitization has affected the reservoir parameters of the Western Canadian Sedimentary Basin in an equitable degree.

**Referee 4 – C2 line 1:** *"Another implicit assumption is that rock properties statistics are not dependent of sample size (no "support effect")."*

**Answer:** We do not claim that property statistics are independent from sample size. According to the hint of Referee 4 we will consider this point and add it to the section about upscaling and downscaling.

**Further comments are included in in the supplementary material: https://www.solid-earth-discuss.net/se-2017-129/se-2017-129-RC4-supplement.pdf**

**Referee 4 – p. 3 line 21:** *"what are precisely these "properties" and at what scale are they considered? Be more precise*
10 *here"*

**Answer**: Homuth et al. (2015) analyzed density, porosity, permeability, thermal conductivity, thermal diffusivity and specific heat capacity on plugs from more than 350 samples taken from 19 outcrops in the Swabian and Franconian Alb as well as on core samples and cutting material from wells in the Molasse Basin in order to obtain a large enough data base for upscaling and correlation of the geothermal properties. The rock samples were classified with respect to the dominant
15 reservoir facies types in order to identify facies dependant trends. To transfer the petrophysical properties to reservoir conditions, a Thermo-Triaxial-Cell was used to simulate pressure and temperature conditions of the reservoir. Furthermore, data from previous studies (bio- and lithostratigraphic model of the target formation) and well data were included.

**Referee 4 – p.3 line 24:** *„if you mean "physical-chemical rock properties" it needs to be demonstrated that outcrop and*
20 *subsurface rocks are analogues (given different burial, structural, and diagenetic history, including telogenesis); if you mean facies, stratigraphic architecture/diemensions, sedimentrry bodies etc...the analogy is probably ok"*

**Answer**: The term "analogue" in this manuscript refers to "facies, stratigraphic architecture, and sedimentary bodies".

Our study results are in agreement with your comment, as described later on in chapter "discussion and conclusions" (p 14, line 25ff).

25

**Referee 4 – p. 3 line 29:** *"how do you measure thermophysical properties at all these scales?"*

**Answer**: The multi-scale method includes analysis on three different scales 1) macro scale (outcrop, well data) 2) meso scale (rock samples/plugs) and 3) micro scale (thin section, chemical analysis etc.). Macro-scale studies include well data (e.g. hydraulic tests, heat flow) as well as outcrop investigation (facies, structures). It is to emphasize, that not every parameter
30 can be "measured" on every scale. In this manuscript we focused on the meso and micro scale (rock samples and thin sections).

**Referee 4 – p. 3 line 30:** *"it is not because you are doing multi-scale assessment that the properties or charactéristics are automatically upscales and downscaled! It depends also of the intrinsic heterogeneity of the system at all scales"*

**Answer**: We agree with Referee 4. To make it clear, we will add a short section about upscaling and downscaling in the manuscript.

**Referee 4 – p.8 line 9:** *"what are the key parameters for this classification? Are they universal or are they dependent of local conditions, basin, etc..."*

**Answer**: This classification especially considers economic and technological factors for geothermal utilization of the Federal state of Hesse in Germany. To assess the rock and reservoir properties in the 3D structural model of the Federal State of Hesse, a multiple criteria approach was used to incorporate their relevance for geothermal systems (Bär et al., 2011, Bär and Sass, 2014). Based on an extensive data base of rock properties and reservoir data, the threshold values (very low to very high) were defined to specify the geothermal potential. Since there exist no such data base for the Alberta Basin (and also no experience with geothermal power/heat generation), this classification was applied to the rock properties presented in this manuscript for a first evaluation.

**Referee 4 – p. 9 line 20:** *"Petrography Outcrops" changed to "Outcrop petrography"*

**Answer**: Accepted.

**Referee 4 – p. 11 line 31:** *"dry samples? It may change when samples are saturated with water?"*

**Answer**: Sentence will be changed to: "… The sample set shows no correlation between thermal conductivity and porosity." According to Popov et al. (2016) thermal conductivity of rocks depends on the mineral composition, interstitial porosity and fluids filling pores and fractures as well as mineral grain size and orientation, anisotropy of the rock matrix and the nature of grain contacts. Thermal conductivity of water is ~ 0.6 W $m^{-1}$ $K^{-1}$, while thermal conductivity of air is approx. 0,025 W $m^{-1}$ $K^{-1}$. Therefore, bulk thermal conductivity generally increases with increasing water content depending on the porosity of the rock sample. Likewise, bulk thermal conductivity of a rock sample generally decreases with increasing porosity.

Our data set indicates no correlation between thermal conductivity and porosity. It is most likely, that in this case mineral composition, grain size/geometry and grain contacts have a stronger influence on thermal conductivity than the porosity of the samples (plug scale).

**Referee 4 – p. 12 line 2:** *"that is in contradiction with the statement in the introduction."*

**Answer**: The "thermo facies concept" was explained in chapter 1 "Introduction" by the example of the Southern German Molasse Basin. This concept was successfully applied in the Southern German Molasse Basin to assess the geothermal potential of the Malm-Aquifer (Homuth et al., 2015). For this reason, we considered it appropriate to apply this methodology to the Upper Devonian carbonates in Alberta. Our results show the limits of this method. Although we could identify several similarities between the outcrops and the reservoir core samples, the outcrops are no valid proxies for the buried reservoirs in

the Alberta Basin. This implies that, besides many similarities (rock types, thicknesses, depth and deformation, hydrogeological properties) the term 'analogue' needs to be specified.

**Referee 4 – p. 12 line 5:** *" in outcrop conditions, not necessarlily in subsurface conditions"*

**Answer**: Yes that is correct, but here we meant it in a different context. According to several transfer models (Allen and Allen, 1990; Pape et al., 1999, etc.) porosity and permeability generally decrease with increasing depth.

Permeability of the outcrop samples is already significantly lower than permeability of the well core samples, even though the results are not corrected for reservoir conditions yet. We will specify this.

**Referee 4 – p.12 line 14:** *"what type of pore space? intergranular, intercrystalline? microporous, vugs, ????"*

**Answer:** On p. 12 line 14 it is written "The highly permeable, extensively dolomitized reef zones represent promising reservoirs for hydrothermal utilization with predominantly convective heat and laminar fluid flow." It is therefore unclear as to what extent this question refers to this sentence.

**Referee 4 – p.12 line 14:** *"why? this is an important conclusion but there is no real demonstration of the heat transport and flow behaviour"*

**Answer**: As mentioned in the sentence before, this conclusion was made according to the classification of the thermofacies concept (Sass and Götz, 2012). The application of the thermofacies concept is useful in an early exploration stage or in cases where detailed reservoir information is not available. This concept allows a first characterization (whether it is a petrothermal, transitional or hydrothermal reservoir), but does not replace further investigation. Due to the high number of well data (several thousand wells) that need to be evaluated for a more detailed assessment, this point is not further elaborated in the manuscript and will be part of the next phase of the project.

This conclusion takes also previous studies into account regarding diagenesis (Machel and Buschkuehle, 2008), fluid flow and hydraulically connected areas (Michael et al., 2003; Rostron et al., 1997; Bachu et al., 2008) as well as the well data from the oil industry (production data).

**Referee 4 – p. 12 line 19:** *"it is not "reservoir scale", it is "plug scale" measurements within reservoir units"*
**Answer**: Accepted.

**Referee 4 – p. 12 line 19:** *"ambiguous"*
**Answer**: Accepted.

**Referee 4 – p.12 line 20:** *"5 % of porosity variation for 3 orders of K permeability: does it suggest micro- fracture"*

**Answer**: We identified hairline fractures e.g. in well 5-22 (Leduc Formation, central basin), which has also been observed in the Strachan pool close to the Rocky Mountains (Marquez and Mountjoy, 1996). These fractures form networks throughout the rock matrix and might explain higher permeability values.

5 **Referee 4 – p. 12 line 28:** *"what do you mean by "apparent" K?"*
**Answer**: See answer below.

**Referee 4 – p.12 line 31:** *" "intrinsic" K ?"*
**Answer**:

10 After Languth and Voigt (2004) the permeability describes the hydraulic parameters of an aquifer. When different fluids (oil, gas, water) occur in a reservoir, the specific, absolute or intrinsic permeability becomes important. The intrinsic permeability describes the aquifer matrix only and does not consider fluid specifications.

Permeability was measured with a column permeameter and a mini permeameter, which is a variation of the column permeameter (Hornung and Aigner, 2004). Both devices are gas driven, which allows quick, contaminant-free and non-

15 destructive measurements (Filomena et al., 2014). The column permeameter measures the intrinsic permeability after Klinkenberg (1941). Thereby the intrinsic permeability corresponds to the effective gas permeability of air under infinitely high pressure. The calculation is based on Darcy's law, supplemented by the addition of compressibility and viscosity of gases. As it's not possible to determine the permeability under infinitely high pressure (Jaritz, 1999), the column permeameter measures the apparent gas permeability of air with at least five pressure stages

20 from 1000 to 5000 mbar. Afterwards the intrinsic permeability is calculated from the apparent permeability using the Klinkenberg method (1941).

**Referee 4 – p.13 line 7:** *"a missing topic in this interesting chapter is the "upscaling-donwscaling" of the properties (reservoir and geothermal).*

25 **Answer**: Accepted. See Answer above.

**Referee 4 – p.13 line 7:** *"there are no measurements or estimation of properties at greater scale than the plugs; so all the extrapolation made from plug measurements to larger scale are implicitely based on averaging and assuming some level of "stationnarity" within stratigraphic units or sedimentary units. Another implicit assumption is that rock properties statistics*

30 *are not dependent of sample size (no "support effect"). Certainly not the case for permeability in carbonate reservoirs! All these assumptions should be discussed prior to the conclusive statement on the geothermal potential and extrapolation of core plug data."*

**Answer**: Accepted. Porosity and permeability is highly variable within the reservoir. In our manuscript we focused on the core sample scale. We will consider this point.

**Referee 4 – p.13 line 7:** *" you might add histograms of diffrent properties for the various stratigraphic units and facies, to help the discussion on upscaling"*

**Answer**: Thank you very much for this kind advice. We will consider this point during revision of the manuscript.

Histograms of the different rock properties might be helpful regarding the upscaling discussion, but they do not replace the plots presented in this manuscript. Figure 7 shows the correlation of the rock properties and Figures 8 to 10 give a first impression how the properties vary within the reservoir, whereby we focused on the latter.

**Referee 4 – p.28:** *"I would add some histograms, for different geologicl units and facies , etc...that would help to discuss the upscaling issue in the conclusion"*

**Answer**: Same answer as above.

**References**

Allen, P. A. and Allen, J. R.. Basin Analysis, Principles and Applications, Blackwell Scientific, Oxford, London, Edinburgh, Boston, Melbourne, x + 451 pp.,1990.

[revised manuscript text omitted]

---

## Author Comment (AC5) · 11 Mar 2018

5   Leandra M. Weydt[1], Claus-Dieter J. Heldmann[1], Hans G. Machel[2], Ingo Sass[1,3]

[1]Department of Geothermal Science and Technology, Technische Universität Darmstadt, Schnittspahnstraße 9, 64287 Darmstadt, Germany
[2]Earth and Atmospheric Sciences, University of Alberta, Edmonton, AB T6G 2E3, Canada
[3] Darmstadt Graduate School of Excellence Energy Science and Engineering, Jovanka-Bontschits-Straße 2, 64287
10  Darmstadt, Germany

*Correspondence to*: Leandra M. Weydt (weydt@geo.tu-darmstadt.de)

**Author's comment on "Referee comment 3 – review" by Anonymous Referee 3**

For the numbered comments of Referee #3 (R3) see our responses below. There are some helpful detail corrections. Referee
15  #3 articulates two major concerns which are general but lacking argumentation to be comprehensible. – The first of these points is the alleged "lack of reference to previous work"; the second point is that R3 requires "more regional information" and alleges "false claims" and "outraging" political statements. We note that the tone of some of the comments by this referee is uncalled for, in parts offensive and/or unprofessional.

20  **Referee 3 – C1 line 9**:*" Most important is a shocking lack of reference to previous work, including numerous papers on the Hinton Geothermal Potential and numerous papers of the hydrogeological properties of the same aquifers studied."*
**Answer:** There is no explanation where the opinion of "shocking" comes from, nor which references he/she has in mind. The only possible aspects relating to this point (the list points with numbers are dealt below) are two demands:
25  *"[...] The authors need to do [...] some basic background research to place their work in context [...]"* and *"[...] There is a surprising lack of reference to numerous previous geothermal studies in the Hinton area, including some on the same reef systems. As well, since these units also produced major oil fields there has been extensive research conducted on the hydrogeology. A simple web search for terms like 'Hinton geothermal' or 'Nisku hydrogeology' will provide the authors with numerous papers of relevance that should be cited. Rather than 'superficially' I would say the area has been extensively
30  studied. Try a bit of background research as part of your study! [...]"*

Our on-line literature research, as conducted during the writing process of the manuscript, scored 1720 hits. As is common practice, we cited but the most pertinent ones of these. Below [1-20] are the first 20 hits from GOOGLE Scholar to "Hinton geothermal" with our comments. Most of these papers are already known to us and are either cited directly by us or are included within a cited reference. Others of this list are not relevant at all for this area. Hence, this comment by Referee 3 has no basis in fact.

**Referee 3 – C1 line 12:**"*As it stands this works is completely out of context of earlier work and its (sic) not clear that it adds much new information to what is already known.*"

**Answer:**

Our reply to this comment by Referee 3 is the same as the concluding remark to the previous one: his/her comment has no basis in fact.

- This work was triggered by previous studies which suggest that at least some of the Upper Devonian carbonate aquifers are suitable for geothermal utilization (Weides and Majorowicz, 2014; Weides et al., 2013, Lam and Jones, 1985). First 3D models were created using data from several thousand wells (Weides et al., 2013, Ardakani and Schmitt, 2016) to assess the geothermal potential of the Upper Devonian carbonates and further formations, but according to the authors, further research is necessary to evaluate the geothermal potential on a smaller scale. Furthermore, previous studies provided very few hard data on the geothermal and petrophysical reservoir properties of the rocks. The aim of this study is to investigate the Upper Devonian carbonates on a more regional scale and to provide an initial data set of rock properties which are relevant to geothermal exploration and modeling. This is presented in this paper.

- The Upper Jurassic Malm-Aquifer in the Southern German Molasse Basin has already been proven to be suitable as a geothermal reservoir (Birner et al., 2012; Böhm et al, 2013; Homuth et al., 2015; Wolfgramm et al., 2017). Despite their different ages, the Upper Devonian aquifer systems in Alberta and the Upper Jurassic Malm-Aquifer in Germany show several similarities regarding rock types, thicknesses, depth and deformation and hydrogeological properties. Therefore, the exchange of knowledge regarding reservoir exploration would be very valuable. This is the aim of the MalVonian project, which is described in the introduction of this manuscript. The work presented in this manuscript represents the initial phase of the MalVonian project.

- The outcrop analogue concept was successfully applied in the Southern German Molasse Basin to assess the geothermal potential of the Malm-Aquifer (Homuth et al., 2015). For this reason, we considered it appropriate to apply this methodology to the Upper Devonian carbonates in Alberta. With respect to geothermal purposes, this was not realized before in the Alberta Basin. Furthermore, the outcrops presented in this manuscript have not been reported yet (except for outcrop Nigel Peak). Internal industrial reports are usually not publicly available and therefore cannot be taken into account at this point. To correlate the outcrops with stratigraphically equivalent

formations in the reservoir, we analyzed core samples from seven wells (Leduc and Nisku Formation). The Devonian succession in the reservoir is described in detail in chapter 3.

- Thermal conductivity measurements presented in Beach et al. (1987) were measured on several hundred plugs of all sediment rocks in the Hinton-Edson area – not only on the Upper Devonian carbonates. In this paper there is no information given about the well location, depth level or formation of the analysed plugs. Thermal conductivity is given as average values for 13 different lithologies in the basin (limestones, dolomite (better dolostones), shales, sandstones …). Therefore this data set provides a more general overview about thermal conductivity in the Hinton-Edson area. The data set presented in this manuscript provides thermal conductivity values specific for the Leduc Formation of the SCCC and RMRT as well as for three members of the Nisku Formation. The data set gives a first impression how the rock properties change within the reservoir (e.g. Nisku Formation).

**Referee 3 – C2 line 5:**" 1) *Introduction: the reason why Alberta has such a high per capita CO2 emission is that it is developing the worlds [sic] second largest oil field, but with a very small population base (approximately 1o% of that of Saudi Arabia that is developing the worlds [sic] largest oil field). Most of Alberta oil is exported to the US, so its [sic] questionable CO2 accounting to log it all against the producer rather than the consumer. Therefore the introduction provides some misleading statistics that have questionable value in a science paper.*"

**Answer:** The $CO_2$ emissions per capita are a matter of public record and have been used for decades in all political and economic discussion of relevance, most recently by the current Federal Government and the IPCC. There is nothing 'misleading' in using them the way we did. Rather, the comment by R3 appears to be tainted by the opinions of the petroleum industry and/or the community of climate change deniers.

**Referee 3 – C2 line 11:**" 2) *Introduction: The reason for such small hydro usage in Alberta is that southern Alberta is the driest part of Canada – there are very limited opportunities for hydro in the province. But as a nation Canada has over 70% of power production by renewable energy, one of the cleanest grids in the world. So again, the intro has very misleading information.*"

**Answer:** Our study is about Alberta, not about Canada as a whole. The considerable amount of hydro-electric power generation in eastern Canada is irrelevant to our study.

**Referee 3 – C2 line 15:**" 3) *Introduction: To suggest that there is a political climate that favours business over environment, and its [sic] doubtful if Alberta will want to transition to a cleaner energy system is simply outrageous – politics does not belong in a science paper. Besides, Albertans have recently elected a government that has one of the most aggressive environmental programs in North America, including implementing the largest carbon tax in Canada. Such statements that speak to the politics of a place the authors do not live in, and to speculate about future decisions Albertan's will make, have absolutely no place in a science paper.*"

**Answer:**  Climate change is science and policy. Geothermal energy research deals with it. The political background was triggering this research. It is a question of taste if a political statement will belong to a science paper or not. That may stay with R3.

Furthermore, to suggest that authors are entitled to comment on scientific matters with a political dimension only if they live in the area of interest smacks of racism. Besides, one of us – Prof. Machel – has been a resident of Alberta for more than 30 years. At this point we are compelled to conclude that R3 is biased to the point of having disqualified himself as a reviewer of our work.

**Referee 3 – C2 line 22:**" *4) The authors make a false claim that there is no geothermal utilisation. For direct heat use Canada is about 7th in the world (see summary by Raymond 2015). Also, within Alberta the Leduc reef is already being used for direct heat. There are also cleaver thermal storage systems in southern Alberta as well as direct heat use of waters produced from the Western Canada Sedimentary Basin in Saskatchewan, as well as planned drilling this year for electrical production wells. The authors clearly need to do more research on the state of geothermal usage in their study area.*"

**Answer:** Several errors caused by misreading. Talking about "no geothermal energy utilization" (p 2, ln 27), we are discussing hydro and wind "power plants" (introduced in p 2, ln 23) "in the province" (Alberta) (p 2, ln 26f); In this sentence we are neither discussing "direct heat", nor "Canada" in general, nor "Saskatchewan".

**Referee 3 – C2 line 29:**" *5) Page 3, line 9: what is Malm?*"

**Answer:** Malm" is mentioned 5 times in the MS: 3times referred to as "Upper Jurassic Malm-Aquifer […] in Southern Germany", the other times as "German Malm Formation" and "Malm and Devonian" (p 3, ln 9). – Despite the given explanations and the hint in the immediately following line – (it's a) "regionally extensive carbonate aquifer system(s)" (p 3, ln 10f) – we will make some changes to make it absolutely clear to everybody who will not read to that line: "" "Malm Formation and Devonian Period

**Referee 3 – C2 line 29:**" *6) Page 3, C2 line 10. This gets very confusing,. Are you meaning that the overall project studies 3 aquifers or this paper? On line 12 you say there are only two aquifers in Alberta you study which is the subject of this paper. Its also not clear what is meant by two of four? What four?*"

**Answer:** These lines should be read as they are written. To make this clear to all:

- The MalVonian project focuses on the geothermal assessment of three regionally extensive carbonate aquifer systems: the Upper Jurassic Malm-Aquifer in Southern Germany and two Upper Devonian carbonate complexes in Alberta – the Southesk-Cairn Carbonate Complex (SCCC) and the Rimbey-Meadowbrook Reef Trend (RMRT).
- The work presented in this manuscript represents the first phase of the MalVonian project, starting with the investigation of the Upper Devonian carbonate complexes in Alberta (SCCC and RMRT = two aquifer systems).

- The Upper Devonian succession in the Alberta Basin comprises four aquifer systems named D1 to D4 as shown in Fig. 3 and 4. In this work, we focused on the Leduc (D3) and Nisku (D2) aquifer system. This makes two out of four.

**Referee 3 – C3 line 4:**" *7) Page 3, ln 17: This discussion on German aquifers seems out of place in a paper on Alberta, what is the relevance? I would remove this section*."

**Answer:** Hard review shall settle on a reading and understanding of the whole text. We doubt that R3 was doing this. In the introduction we made a very clear statement that our research is dealing with a comparison of the carbonate reservoir systems in Alberta and in the Northern Alps on a large scale. This is what the paper is about. There is no sense in removing this section, because it is relevant to understand the chosen combination of methods.

**Referee 3 – C3 line 5:**" *8) Page 3, ln 23: It seems very odd to introduce looking at outcrops to understand the subsurface as some kind of new approach. Geologists in the petroleum and mining industry have been doing this pretty much ever since geology was invented – its pretty much the very foundation of geology in fact. This is nothing new and certainly not an original idea of Homuthetal2015.*"

**Answer:** We did not claim that outcrop analogue studies are a new approach. It is new in the context described above. In Alberta there are no papers published yet reporting such an approach. It is not of any scientific value to hint on reports oil industry may have in their tresors.

**Referee 3 – C3 line 10:** "*9)Page4,ln2: Delete 'literally' and also, use the Canadian spelling of "Centre" not 'Center' as that is the formal spelling of the Core Centre.*"

**Answer:** Agreed.

**Referee 3 – C3 line 11:**" *10) Page 4, ln 3 ".., results of drill stem test.."*

**Answer:** Agreed.

**Referee 3 – C3 line 12:**" *11) Page 4, ln 6: There is a surprising lack of reference to numerous previous geothermal studies in the Hinton area, including some on the same reef systems. As well, since these units also produced major oil fields there has been extensive research conducted on the hydrogeology. A simple web search for terms like 'Hinton geothermal' or 'Nisku hydrogeology' will provide the authors with numerous papers of relevance that should be cited. Rather than 'superficially' I would say the area has been extensively studied. Try a bit of background research as part of your study!*"

**Answer:** See answer above.

**Referee 3 – C3 line 19:**" *12) Page 4, ln 9: here you say 3 aquifers and above it was two?*"

**Answer:** See answer above.

**Referee 3 – C3 line 19:**" *13) Page 4, ln 21: the formal name is 'Rocky Mountains'*"
**Answer:** Accepted.

**Referee 3 – C3 line 20:** "*14) Page 4, ln 25: closer to 2 million.*"
**Answer:** (Dec 21) Oct 1, 2017: "Half of population" literally is (4306039 / 2) = 2.153 million (www.finance.alberta.ca or http://www.finance.alberta.ca/aboutalberta/osi/demographics/Population-Estimates/index.html). R3 again is dealing with invalid data.

10

**Referee 3 – C3 line 21:**" *15) Page 5, ;n 32: how does this reef trend relate to the Leduc ? need more clear descriptions of everything, same for page 6, ln 9 and 26. Lots of various terms used with no clear description what they all are.*"
**Answer:** It is not clear, what "[…] need (sic) more clear descriptions of everything." implies exactly. This is a listing which geological groups and members built the aquifer according to the given references. For more detail a reader of the paper shall
15   consult the literature cited herein.

**Referee 3 – C3 line 23:**" *16) Page 9, ln 11: need ref for timing of larimide*"
**Answer:** Accepted.

20   **Referee 3 – C3 line 24:**" *17) Page 9, line 13: if there are similar fractures in both, isn't I more reasonable that they have the same origin, rather than invoking two different ones?*"
**Answer:** In outcrop and thin section two fracture types were identified. They differ in scale (microns versus centimeters). The larger ones are crosscutting the smaller ones. There is no further structural analysis in this paper. We reject this comment.

25

**Referee 3 – C3 line 25:**" *18) Section 5.2 this is not petrography*"
**Answer:** What else?

30

**List of References**

[1] Lam, H. L., Jones, F. W .and Lambert, C.: Geothermal gradients in the Hinton area of west-central Alberta, Canadian Journal of Earth Sciences, 19, 755-766, 1982.

[2] Lam, H. L., and Jones, F.W: Geothermal gradients of Alberta in western Canada, Geothermics, 13, 181-192, 1984.

[3] Lam, H-L., and Jones, F. W.: Geothermal energy potential in the Hinton–Edson area of west-central Alberta, Canadian Journal of Earth Sciences, 22, 369-383, 1985.

[4] Kushigbor,C., Lam, HL., Majorowicz, J.A, and Rahman, M.: Estimates of terrestrial thermal gradients and heat flow variations with depth in the Hinton-Edson area of the Alberta basin derived from petroleum bottom-hole temperature data, Geophysical prospecting, 32, 1111-1130, 1984.

[5] Lam, H. L., and Jones, F. W.: A statistical analysis of bottom-hole temperature data in the Hinton area of west-central Alberta, Tectonophysics, 103, 273-281, 1984.

[6] Adams, M. C., Moore, J. N., Fabry, L. G. and Ahn, J. H..: Thermal stabilities of aromatic acids as geothermal tracers, Geothermics, 21, 323-339, 1992.

[7] Nieuwenhuis, G., Lengyel, T., Majorowicz, J., grobe, M., Rostron, B., Unsworth, M. J. and Weides, S.: Regional-scale geothermal exploration using heterogeneous industrial temperature data; a case study from the Western Canadian Sedimentary Basin, Proceedings of the World Geothermal Congress, 2015.

[8] Majorowicz, J. A., Jones, F. W., and Lain, H. L.: A comment on 'A magnetovariational study of a geothermal anomaly'by MR Ingham, DK Bingham and DI Gough, Geophysical Journal International, 76, 667-672, 1984.

[9] Majorowicz, J., Jones, F., Lam, H., and Jessop, A.: Terrestrial heat flow and geothermal gradients in relation to hydrodynamics in the Alberta Basin, Canada, Journal of geodynamics, 4, 265-283, 1985.

[10] Lam, H. L., and Jones, F. W.: AN ASSESSMENT OF THE LOW—GRADE GEOTHERMAL POTENTIAL IN A FOOTHILLS AREA OF WEST—CENTRAL ALBERTA. Energy Developments: New Forms, Renewables, Conservation: Proceedings of ENERGEX'84, The Global Energy Forum, Regina, Saskatchewan, Canada, May 14-19, 1984. Elsevier, 2013.

[11] Kalkreuth, W., and McMechan, M.: Coal rank and burial history of Cretaceous–Tertiary strata in the Grande Cache and Hinton areas, Alberta, Canada: implications for fossil fuel exploration, Canadian Journal of Earth Sciences, 33, 938-957, 1996.

[12] Kalvey, A. R., and Jones, F.W.: Magnetotelluric measurements in an area of west-central Alberta where deep electrical conductivity and basin sediment geothermal anomalies coincide, Journal of applied geophysics, 34, 35-40, 1995.

[13] Adams, M. C., and Davis, J.: Kinetics of fluorescein decay and its application as a geothermal tracer, Geothermics, 20, 53-66, 1991.

[14] Jones, F. W.: A Preparatory Study for Application of Geothermal Energy in the Hinton/Edson Area of Alberta, University of Alberta, Department of Physics, 1983.

[15] Majorowicz, J. A.: Interactive comment on "From oil field to geothermal reservoir: First assessment for geothermal utilization of two regionally extensive Devonian carbonate aquifers in Alberta, Canada by Leandra M. Weydt et al., 2017.

[16] Weides, S., and Majorowicz, J. A.: Implications of spatial variability in heat flow for geothermal resource evaluation in large foreland basins: the case of the Western Canada Sedimentary Basin, Energies, 7, 2573-2594, 2014.

5  [17] Jones, F. W., Lam, H-L., and Majorowicz, J. A.: Temperature distributions at the Paleozoic and Precambrian surfaces and their implications for geothermal energy recovery in Alberta, Canadian Journal of Earth Sciences, 22, 1774-1780, 1985.

[18] Lam, H. L., and Jones, F. W.: An investigation of the potential for geothermal energy recovery in the Calgary area in southern Alberta, Canada, using petroleum exploration data, Geophysics, 51, 1661-1670, 1986.

[19] Majorowicz, J. A., and Moore, M.: The feasibility and potential of geothermal heat in the deep Alberta foreland basin-

10  Canada for CO 2 savings, Renewable Energy, 66, 541-549, 2014.

[20] Jones, F. W., Majorowicz, J. A. and Dietrich, J.: The geothermal regime of the northern Yukon and Mackenzie delta regions of northwest Canada—studies of two regional profiles, pure and applied geophysics, 127, 641-658, 1988.

**Further References**

15  Ardakani, E. P., and Schmitt, D. R.: Geothermal energy potential of sedimentary formations in the Athabasca region, northeast Alberta, Canada, Interpretation, 4, 19–33, doi: 10.1190/INT-2016-0031.1, 2016.

Beach, R, D. W., Jones, F. W. and  Majorowicz, J. A.: HEAT FLOW AND HEAT GENERATION ESTIMATES FOR THE CHURCHILL BASEMENT OF THE WESTERN CANADIAN BASIN IN ALBERTA, CANADA, Geothermics, 16, 1-16, 1987.

20  Birner, J., Fritzer, T., Jodocy, M., Savvatis, A., Schneider, M. and Stober, I.: Hydraulische Eigenschaften des Malmaquifers im Süddeutschen Molassebecken und ihre Bedeutung für die geothermische Erschließung, Z. geologische Wissenschaften, 40, 133–156, 2012.

Böhm, F., Savvatis, A., Steiner, U., Schneider, M., and Koch, R.: Lithofazielle Reservoircharakterisierung zur geothermischen Nutzung des Malm im Großraum München, Grundwasser, 18, 3–13, doi: 10.1007/s00767-012-0202-4,

25  2013.

Homuth, S., Götz, A. E., and Sass, I.: Reservoir characterization of the Upper Jurassic geothermal target formations (Molasse Basin, Germany): role of thermofacies as exploration tool, Geoth. Energ. Sci, 3, 41–49, doi: 10.5194/gtes-3-41-2015, 2015.

Weides, S., Moeck, I., Majorowicz, J., Palombi, D., and Grobe, M.: Geothermal exploration of Paleozoic formations in

30  Central Alberta, Can. J. Earth Sci., 50, 519–534, https://doi.org/10.1139/cjes-2012-0137, 2013.

Wolfgramm, M.: Fazies, Diagenese und hydraulische Durchlässigkeit von Karbonaten des Oberjura – neue Erkenntnisse aus dem F&E-Projekt „Malmfazies", Workshop: „Wissenstransfer der Geothermie-Allianz Bayern, Geothermie in der süddeutschen Moalsse", 26.01.2017, TU München, Munich, Germany, 2017.

---

## Author Response (AR1)

| Page, Line | Comments from Referees | Author's response | Author's changes in the manuscript |
|---|---|---|---|
| **Referee 1 – C2 line 1** | *"However, besides the thermo- and petrophysical properties, the heat flow of the basin has to be taken into account to assess the economic feasibility. In the conclusions I would like to see this point addressed – alternatively the authors might add an outlook where they list the next steps to localize the most promising area with regard to depth and temperature including the available information from published data. Eventually, it is the economically recoverable heat (ERH) which increases the feasibility and this should be clearly stated at the end of the paper."* | We agree with Referee 1 that heat flow is an important parameter and should be taken into account during the assessment of the economic feasibility of a geothermal reservoir. This study was intended to create an initial data set of Upper Devonian carbonate rock properties relevant to geothermal modelling. Statements about economic feasibility were not planned for this early stage of the project, however, a short section will be added to the chapter "discussion and conclusions" summarizing the most important parameters necessary to assess the economic feasibility of this reservoir. Outlook: To complement the data set presented in this study, measurement of further parameters (e. g. thermal diffusivity, specific heat capacity and ultrasonic wave velocity) on well core samples has been planned. Well data provided in the AccuMap or GeoScout databases will be evaluated, interpreted and probably mapped to identify the most promising areas for geothermal utilization in the reservoir. Due to the high amount of well data, this will be only possible on a regional scale. The construction of a regional geological 3D model, using already existing data from previous studies (Majorowicz et al., 2012; Nieuwenhuis et al, 2015), is planned for the most promising areas. | We added a new section to line 5 on page 15: " The classifications by Sass and Götz (2012) and Bär et al. (2011) were applied on the rock properties for a first assessment. The application of the thermofacies concept (Sass and Götz, 2012) is useful in an early exploration stage or in cases where detailed reservoir information is not available but it does not replace further exploration. Likewise, the threshold values applied in Bär et al. (2011) were defined for a low enthalpy region in Germany and are used as indicators, which will likely need to be redefined for the Upper Devonian aquifer systems with increasing experience of geothermal exploitation in this specific reservoir. Rock property measurements provide matrix properties only (mesoscale). Furthermore, they need to be corrected for reservoir conditions and transferred to reservoir scale (up- and downscaling). For a more reliable geothermal assessment, further reservoir parameters (macroscale) like reservoir temperature, flow rates, heat flow, potentiometric surfaces, TDS and $H_2S$ content etc. are needed. Therefore it is necessary to evaluate well data provided in the AccuMap or GeoScout database. These databases comprise an enormous amount of well data from various companies from the last seven decades. However, there is no information given about the quality of the data. In particular, temperature data were identified as inaccurate (Weides and Majorowicz, 2014). Furthermore, heat flow values have been corrected for paleoclimatic surface temperature forcing (Majorowicz et al., 2012). As the Leduc Formation is relatively homogenous within the Alberta Basin, reservoir temperature is one of the crucial parameters. An important point will be to identify 'hot spots' at an economical depth. Likewise, flow rates are very variable at a local scale (Lam and Jones, 1985) and need to be analyzed to evaluate the economic geothermal potential. Furthermore, the aquifer systems need to be operated as transitional systems which need stimulation. This point needs to be considered during cost calculation. Most natural geothermal systems need stimulation for economic and technical reasons (Sass and Götz, 2012). According to the AccuMap database, reservoir stimulation was also a common process for increasing the productivity during oil and gas production. Previous studies (Hofmann et al., 2014) indicate that stimulation treatments can increase the economic feasibility for geothermal utilization in the WCSB for at least some reservoir formations. |

| | | | A limiting factor for geothermal utilization in the study area is the complex hydrogeology. Both formations (Leduc and Nisku) contain highly concentrated water with average TDS values of approximately 200 g l$^{-1}$ (Rostron et al., 1997; Michael et al., 2003). Within the SCCC, salinity increases with increasing distance from the Rocky Mountains ('squeegee flow', Buschkuehle and Machel, 2002; Machel and Buschkuehle, 2008). This parameter can be also very variable at a local scale. In the Hinton-Edson area, salinity ranges from less than 50 g l$^{-1}$ up to 180 g l$^{-1}$ (Lam and Jones, 1985). Likewise, the Nisku Reef Trend is well known for its (over pressured) sour gas pools (Bachu et al., 2008). Problems like scaling and corrosion during operation can lead to higher production costs or in the worst case scenario to the abandonment of the well. These problems are not solved in the geothermal industry yet but should be addressed in the current discussion about a geothermal pilot project in the WCSB. For this reasons a close cooperation with the oil industry is important. Using the experience in operating this reservoir, the existing equipment and infrastructure likely reduces exploration risks and costs for a geothermal project which could provide new business strategies. For the next phase of the project, further parameters (e.g. heat capacity, thermal diffusivity and ultrasonic wave velocity) will be measured on well core samples to complement the data set provided in this study. It is also considerable to extent the study area or include further formations to cover areas with higher heat flow or temperatures. Well data provided in the AccuMap or GeoScout databases will be evaluated, interpreted and probably mapped to identify the most promising areas for geothermal utilization in the reservoir. Due to the high amount of well data, this will be only possible on a regional scale. The construction of a regional geological 3D model, including already existing data from previous studies (Majorowicz et al., 2012; Nieuwehuis et al., 2015), is planned for the most promising areas. |
|---|---|---|---|
| **Referee 1 – C2 line 7** | *"The dataset clearly shows that the aquifer systems under discussion have to be operated as transitional system and thus need stimulation. Again, a point to be considered for economic operation."* | Agreed**.** This point needs to be considered during cost calculation and for economic operation. We will add this point in chapter 6 "Discussion and conclusions". | See changes above. |

| | | | |
|---|---|---|---|
| **Referee 1 – C2 line 12** | ”*Please, check the reference list for consistency (also fonts).*” | Thank you very much for the detailed proofreading. The reference list was checked and corrected accordingly. | The reference list was checked and corrected. |
| **Referee 1 – C2 line 12** | ”*Can you please add the coordinates of the outcrop and well locations in Table 1.*” | A list of coordinates of the outcrops and well locations was added to Appendix B. | A list of coordinates of the outcrops and well locations was added to Appendix B. "The exact coordinates of the wells and outcrops are added to Appendix B." was added to line 19 on page 6 in chapter Material and Methods. |
| **Referee 2 – C2 line 8** | "*Some information on the samples prep. and orientation should be given (saturated or dry; whether or not the thermal conductivities are known to be the vertical (perpendicular thermal conductivity)?*" | Thermal conductivity was measured according to the method of Popov et al. (1999) on dry samples as shown in Fig. 1. Therefore, thermal conductivity of the core samples represents the "horizontal" thermal conductivity. We will add Fig. 1 to the relevant chapter, Material and Methods, in the manuscript. Sample preparation: To minimize the transmission of optical heater radiation into reference standards and rock samples resulting from optical transparent surfaces (Popov et al., 2016), black paint was applied along a scan line on the sample surfaces as well as on the standards. Measurement of the plane surfaces of the core samples in order to estimate the thermal anisotropy was not allowed. | Fig. 1 was added to chapter 'Material and Methods' (=new Fig. 5). The section in line 10 on page 7 was changed to:" For determination of thermal conductivity and thermal diffusivity, a thermal conductivity scanner was used (Popov et al., 10 1999), allowing non-destructive as well as contactless measurements by using infrared sensors. To minimize the transmission of optical heater radiation to reference standards and rock samples resulting from optical transparent surfaces (Popov et al., 2016), black paint was applied along a scan line on the sample surfaces as well as on the standards. Both parameters were measured three to four times on each plug (see Fig. 5). The measurement accuracy is 3 % (Lippman and Rauen, 2009)." We also changed the section in line 27 on the same page to: "The study included further core analyses and measurements of thermal conductivity and permeability on core samples 5 cm to 70 cm long. Thermal conductivity was determined on the mantle surface at dry conditions in the same procedure described for the outcrop analogue samples (as shown in Fig. 5)."  |

| | | | |
|---|---|---|---|
| **Referee 2 – C2 line 11** | *"As to compare above thermal conductivity , porosity new measurements with previously published results, I would recommend reference to Beach et al., Geothermics, Vol. 16, No. I, pp. 1-16, 1987 with averages based on hundreds of thermal conductivity and porosity for carbonates and other rock types from mainly Hinton- Edson area."* | Ok. The data set included in Beach et al. 1987 comprises thermal conductivity values measured on several hundred "water-saturated porous samples" (Beach et al. 1987, p.3). These thermal conductivity values were measured with a divided-bar apparatus and are stated as the vertical or perpendicular thermal conductivity. This data set was not mentioned in the manuscript because it does not contain any information about the origin (well location and depth) of the samples, a reference to the different formations in the basin, nor a detailed rock description. The data set provided in Beach et al. (1987) gives a good overview of thermal conductivity of 13 rock types of the Mesozoic, Cenzcoic and Paleozoic sediments in the Hinton-Edson area. The thermal conductivity values of each rock type represent mean values calculated from different depth levels with varying thickness. The data set gives no information about specific formations and how the properties change within a formation. Therefore, this data set is more useful for large scale observations. In Jones et al. (1984) thermal conductivity in the Hinton-Edson area (most likely the same data set – 936 water saturated samples from 48 wells measured with a divided bar apparatus) is given for four geological formation groups. Jones et al. (1984) states that although a lot of samples were analysed, that some came from very small depth intervals and most of the cores are from very porous and permeable formations. Therefore, this data set is not representative for all relevant parts in the reservoir. The thermal conductivity values presented in this manuscript were measured on dry samples, not on saturated samples. It is to emphasize that thermal conductivity values measured on dry samples as well as thermal conductivity values | We added the following section to line 5 on page 3 (Introduction): "Within the study area, thermal properties measured on core samples exist only for the Hinton-Edson area. Beach et al. (1987) gives a good overview of thermal conductivity of 13 different rock types of the Mesozoic, Cenozoic and Paleozoic sediments in this area of the Alberta Basin but not for specific formations or how the properties might change within a reservoir. We changed line 22 on page 11 to: "Thermal conductivity of the reservoir samples varies between 2.4 W m-1 K-1 (bank-edge reef, well 2-19, Nisku) and 5.5 W m-1 K-1 (reef facies, well 7-33, Nisku). Table 3 provides an overview of average thermal conductivities (arithmetic mean and standard deviation) for each analyzed member and formation. There is no difference between the Leduc Formation of the RMRT and the SCCC. Within the Nisku Formation, the argillaceous limestones of the Lobstick Member show the lowest thermal conductivity, while thermal conductivity of the Zeta Lake Member is within the range of the Leduc Formation." We added "Likewise, the" to 27 on the same page. We added "independent of depth" to line 32 on page 12. We added the following section to line 24 on page 14:"The measured thermal conductivity values presented in this work are within the range of data sets included in previous studies. Some examples are given in Table 4. Further data sets are also listed in Grasby et al. (2012). Measurements of thermal conductivity of whole drill cores have shown that thermal conductivity is independent of depth in the study area. Additionally, thermal conductivity shows no correlation with porosity. Similar findings were made in Jones et al. (1984). With the exception of the Hinton-Edson area, no 'hard' data exists for thermal conductivity in the study area. Beach et al. (1987) and Jones et al. (1984) provide average thermal conductivity values for different lithologies measured on several-hundred water-saturated core samples with a divided bar apparatus mainly taken from wells in the Hinton-Edson area. The data set provided in Beach et al. (1987) represents mean values for different rock types of Mesozoic, Cenzoic and Paleozoic sediments and is more useful for large scale observations. Thermal conductivity given for dolomites is about 1 W m$^{-1}$ K$^{-1}$ lower than that presented in this work (note: not recalculated for water saturated conditions). In Jones et al. (1984), thermal conductivity is given for four geological formation groups. Thermal conductivity given for dolomitic limestones fits very well |

| | | measured on water-saturated samples do not reflect the real conditions in the reservoir and need to be corrected before modelling. | to the results for partially dolomitized limestones shown in Table 3. However, the core samples which were used for the thermal conductivity measurements in the Hinton-Edson area are mainly taken from very porous and permeable zones and, in some cases, from very small depth intervals (Jones et al., 1984). Therefore, they do not represent all relevant parts of the reservoir. The other thermal conductivity values presented in Table 4 are included to demonstrate the variability of this parameter for different facies and/or reservoirs. According to Popov et al. (2016), thermal rock properties are critical parameters for thermo-hydrodynamic models and for predicting the lifetime performance of geothermal systems. Therefore an accurate determination of thermal properties of each relevant formation is necessary." |
|---|---|---|---|

| **Referee 2 – C2 line 25** | *"b) There are many statements related to an assessment of the geothermal energy potential of the carbonate aquifers and reefs in the study area. While porosity, permeanbility , temperature conditions, thermal conductivity ,diffusivity, are important to such evaluation it is not possible to recommend geothermal energy potential without take on other parameters like the hydraulic head, piezometric surfaces, mineralization of aquifer fluids and most important estimate of potential flow rates at well head. In that sense cited by the authors paper by Jones and Lam Can. J. Earth Sci. 1985 went farther and gives such information (see their figs.10-12 and their Appendix figures). I recommend that their results be described, evaluated and briefly discussed in the scope of geothermal energy eval.."* | Ok. As mentioned before, at this early stage of the project, we focused on examining the Upper Devonian carbonates for geothermal purposes and to measure rock properties which are relevant to geothermal exploration and modelling. The classifications by Sass and Götz (2012) and Bär et al. (2011) were used for the initial evaluation of the measured rock properties. We do not claim that they replace further investigation. As mentioned in the manuscript (p. 13, line 22) "rock property measurements produce conservative results and represent matrix properties only" and additional parameters need to be integrated in a geological model for a reliable reservoir prediction. Therefore, our statements are not contradictory to the comments by Referee 2. Due to the high number of well data (several thousand wells) that need to be evaluated for a reliable assessment of the Upper Devonian aquifer systems in the study area, the parameters required by Referee 2 are not included in this manuscript. This will be considered in the next phase of the project. To make this clear, we will add a short section about the most relevant parameters and give an outlook of the next steps according to the hints of Referee 2. | See changes above: R1 C2 line 1. |
|---|---|---|---|
| **Referee 2 – C3 line 8** | *"It is not entirely justified to make statements in the paper like this one: ..."* | Agreed. | This section in line 11 on page 15 was deleted according to the hints of R2. Likewise the sentence in the Abstract on page 2. |

| **Referee 4 – C3 line 20** | "*At their paper The Autors do not address the issue of potential brine production as they do not address parameters needed to estimate it in their paper.*" | Thank you very much for this advice. As mentioned above, this manuscript focuses on rock properties. It was not intended for statements about economic and technical risks during production. Regarding the recent efforts in the Hinton-Edson area to create an initial geothermal project, it is important to consider the complex hydrogeology in these aquifer systems. Both formations (Leduc and Nisku) contain highly concentrated waters with average TDS values of approximately 200 g/l (Rostron et al., 1997; Michael et al., 2003). Likewise, the Nisku Formation is well known for its sour gas pools (Bachu et al., 2008). Problems like scaling and corrosion during operation can lead to higher production costs or, in the worst case scenario, to the abandonment of the well. These problems have not been solved in the geothermal industry yet. The hydrochemistry of the aquifer systems in the study area has been the subject of several previous studies (e.g. Lam and Jones, 1985; Rostron et al., 1997; Buschkuehle and Machel, 2002; Michael et al., 2003, Machel and Buschkuehle, 2008) because it was also the main interest of the oil industry. As Lam and Jones (1985) have showed, these parameters can be very variable on a local scale. In the Hinton-Edson area the salinity ranges from less than 50 g/l up to 180 g/l. | See changes above: R1 C2 line 1. |

| Referee 4 – C1 line 11 | "a missing topic in this interesting chapter is the "upscaling-donwscaling" of the properties (reservoir and geothermal)" | We agree that upscaling and downscaling of the rock properties is essential for reservoir modelling. As far as we did not model in this study, we did not elaborate this point in this manuscript. For the upscaling of the rock properties from plug to reservoir scale, further data should be included, which is beyond the scope of this study. So we agree to give information about "upscaling-downscaling" according to the hint of Referee 4: | To make it clear we did not upscale the properties here. See changes R4 p.13 line 7. |
|---|---|---|---|
| | | -     upscaling in porous media: e.g. Renard et al. (1997), Farmer (2002) | |
| | | -     stratigraphy based upscaling permeability and porosity in carbonate platforms: e. g. Leonide et al. (2012), Borgomano et al. (2013), Brigaud et al. (2014) | |
| | | -     upscaling of thermal conductivities: Rühaak et al. (2015). | |
| | | Reservoir properties of carbonate rocks and their upscaling are related to architecture and facies. Nevertheless, our results show that (pervasive) dolomitization has affected the reservoir parameters of the Western Canadian Sedimentary Basin in an equitable degree. | |
| Referee 4 – C2 line 1 | "Another implicit assumption is that rock properties statistics are not dependent of sample size (no "support effect")." | We do not claim that property statistics are independent from sample size. According to the hint of Referee 4 we will consider this point and add it to the section about upscaling and downscaling. | See changes R4 p.13 line 7. |

| Referee 4 – p. 3 line 21 | *"what are precisely these "properties" and at what scale are they considered? Be more precise here"* | Homuth et al. (2015) analyzed density, porosity, permeability, thermal conductivity, thermal diffusivity and specific heat capacity on plugs from more than 350 samples taken from 19 outcrops in the Swabian and Franconian Alb as well as on core samples and cutting material from wells in the Molasse Basin in order to obtain a large enough data base for upscaling and correlation of the geothermal properties. The rock samples were classified with respect to the dominant reservoir facies types in order to identify facies dependant trends. To transfer the petrophysical properties to reservoir conditions, a Thermo-Triaxial-Cell was used to simulate pressure and temperature conditions of the reservoir. Furthermore, data from previous studies (bio- and lithostratigraphic model of the target formation) and well data were included. | We added the following sentence to page 3 line 21: "The samples were analyzed for thermal- and petrophysical properties like density, porosity, permeability, thermal conductivity, thermal diffusivity as well as specific heat capacity. Furthermore the samples were classified with respect to the dominant reservoir facies types in order to identify facies dependent trends."

For more details about measurements methods the reader should read the paper about the Molasse Basin. This section is already very long and should just give a rough idea why we did this in the Alberta Basin. |
| Referee 4 – p.3 line 24 | *"if you mean "physical-chemical rock properties" it needs to be demonstrated that outcrop and subsurface rocks are analogues (given different burial, structural, and diagenetic history, including telogenesis); if you mean facies, stratigraphic architecture/diemensions, sedimentrry bodies etc...the analogy is probably ok"* | The term "analogue" in this manuscript refers to "facies, stratigraphic architecture, and sedimentary bodies". Our study results are in agreement with your comment, as described later on in chapter "discussion and conclusions" (p 14, line 25ff). | How the term "analogue" is meant in this context is described on page 3 in line 14. |

| Referee 4 – p. 3 line 29 | *"how do you measure thermophysical properties at all these scales?"* | The multi-scale method includes analysis on three different scales 1) macro scale (outcrop, well data) 2) meso scale (rock samples/plugs) and 3) micro scale (thin section, chemical analysis etc.). Macro-scale studies include well data (e.g. hydraulic tests, heat flow) as well as outcrop investigation (facies, structures). It is to emphasize, that not every parameter can be "measured" on every scale. In this manuscript we focused on the meso and micro scale (rock samples and thin sections). | No changes necessary. |
|---|---|---|---|
| Referee 4 – p. 3 line 30 | *"it is not because you are doing multi-scale assessment that the properties or charactéristics are automatically upscales and downscaled! It depends also of the intrinsic heterogeneity of the system at all scales"* | We agree with Referee 4. To make it clear, we will add a short section about upscaling and downscaling in the manuscript. | See changes R4 p. 13 line 7. |

| | | | |
|---|---|---|---|
| **Referee 4 – p.8 line 9** | *"what are the key parameters for this classification? Are they universal or are they dependent of local conditions, basin, etc..."* | This classification especially considers economic and technological factors for geothermal utilization of the Federal state of Hesse in Germany. To assess the rock and reservoir properties in the 3D structural model of the Federal State of Hesse, a multiple criteria approach was used to incorporate their relevance for geothermal systems (Bär et al., 2011, Bär and Sass, 2014). Based on an extensive data base of rock properties and reservoir data, the threshold values (very low to very high) were defined to specify the geothermal potential. Since there exist no such data base for the Alberta Basin (and also no experience with geothermal power/heat generation), this classification was applied to the rock properties presented in this manuscript for a first evaluation. | This section in line 8 on page 8 was changed to give more information: "For an initial evaluation of the geothermal potential (listed in Table 2), the rock properties of the outcrop and core samples (Fig. 8 to 10) were classified into five levels of potential, as previously done in a 3D structural model of the German federal state of Hesse (Bär et al., 2011; Arndt et al., 2011; Bär and Sass, 2014). Within the 3D structural model, volumetric stratigraphic grids (SGrids) were created for each particular unit. After parameterizing the grids based on an extensive database of petro- and thermophysical rock properties for each unit combined with data from more than 4150 wells (e. g. results of pump tests and in-situ temperature measurements), a multi criteria approach was used to incorporate the relevance of the rock and reservoir properties for geothermal systems (Arndt et al, 2011).

Threshold values ranging from 'very low' to 'very high' were defined for every parameter to specify the geothermal potential. These are based on experience in geothermal exploitation in Germany and particularly consider technical and economic factors. For example, the minimum temperature for district heating is defined as 60 °C and 100 °C defines the minimum temperature where electricity productionis technically possible. At temperatures above 120 °C (in combination with production rates above 50 m³/h), electricity production becomes economically intresting. The classification of potential is explained in detail in Arndt et al. (2011). Reservoir properties are not considered here. |
| **Referee 4 – p. 9 line 20** | *"Petrography Outcrops" changed to "Outcrop petrography"* | Accepted. | Line 20 on page 9 was changed to: Outcrop Petrography |

| **Referee 4 – p. 11 line 31** | *"dry samples? It may change when samples are saturated with water?"* | Sentence will be changed to: "… The sample set shows no correlation between thermal conductivity and porosity." According to Popov et al. (2016) thermal conductivity of rocks depends on the mineral composition, interstitial porosity and fluids filling pores and fractures as well as mineral grain size and orientation, anisotropy of the rock matrix and the nature of grain contacts. Thermal conductivity of water is ~ 0.6 W m$^{-1}$ K$^{-1}$, while thermal conductivity of air is approx. 0,025 W m$^{-1}$ K$^{-1}$. Therefore, bulk thermal conductivity generally increases with increasing water content depending on the porosity of the rock sample. Likewise, bulk thermal conductivity of a rock sample generally decreases with increasing porosity. Our data set indicates no correlation between thermal conductivity and porosity. It is most likely, that in this case mineral composition, grain size/geometry and grain contacts have a stronger influence on thermal conductivity than the porosity of the samples (plug scale). | The sentence in line 31 on page 11 was changed to: "The sample set shows no correlation between thermal conductivity and porosity Fig. 7c)." . |
|---|---|---|---|

| | | | |
|---|---|---|---|
| **Referee 4 – p. 12 line 2** | *"that is in contradiction with the statement in the introduction."* | The "thermo facies concept" was explained in chapter 1 "Introduction" by the example of the Southern German Molasse Basin. This concept was successfully applied in the Southern German Molasse Basin to assess the geothermal potential of the Malm-Aquifer (Homuth et al., 2015). For this reason, we considered it appropriate to apply this methodology to the Upper Devonian carbonates in Alberta. Our results show the limits of this method. Although we could identify several similarities between the outcrops and the reservoir core samples, the outcrops are no valid proxies for the buried reservoirs in the Alberta Basin. This implies that, besides many similarities (rock types, thicknesses, depth and deformation, hydrogeological properties) the term 'analogue' needs to be specified. | See changes above R1 C2 line 1. |
| **Referee 4 – p. 12 line 5** | *" in outcrop conditions, not necessarlily in subsurface conditions"* | Yes that is correct, but here we meant it in a different context. According to several transfer models (Allen and Allen, 1990; Pape et al., 1999, etc.) porosity and permeability generally decrease with increasing depth. Permeability of the outcrop samples is already significantly lower than permeability of the well core samples, even though the results are not corrected for reservoir conditions yet. We will specify this. | See changes below: C4 p13 line 7. |

| | | |
|---|---|---|
| **Referee 4 – p.12 line 14** | *"what type of pore space? intergranular, intercrystalline? microporous, vugs, ????"* | On p. 12 line 14 it is written "The highly permeable, extensively dolomitized reef zones represent promising reservoirs for hydrothermal utilization with predominantly convective heat and laminar fluid flow." It is therefore unclear as to what extent this question refers to this sentence. | No changes possible. |
| **Referee 4 – p.12 line 14** | *"why? this is an important conclusion but there is no real demonstration of the heat transport and flow behaviour"* | As mentioned in the sentence before, this conclusion was made according to the classification of the thermofacies concept (Sass and Götz, 2012). The application of the thermofacies concept is useful in an early exploration stage or in cases where detailed reservoir information is not available. This concept allows a first characterization (whether it is a petrothermal, transitional or hydrothermal reservoir), but does not replace further investigation. Due to the high number of well data (several thousand wells) that need to be evaluated for a more detailed assessment, this point is not further elaborated in the manuscript and will be part of the next phase of the project. This conclusion takes also previous studies into account regarding diagenesis (Machel and Buschkuehle, 2008), fluid flow and hydraulically connected areas (Michael et al., 2003; Rostron et al., 1997; Bachu et al., 2008) as well as the well data from the oil industry (production data). | No changes. This is explained in the text. |

| | | | |
|---|---|---|---|
| **Referee 4 – p. 12 line 19** | *"it is not "reservoir scale", it is "plug scale" measurements within reservoir units"* | Accepted. | Line 19 on page 12 was changed to: "Regarding the rock property measurements (core sample scale) of the Leduc Formation (Fig. 9 and 10) at reservoir scale they indicate rather a homogenous matrix (with permeability deviating by three orders of magnitude and porosity varying between 5 to 10 %), in rocks comprised predominantly of stromatoporoid- and coral-rich grainstones/wackestones to floatstones/rudstones. Regarding the rock properties at meter scale, permeability and porosity ..." |
| **Referee 4 – p. 12 line 19** | *"ambiguous"* | Accepted. | See changes above. |
| **Referee 4 – p.12 line 20** | *"5 % of porosity variation for 3 orders of K permeability: does it suggest micro- fracture"* | We identified hairline fractures e.g. in well 5-22 (Leduc Formation, central basin), which has also been observed in the Strachan pool close to the Rocky Mountains (Marquez and Mountjoy, 1996). These fractures form networks throughout the rock matrix and might explain higher permeability values. | No changes. It is described in chapter 5.1 and Discussion and Conclusions. |
| **Referee 4 – p. 12 line 28** | *"what do you mean by "apparent" K?"* | See answer below. | See changes below. |

| | | | |
|---|---|---|---|
| **Referee 4 – p.12 line 31** | " "intrinsic" K ?" | After Languth and Voigt (2004) the permeability describes the hydraulic parameters of an aquifer. When different fluids (oil, gas, water) occur in a reservoir, the specific, absolute or intrinsic permeability becomes important. The intrinsic permeability describes the aquifer matrix only and does not consider fluid specifications. Permeability was measured with a column permeameter and a mini permeameter, which is a variation of the column permeameter (Hornung and Aigner, 2004). Both devices are gas driven, which allows quick, contaminant-free and non-destructive measurements (Filomena et al., 2014). The column permeameter measures the intrinsic permeability after Klinkenberg (1941). Thereby the intrinsic permeability corresponds to the effective gas permeability of air under infinitely high pressure. The calculation is based on Darcy's law, supplemented by the addition of compressibility and viscosity of gases. As it's not possible to determine the permeability under infinitely high pressure (Jaritz, 1999), the column permeameter measures the apparent gas permeability of air with at least five pressure stages from 1000 to 5000 mbar. Afterwards the intrinsic permeability is calculated from the apparent permeability using the Klinkenberg method (1941). | This section in Material and Methods (p. 7, l. 5) was changed to: "Matrix permeability of the outcrop samples was determined with a column permeameter using different air pressure levels from 1 to 3 bar. This method is based on Darcy's law enhanced by factors for compressibility and viscosity of gases (Jaritz, 1999). It allows calculation of the intrinsic permeability (Ki) from apparent permeability (Ka) by using the Klinkenberg method (Klinkenberg, 1941). Thereby, intrinsic permeability describes the aquifer matrix only and does not consider fluid properties (Languth and Voigt, 2004). The intrinsic permeability corresponds to the effective gas permeability of air under infinitely high pressure. As it is not possible to determine permeability under infinitely high pressure, the column permeameter measures the apparent gas permeability of air with at least five pressure stages (Jaritz, 1999). Afterwards, the apparent permeability is plotted in the Klinenberg plot to calculate the intrinsic permeability. Measurement accuracy varies from 5 % for highly permeable rocks (K ≥ 10-14 m²) to 400 % for impermeable rocks (K ≤ 10-16 m²) (Filomena et al., 2014)." |
| **Referee 4 – p.13 line 7** | "a missing topic in this interesting chapter is the "upscaling-donwscaling" of the properties (reservoir and geothermal)." | Accepted. See Answer above. | See changes below. |

| Referee 4 – p.13 line 7 | *"there are no measurements or estimation of properties at greater scale than the plugs; so all the extrapolation made from plug measurements to larger scale are implicitely based on averaging and assuming some level of "stationnarity" within stratigraphic units or sedimentary units. Another implicit assumption is that rock properties statistics are not dependent of sample size (no "support effect"). Certainly not the case for permeability in carbonate reservoirs! All these assumptions should be discussed prior to the conclusive statement on the geothermal potential and extrapolation of core plug data."* | Accepted. Porosity and permeability is highly variable within the reservoir. In our manuscript we focused on the core sample scale. We will consider this point. | There must be a misunderstanding here: We did not upscale any properties here. We correlated existent data with our measurements. At reservoir scale meant: From base to top of the Reservoir and the distribution within the basin. There is a huge data set of porosity and permeability available and combined with core analysis from literature, they indicate that the formation is relatively homogenous within the basin (= the range of porosity and permeability at plug scale, nearly everywhere the same grade of dolomitization). Is is to mention that the AccuMap data base predominantly provides data for high porous reservoir sections. Unfortunately, there is no data available for the analyzed wells of the Lobstick, Dismal Creek and Bigoray Member. Therefore it would not be very representative to start the upscaling with the data set presented in this study. More data is needed and probably own porosity and permeability measurements for formations where no data exists.

As the outcrops we found in the Front Ranges show distinct differences compared to the reservoir core samples, we have decided that the analogue samples are not useful for geothermal modeling. Therefore we did not consider any scale effects. The outcrop analogue plugs have predominantly a size of 40 mm in diameter, but also 25 mm or 64 mm. We could not indicate any significant size dependent errors in the measurement results at this sample size or within this data set. I think the diagenetic overprint is to high and to variable to identify such errors and affects the results the most. This will be also a critical point for the histograms and variograms presented in your work. I agree that bigger samples are always better to reflect the outcrops/reservoir properties and that small samples cannot represent bigger fracture zones or karstification. It is clear that plugs measurements most likely underestimate the outcrop/ reservoir permeability. |
|---|---|---|---|

| Referee 4 – p.13 line 7 | " you might add histograms of diffrent properties for the various stratigraphic units and facies, to help the discussion on upscaling" | Thank you very much for this kind advice. We will consider this point during revision of the manuscript. Histograms of the different rock properties might be helpful regarding the upscaling discussion, but they do not replace the plots presented in this manuscript. Figure 7 shows the correlation of the rock properties and Figures 8 to 10 give a first impression how the properties vary within the reservoir, whereby we focused on the latter. | We decided not to defocus the aim of this paper. As explained above, porosity and permeability data is not available for all analyzed wells. Upscaling with this data set would not be very representative. More data provided in the AccuMap data base needs to be evaluated, which is beyond of the scope of this study.

Therefore we added just a brief summary about the next steps and the processing of the data concerning upscaling:
The here presented data set was analyzed under lab conditions (20 °C) and represents matrix properties only (representative at cm-scale, macroscale). It is to emphasize that rock property measurements can differ with samples size. Measurements at plug scale do not represent larger features as large vugs and molds (bigger than sample size), karstification or fracture zones and most likely underestimate porosity or permeability of the outcrop/reservoir. Especially in carbonate reservoirs porosity and permeability can be very variable.
In order to utilize the rock properties in a geological model, they need to be corrected for reservoir conditions and subsequently transferred to reservoir scale (macroscale). For example thermal conductivity: With given information about reservoir fluid properties and porosity, thermal conductivity can be recalculated for water-saturated conditions (Clauser and Huenges, 1995; Popov et al., 2003). Temperature dependency models are used to transfer thermal properties to reservoir conditions as described in Vosteen and Schellschmidt, (2003) and Sommerton, (1992). In general, thermal conductivity increases with increasing water content, porosity and pressure, but decreases with increasing temperature (Clauser and Huenges, 1995). For example, thermal conductivity of matrix dominated limestones of the Jurassic Malm Formation in Southern Germany ranges between $1.35 - 2.62$ W m$^{-1}$ K$^{-1}$ at 20 °C and dry conditions, $1.60 - 2.79$ W m$^{-1}$ K$^{-1}$ at 20 °C and saturated conditions and $1.41 - 2.25$ W m$^{-1}$ K$^{-1}$ at reservoir conditions (150°C). Geothermal reservoirs have a smaller margin to be economically profitable than oil reservoirs. Therefore Rühaak et al. (2015) tested different upscaling procedures for thermal conductivity to testify their accuracy. Thereby the intention is to keep as much of the small scale information as possible. The results indicate that harmonic and geometric mean upscaled values reflect most accurately local values.
A reliable porosity/permeability prediction is crucial for reservoir characterization and modeling (Borogomano et al., 2008) and mostly has to be carried out with a limited number of core measurements. Upscaling |
|---|---|---|---|

| | | | techniques for porosity, permeability and hydraulic conductivity have been subject of several studies and are described in Clauser (1992), Renard and de Marsily (1997) and Farmer (2002). Previous studies in the WCSB analyzed a high amount of well data and used the calculation of arithmetic and geometric mean values to upscale their parameters from plug to reservoir scale (Weides et al., 2013, Ardakani and Schmitt, 2016). Borgomano et al. (2013) analyzed the porosity-permeability relationship of plug samples combined with detailed facies analysis to identify the predictability of these parameters. Additionally, the creation of vertical and horizontal histograms and variograms can help to identify the heterogeneity, anisotropy and lateral distribution of the properties and whether the upscaling from plug to reservoir scale is linear or not. |
|---|---|---|---|
| **Referee 4 – p.28** | *"I would add some histograms, for different geologicl units and facies , etc...that would help to discuss the upscaling issue in the conclusion"* | Same answer as above. | See changes above. |
| **Jacek Majorowicz – C2 line 7** | "*1.The reference should be Weides and Majorowicz (2014) as given below in the References."* | I apologize for this mistake. It has been corrected accordingly. | It was corrected on page 2, line 32, page 3, line 2 and 4, page 4 , line 7, page 18, line 16 |

| Jacek Majorowicz – C2 line 9 | "*2, Heat flow in the cited map in Weides and Majorowicz (2014, their Fig. 3) for the studied area of the Alberta basin is not reaching 80 mW m-2 It is less than 70mW m2. The heat flow in the WCSB generally ranges from 30 to 100 mW/m2, being 60.4 mW/m2 on average according to Weides and Majorowicz (2014). The heat flow values has been corrected for paleoclimatic surface temperature forcing (Majorowicz et., al. 2012)."* | Thank you very much for this correction. The associated section in chapter "1 Introduction" in page 3 line 1 has been changed to: "The area around the town site of Hinton in the western region of the Alberta Basin (Fig. 1) is of particular interest because well data analysis indicates flow rates of more than 400 $m^3$ $h^{-1}$ and temperatures up to 150 °C at depths of approximately 5 km (Lam and Jones, 1985)." General information about the geothermal gradient and heat flow in the WCSB was added to page 2, line 30: "This appears feasible because, although this province is characterized as a 'low enthalpy region' (Grasby et al., 2012; Lam and Jones, 1985 and 1986) with a moderate average geothermal gradient of 33.2 °C $km^{-1}$ and an average heat flow of 60.4 W $m^{-2}$ in the WCSB, recent studies using data from several tens of thousands of oil and gas wells suggest that at least some of the Upper Devonian carbonate aquifers are suitable for geothermal utilization (Weides and Majorowicz, 2014)". | line 1 on page 3 has been changed to: "The area around the town site of Hinton in the western region of the Alberta Basin (Fig. 1) is of particular interest because well data analysis indicates flow rates of more than 400 $m^3$ h-1 and temperatures up to 150 °C at depths of approximately 5 km (Lam and Jones, 1985).";

page 2, line 30 has been changed to: "This appears feasible because, although this province is characterized as a 'low enthalpy region' (Grasby et al., 2012; Lam and Jones, 1985 and 1986) with a moderate average geothermal gradient of 33.2 °C km-1 and an average heat flow of 60.4 W m-2 in the WCSB, recent studies using data from several tens of thousands of oil and gas wells suggest that at least some of the Upper Devonian carbonate aquifers are suitable for geothermal utilization (Weides and Majorowicz, 2014)." |
|---|---|---|---|

| | | | |
|---|---|---|---|
| **Jacek Majorowicz – C2 line 15** | *:"The attached average geothermal gradient map (Fig.1) shows that there are much 'hotter' areas in the WCSB in Alberta and these are to the north and east of deep part of the foreland basin in the Hinton area."* | This is correct. The two carbonate complexes were selected because  1) A lot of communities in Alberta are located in this area (this applies especially for the Rimbey-Meadowbrook Reef Trend), 2) There is increasing public interest in geothermal energy utilization in the Hinton-Edson area (Southesk-Cairn Carbonate Complex), 3) the general growing interest in repurposing abandoned oil and gas wells to find new possibilities to reduce the demand of fossil fuels and to reduce $CO_2$-emissions, 4) there are similarities in rock type, depth and structure of the Devonian aquifer systems with the Jurassic Malm-aquifer in the Southern German Molasse Basin, which offers the possibility of knowledge transfer between Alberta and Germany. | We added the following sentences to page 4 for a better understanding:

 Page 4 line 20: "Therefore the carbonate platforms are in focus of interest due to the general increasing public interest in repurposing abandoned oil and gas wells."

 Page 4 line 27:" Within the WCSB, the geothermal gradient and heat flow varies from 20 °C km$^{-1}$ to over 55 °C km$^{-1}$ and 30 mW m$^{-2}$ to 100 mW m$^{-2}$, respectively (Majorowicz et al., 2012; Weides and MAjorowicz, 2014). Areas with higher heat flow values are identified in the northern part of the basin. Compared to the study area, these zones are sparsely populated and thus not chosen for this pilot study. |
| **Jacek Majorowicz – C2 line 20** | *"However, such deep wells are expensive and economics of drilling two 5km wells into the deepest sedimentary horizons will end up with extremely high mineralized waters and rather poor porosity/permeability (Lam and Jones, 1985)."* | This pilot study was intended to be an initial inquiry into the the Upper Devonian carbonates with respect to geothermal utilization and to create an initial data set of rock properties relevant to geothermal exploration and modelling. To assess the economic feasibility of this reservoir more data is needed, e.g. the corrected heat flow and temperature data (Majorowicz et al., 2012; Nieuwenhuis et al., 2015), salinity, flow rates, potentiometric surfaces, and hydraulic heads to name just a few. The high TDS content in the formation waters of the Leduc and Nisku formations (Rostron et al., 1997; Bachu et al., 2008) is a critical parameter for geothermal production. Therefore, well data provided in the AccuMap or GeoScout databases need to be evaluated and interpreted carefully, which is beyond of the scope of this study. A | See changes above: R1 C2 line 1. |

| | | short section will be added to the chapter "discussion and conclusions" to provide an overview of further steps. We don't agree with the statement that porosity and permeability are generally poor at these depth levels. According to Amthor et al. (1994), there is an overall decrease of porosity/permeability with depth in the Leduc Formation in the WCSB, but it has also been shown that especially the dolomitized reef sections retain their porosity/permeability compared to limestones at the same depth level or compared to well cores which are located in the shallower parts of the basin. An example is presented in this manuscript: Well 2-36-54-23W5 (>4 km depth), located in the western part of the Southesk-Cairn Carbonate Complex, shows the highest porosity and permeability of all wells in this dataset. Porosity and permeability can be highly variable within aquifers at a local scale. However, it must be taken into account that high porosity and permeability values do not naturally guarantee high flow rates. Therefore, a careful evaluation of the well data and other existing information on the reservoir must be carried out to localize the most promising areas. | |
|---|---|---|---|

| | | | |
|---|---|---|---|
| **Referee 3 – C2 line 15** | 3) *Introduction: To suggest that there is a political climate that favours business over environment, and its [sic] doubtful if Alberta will want to transition to a cleaner energy system is simply outrageous – politics does not belong in a science paper. Besides, Albertans have recently elected a government that has one of the most aggressive environmental programs in North America, including implementing the largest carbon tax in Canada. Such statements that speak to the politics of a place the authors do not live in, and to speculate about future decisions Albertan's will make, have absolutely no place in a science paper."* | Climate change is science and policy. Geothermal energy research deals with it. The political background was triggering this research. It is a question of taste if a political statement will belong to a science paper or not. That may stay with R3. | Line 21 to 24 on page 15 was deleted. |
| **Referee 3 – C2 line 29** | " 5) *Page 3, line 9: what is Malm?*" | Malm" is mentioned 5 times in the MS: 3times referred to as "Upper Jurassic Malm-Aquifer […] in Southern Germany", the other times as "German Malm Formation" and "Malm and Devonian" (p 3, ln 9). – Despite the given explanations and the hint in the immediately following line – (it's a) "regionally extensive carbonate aquifer system(s)" (p 3, ln 10f) – we will make some changes to make it absolutely clear to everybody who will not read to that line: "" "Malm Formation and Devonian Period" | Malm and Devonian was changed to Malm Formation and Devonian Period in line 9 on page 3 |
| **Referee 3 – C3 line 10** | "*9)Page4,ln2: Delete 'literally' and also, use the Canadian spelling of "Centre" not 'Center' as that is the formal spelling of the Core Centre.*" | Agreed. | Literally was deleted in line 2 on page 4, Core Research Center was changed to Core Research Centre in line 3 on page 4 and in line 18 on page 6 |

| **Referee 3 – C3 line 11** | " *10) Page 4, ln 3 ".., results of drill stem test.."* | Agreed. | "results of " was added to line 3 on page 4 |
|---|---|---|---|
| **Referee 3 – C3 line 19** | " *13) Page 4, ln 21: the formal name is 'Rocky Mountains'"* | Accepted. | Rockies was changed to Rocky Mountains in line 21 on page 4 |
| **Referee 3 – C3 line 23** | " *16) Page 9, ln 11: need ref for timing of larimide*" | Accepted. | We changed line 9 on page 9 to: "These fractures probably resulted from overpressuring of the well-sealed reservoir during deep burial, driven by thermal cracking of crude oil to gas and possibly aided by tectonic compression, both happening simultaneously during the Late Cretaceous - Early Tertiary and coinciding with the peak phase of the Laramide orogeny (appr. 80 -55 Ma., English and Johnston, 2004)."

We changed line 19 on page 5 to: "This region of the WCSB has undergone four orogenies since the Devonian period (1. Antler (Devonian-Carboniferous), 2. Sonoma (Late Permian), 3. Columbian (Jurassic-Early Cretaceous) and 4. Laramide (Mid-Late Cretaceous-Tertiary); Machel, 2010), which ultimately resulted in the wedge-shaped, triangular geometry in cross section of the foreland basin and its sedimentary filling (Fig. 2), now generally referred to as the Alberta Basin. |
|  |  |  |  |
| **Further corrections:** |  |  |  |
|  | Table 1: The depth levels of well 2-36 and 16-18 in Table 1 were reversed. The analysed depth interval of well 16-18 is now "2741.00 m to 2779.77 m" and the analysed depth intervals of well 2-36 are "4068.77 m to 4095.00 m" and "4145.00 m to 4165.39 m" as shown in Fig. 9. | The depth intervals were corrected. |  |

| | | |
|---|---|---|
| | Table 2 – Perdrix Formation: The number of measured plugs for the density measurements is N = 17. | The table was corrected. | |
| | Page 6 line 22: It is IHS instead of HIS – also in the references. I apologize for the unfortunate auto correction. | HIS was changed to IHS on page 6 line 22 and in line 13 on page 17. | |
| | Page 8, line 14: The investigated Leduc Formation mainly represents stromatoporoid and coral-rich reefs and reef margin lithologies with dissolution enlarged vugs and molds. | Changed to: "The investigated Leduc Formation mainly represents intensively dolomitized stromatoporoid and coral-rich reefs and reef margin lithologies with dissolution enlarged vugs and molds. | |
| | Page 4 line 8 | To make the aim of this paper clear: Accurate thermal properties are critical parameters for reliable geothermal assessment (Popov et al., 2016). The aim of this work was to create an initial data set of rock properties (relevant to geothermal modeling) specific to the Upper Devonian aquifer systems which have become of particular interest for geothermal utilization and also for identifying variations of rock properties within the reservoir. | |
| | Page 2 line 2 | We changed $10^{-14}$ to $10^{-15}$ to avoid over estimation | |
| | Page 3 line 14 | We added (Fig. 3) for a better understanding. | |
| | Page 4 line 17 | We changed Nisku reef trend to Nisku Reef Trend | |
| | Page 4 line 18 | We changed Both to The | |

| | | |
|---|---|---|
| | Page 6 line 4 | We added (D1 – D4, Fig. 3) for better understanding. |
| | Page 6 line 16 | We changed termophysical to "thermo-" and added "outcrop". |
| | Page 7 line 4 | We added the following explanation: "For direct comparison with the provided data in the AccuMap data base only particle density is presented here". |
| | Page 13 line 21 | We replaced > with "up to", otherwise readers could think that permeability is commonly higher than $10^{-12}$ m² in the reservoir." |
| | Page 13 line 26 | The sentence was deleted, because we added a new section addressing the next steps. |
| | Page 15 line 19 | We added the following sentence to Acknowledgements: "We would like to thank Jean Borgomano, Jacek Majorowicz and three anonymous reviewers for their constructive comments." |
| | Tables; Figures, Appendix B, References | The sections tables, figures have been modified. Appendix B has been added. References has been actualized. |

---

## Author Response (AR2)

| Page, Line | Comments from Referees | Author's response | Author's changes in the manuscript |
|---|---|---|---|
| **Referee 3 – Report 2 Comment 1** | *The authors examine porosity and permeability and make some qualitative statements regarding relevance to geothermal utilisation. But as it stands, the main conclusion in the paper is that there are parts of the Devonian reservoirs that have high permeability and this can be related to dolomitization. This has been known since the first major oil field in Alberta was discovered in one of these reefs in 1947. There has been extensive studies of these aquifer systems and their physical properties for petroleum reservoirs over the last 50 years (including one of the authors who has built a career on this) as well as more recently for potential as CO2 storage. So the very basic observations made by the authors don't add anything new to what has been long known – or they fail to show how they have added any new knowledge. Despite being the stated main goal of the study, there is little done though to assess any geothermal potential based on the observations the do make – the key question is how much energy could be extracted and at what rates. The authors could do this if they went further, and used their data to estimate geothermal resource potential, and then maybe make some new and important contributions.* | This is not exactly what we did in this manuscript. We conducted an outcrop analogue study in order to apply and testify the methods successfully carried out in Germany. Furthermore we analyzed wellbore core samples from 7 wells (~ 530 core meters) and measured thermal conductivity and permeability on the core samples from the base to the top of each core to cover as much as possible of the reservoir. The aim was to identify variations of rock properties within the reservoir (specificly for the Nisku and Leduc Formation) and to create an initial data base for geothermal modeling. I couldn't find equivalent studies dealing with this topic – not for the Upper Devonian aquifer systems. Additionally, we used existent porosity, density and permeability data from the AccuMap data base in order to correlate the existent data with our results and to show the relation between the different rock properties. Compared to the high amount of well data, there are not many core profiles including detailed rock description published.

The diagenetic evolution of the formations under discussion is described in chapter 3 and could be studied in more detail in the cited references. I agree that Hans Machel 'built his career on this' topic, but it would be misleading to cite all his papers. For example Machel (2010) comprises a review about the Upper | We added "Machel, 2010; Kuflevskyi, 2015" to line 28 on page 6 |

Devonian aquifer systems in general and Kuflevskyi (2015) comprises all previous studies, well data and new data of the Rimbey-Meadowbrook Reef Trend. Therefore I don't think it is necessary to describe all ~ 20 diagenetic events again. Both papers are cited several times.

Rock property measurements are crucial for detailed and precise modeling (Popov et al., 2016). Therefore it is necessary to investigate the study area on a local scale, which requires new data. As shown in chapter 6 Table 4, thermal conductivity of sedimentary basins is very variable and can be also very variable within the reservoir. It is very important to characterize the thermal properties as accurate as possible, because they form the basis for further investigations For example heat flow in the study area was calculated from thermal conductivity measurements (Beach et al., 1987) and Hofmann et al. (2013) created a geological model for the Edmonton area assuming thermal conductivity values of about 2,42 W m$^{-}$1K$^{-1}$ for the carbonatic rocks and 1,38 W m$^{-1}$ K$^{-1}$ for shales. Therefore this study provides new data for a more precise modeling (thermal conductivity of the Leduc Formation is about 4 W m$^{-1}$ K$^{-1}$).

Beside the petrological investigations, one finding is that for further studies the dolomitized reef sections might be promising targets for hydrothermal energy utilization.

The fact that the Upper Devonian aquifer systems have been intensively analyzed

| | | by the oil industry is a reason WHY we have chosen this study area. The aim is to transfer new findings, methods, exploration and exploitation tools between Germany and Alberta to push geothermal energy utilization in both countries. This is well described in the introduction. We don't claim to provide a full assessment for geothermal utilization for the Alberta basin starting from geological investigation, over 3D modeling, to the construction of a power plant and to provide solutions for technical issues in one manuscript. Our aim is to create geological models integrating all available data (rock properties, reservoir data etc.), which is beyond of the scope of this study. | |
|---|---|---|---|
| **Comment 2** | *Title – the authors still claim that there work is the 'First assessment' of geothermal utilization of Devonian Aquifers in Alberta. This is clearly untrue and points to an overall problem of the authors not being aware of previous work in their study area. Just to name a few of the many previous studies done before: Lam et al., 1982; 1985; Jones et al., 1985; Bachu et al., 1991; Weides et al., 2012; Grasby et al., 2011; Majorowicz et al., 1981; Gray et al, 2012; Fergusin and Ufondu, 2017.* | The title refers to the project and the applied methods. If this is misleading, the editor should decide whether we should delete 'first' or a new title for the manuscript is needed. Referee 3 claims that we are not aware of previous work in the study area: Lam et al. 1982 "Geothermal gradients in the Hinton area of west central Alberta" used temperature data of more than 3300 wells in the Hinton area in order to estimate thermal gradients. In our manuscript already included: Lam et al. 1985 'Geothermal energy potential in the Hinton-Edson area of west-central Alberta' and Lam et al 1986 'An investigation of the potential for | The following section was added to page 2 line 6: "Previous studies predominantly focused on determination of heat flow, geothermal gradients and reservoir temperature (e.g. Garland and Lennox, 1962; Majorowicz and Jessop, 1981; Lam et al., 1982; and more recent Majorowicz et al. 2012, 2014), while only a few considered water chemistry and recovery (Lam and Jones, 1985, 1986). More recent studies considered parameters like porosity and permeability (e.g. Weides et al., 2013, Weides and Majorowicz, 2014, Ardakani and Schmitt, 2016) or injection and production rates in combination with reservoir tempertatures (Ferguson and Ufondu, 2017)." |

geothermal energy recovery in the Calgary area in southern Alberta'. We preferred to cite the most important and/or newest paper.

Jones et al 1985 is already included in the manuscript.

Bachu 1991'On the effective thermal and hydraulic conductivity of binary heterogeneous sediments' focus on several upscaling methods on the example of the Upper Cretaceous Mannville Group. This study would be relevant in the next step of the project, when it is necessary to upscale the properties to reservoir scale in a geological model.

Weides et al. 2012 'Geothermal exploration of Paleozoic formations in Central Alberta ' is cited in the manuscript, not as the online version from 2012, but as the printed version in Canadian Journal of Earth Science from 2013 as Simon Weides did it in his PhD thesis.

Grasby et al 2011 'Geothermal energy resource potential of Canada' includes nearly the same extensive overview about geothermal research in Alberta like Grasby et al. 2012 'Geothermal energy resource potential of Canada. Open-File report' which is included in the manuscript.

"Majorowicz et al 1981" – It exists only "Majorowicz and Jessop 1981" with the title 'Regional heat flow patterns in the

| | | | |
|---|---|---|---|
| | | western Canadian sedimentary basin' which presents one of the first studies concerning heat flow in the study area. Updated and new findings are included in the more recent works from Majorowicz in Majorowicz et al. 2014 and Majorowicz et al 2012, which are cited in the manuscript. We preferred to cite the current state of knowledge.

Ferguson and Ufondu 2017 focus on the geothermal assessment in the WCSB with focus on injection and production rates as well as estimated reservoir temperatures. It was published after the manuscript had already been written. I agree to mention this study in the manuscript | |
| **Comment 3** | *Page 2, Line 17: The authors continue to make false claims regarding the place of Alberta in global CO2 emissions. The Environment Canada data they reference does not support their claim. They may have miss-read the table of reported emissions rather than the relevant table of total emissions. Environment Canada has clearly recorded that Saskatchewan is the largest per Capita Co2 emitter in Canada for the last decade or so. As well, the reference they cite as support does not make reference to Alberta in a global context. There are clearly jurisdictions within the United States that have higher per capita emissions than any province in Canada. I could not find the data easily, but I am sure there would be jurisdictions within Saudi Arabia or other major oil producing regions of the world that* | It would have been beneficial if Referee 3 would cite the data he mentioned. Furthermore the statement is in the manuscript is written in **past tense** and we don't claim that Alberta has the highest per capita $CO_2$-equivalent emission today. This statement was intended to show the need for new alternatives for energy production.

To avoid further discussions we decided to delete this section. | We changed the beginning of chapter 1:
"Canada currently emits about 730 Mt a$^{-1}$ $CO_2$-equivalent, of which the Province of Alberta emits nearly 300 Mt a$^{-1}$ (Environment Canada, 2016, 2017: last reliable numbers are for 2015). Therefore Alberta belongs to the five provinces in Canada with the highest emission rates (Environment Canada, 2016, 2017). The main reason for this pattern is the industrial generation of energy (electricity and heat) from coal and gas, which currently provide about 40 % each of the energy mix in this province, along with the huge mining operations of the oil sands deposits. The oil sands industry alone currently accounts for about 10 % of Canada's $CO_2$ -equivalent emissions (Canadas Oil Sands, 2017), tendency rising. However, the trend of increasing $CO_2$-emissions could be significantly reduced if alternative and/or renewable energy sources were implemented to a larger degree. For Canada to meet or at least approach the targets of the Paris Accord from 2015 regarding the reduction of $CO_2$-emissions, geothermal energy should become part of the energy mix in Alberta."

We changed the Abstract. "The Canadian Province of Alberta has |

| | | | |
|---|---|---|---|
| | *would also have larger per capita emissions.* | | one of the highest per capita $CO_2$-equivalent emissions in Canada, predominantly due to industrial burning of coal for the generation of electricity and the mining operations in the oil sands deposits." |
| **Comment 4** | *Page 2, Line 24: The authors continue to make inappropriate political comments. How do they know that Alberta has favoured 'business over environment' .are they privy to secret cabinet documents that can support that claim? What does it even mean, does not every political jurisdiction in the world have to make that choice to balance development and environmental protection every day? I've been to Germany, were the authors are from, many times – and I see a heavily industrialised country with very little natural environment left – they have a centuries long history of favouring industrial development over environmental protection. Maybe that's the reason Alberta has such a high rate of tourists from Germany who come to see true nature? So then, what gives the authors the right to speculate that Alberta will not change to be more environmentally concerned – especially as the province has recently made a major political shift to a government that is advancing climate polices. Certainly Germany has not shown any positive trends as they are Europe's largest producer and burner of coal and are the worlds largest producer of lignite, the dirtiest coal there is. Will Germany's 'business over environment' policies every change? Hard to say. This all reminds me of a recent news* | This sentence was already deleted in the last version of the manuscript. Referee 3 should read the manuscript more carefully.

It is not our aim to degrade Alberta and we don't claim that European countries – in this case Germany – are handling environmental problems better. But the fact that renewable energy should be integrated in the energy mix of both countries triggered this study. The study is part of a larger project which focuses on the assessment of carbonatic aquifers for geothermal utilization in GERMANY and Alberta.
The cooperation aims to transfer the knowledge between these two countries to push geothermal utilization in general.

I will not consider the remaining part of R3's comment. | No changes needed. |

| | | | |
|---|---|---|---|
| | *headline I saw titled: "Germany is a coal-burning, gas guzzling climate change hypocrite'. Perhaps this may offend the authors, much as their comments on Alberta politics offends me as an Albertan – in the end though, none of this belongs in a science discussion and I'm perplexed why they insist to keep these inappropriate political comments in a science paper.* | | |
| **Comment 5** | *Page 3, Line 5: This claim of 'few hard data' is false. The authors are only referring to geothermal studies. However the Devonian Reef systems in Alberta hosted major oil and gas pools and as such have been the subject of extensive investigation of there petrophysicsal properties and aspects such as porosity, permeability etc. There are numerous studies and papers on this. Fergusin and Ufondu, 2017 examined all available data. A simple search for 'Nisku Reservoir' returns over 2000 results on Google Scholar. In addition to extensive petroleum industry research, these reservoirs have also been examined for CO2 storage, and as such there has been also extensive work done to characterise hydrogeological properties for that purpose.* | Page 3 line 5 includes "The area around the town site of Hinton in the western region of the Alberta Basin (Fig. 1) is of particular interest because…." I don't see a connection to the reviewer's comment?

Exactly -'hard data' refers to geothermal rock properties like thermal conductivity, thermal diffusivity and heat capacity. This is written in the sentence!
In this study we focused on the **geothermal rock properties**. Previous studies like Grasby et al. 2012 point out a lack of knowledge for thermal conductivity measurements in the study area.
We don't claim that we are the first group doing reseach concerning geothermal energy potential in the study area. Most of the previous studies focus on geothermal gradients and heat flow estimation. We are focusing on rock properties. In my opinion that are different topics covering different scales.
Furthermore, the AccuMap data base does not include detailed core profiles including lithology and rock description. Only a few papers exist in the study areas | We deleted 'and/or petrophysical' in line 10 on page 3 to avoid misinterpretations. |

|  |  | which include detailed rock description.

Again – the fact that the Devonian aquifer systems already have been intensively analyzed is a reason why we started this project. We don't want to repeat analyses which have been carried out before. We want to use the existing data and integrate them into a 3D geological model considering heat flow, temperature, hydraulic and rock properties etc., but also limiting factors like salinity.
Ferguson and Ufondu 2017 definetly did not examined **all** available data (well data from >600000 wells?). They used injection and production rates as well as estimeated temperature data. Until now, no study exist which included all available data. |  |
| **Comment 6** | *Page 4, line 6: This claim is not true, there have been early studies that have done detailed assessments of geothermal potential of Devonian aquifers, including those that the authors now include in the reference list (e.g. Lam and Jones). As well, Fergusin and Ufondu, 2017 also examined Devonian systems. The authors need to do a better job at describing previous work and how their contribution is different and adds to that.* | Page 4 line 6 include "Such analogue studies offer a cost-effective opportunity in areas with a low density of drill holes…" Again, I don't see a connection to the reviewer's comment.

Same answer as above. | See changes above. |

| | | | |
|---|---|---|---|
| **Comment 7** | *Page 5, line 1: British Columbia is not north of Alberta, and as the authors define the WCSB as east of the Rocky Mountains, then it does not extend to the SW into BC as that area is within the Rocky Mountains. Also, I would not say that the portion with in Saskatchewan is 'minor' at all.* | Page 5, line 1 includes: "Most of our work is on these two aquifers. In addition, for comparison we also investigated a small part of a third Devonian aquifer…" I can't identify the connection to the reviewer's comment.

It is written on page 5 line 19: "The Western Canada Sedimentary Basin (WCSB) is a large geological feature that is located mainly in Alberta east of the Rocky Mountains and to a minor extent in the adjacent provinces of Saskatchewan and Manitoba, with marginal excursions into the northern United States to the south and into British Columbia to the southwest, west and north." | We changed line 19 on page 5 as followed
"The Western Canada Sedimentary Basin (WCSB) is a large geological feature that is located mainly in Alberta east of the Rocky Mountains and in the adjacent provinces of Saskatchewan and Manitoba, as well as in the northern United States and in northeastern British Columbia (Grasby et al., 2012)." |
| **Comment 8** | *Page 6, ln 10: These comments have uncertain value, is there a problem with data just because its 'old'? If the data was collected in acceptable means then its perfectly fine. These comments only have value if there are new data measurement techniques that supersede previous work. As well, the outcrops indicated aren't really inaccessible.* | Page 6 line 10 describes the subsurface geology in the study area "which ultimately resulted in the wedge-shaped triangular geometry in cross section of the foreland basin…" I don't think R3 read the actual version of the manuscript?

We don't claim that "old" data is not useful for actual research. However, the divided bar apparatus is less commonly used in the geothermal industry because the method was identified as less accurate/is more error-prone than the methods used in this study. But it is not my aim to degrade previous work.
I don't see any problems adding new data to already existing data sets.

"inaccessible" refers to page 7 line 3 "However, except for a few old studies | No changes needed. |

| | | from remote and almost inaccessible areas such as the Ancient Wall and Miette reef complexes (Mountjoy, 1965, Mountjoy and McKenzie, 1974, Mattes and Mountjoy, 1980), only one 'modern' study is available that provided data on the diagenetic alteration of outcrops in this region, i.e., from Nigel Peak (Köster et al., 2008)". The sentence before clearly says that the outcrop we choose or found where relatively accessible which is necessary for sampling large rock samples. | |

---

## Author Response (AR3)

| Page, Line | Comments from Referees | Author's response | Author's changes in the manuscript |
|---|---|---|---|
| **Editor Comment 1** | Please remove the word "First" in the title | Ok. | The title was changed accordingly to the hint from the editor. |
| **Comment 2** | Add arrows or other symbols into the pictures and photomicrographs to help to reader to identify features you described in text and captions (e.g., fractures and bitumen in Fig. 6; all features the reader is supposed to see in Plates 1-6). | According to the hint of the editor, we will add arrows or other symbols for a better understanding. | Whenever possible arrows or symbols were added to Figure 6 and Plates 1 – 6. In some cases, it was avoided for reasons of clarity and where it was not necessary or reduces the quality of the image. |